# An intermediate Rb–E2F activity state safeguards proliferation commitment

Yumi Konagaya[1,2,3 ✉], David Rosenthal[1], Nalin Ratnayeke[1,2], Yilin Fan[2,4] & Tobias Meyer[1,2 ✉]

Tissue repair, immune defence and cancer progression rely on a vital cellular decision between quiescence and proliferation[1,2]. Mammalian cells proliferate by triggering a positive feedback mechanism[3,4]. The transcription factor E2F activates cyclin-dependent kinase 2 (CDK2), which in turn phosphorylates and inactivates the E2F inhibitor protein retinoblastoma (Rb). This action further increases E2F activity to express genes needed for proliferation. Given that positive feedback can inadvertently amplify small signals, understanding how cells keep this positive feedback in check remains a puzzle. Here we measured E2F and CDK2 signal changes in single cells and found that the positive feedback mechanism engages only late in G1 phase. Cells spend variable and often extended times in a reversible state of intermediate E2F activity before committing to proliferate. This intermediate E2F activity is proportional to the amount of phosphorylation of a conserved T373 residue in Rb that is mediated by CDK2 or CDK4/CDK6. Such T373-phosphorylated Rb remains bound on chromatin but dissociates from it once Rb is hyperphosphorylated at many sites, which fully activates E2F. The preferential initial phosphorylation of T373 can be explained by its relatively slower rate of dephosphorylation. Together, our study identifies a primed state of intermediate E2F activation whereby cells sense external and internal signals and decide whether to reverse and exit to quiescence or trigger the positive feedback mechanism that initiates cell proliferation.

The activity of the main upstream regulators of E2F, such as ERK, AKT and p53, frequently fluctuate in individual cells[5]. This pattern raises the question of how cells ensure that the positive feedback between E2F and CDK2 does not cause excessive proliferation by unwanted amplification of small increases in E2F activity (Fig. 1a). To answer this question, we developed an E2F transcription activity reporter by testing various promoter sequences of E2F targets. We conjugated these sequences to a nuclear-localized, short-lived fluorescent protein, mVenus, and selected an E2F reporter based on the promoter from the licensing factor CDC6 (ref. 6) (Fig. 1b–d and Extended Data Fig. 1a–e; see Methods for details on development of the E2F reporter). We used several different methods to validate that the reporter selectively measures E2F activity (Extended Data Figs. 1f–n and 2a–k and Methods).

### CDK4–CDK6 and CDK2 redundantly activate E2F

We studied the regulation of E2F activation by arresting MCF-10A human epithelial cells through mitogen removal followed by mitogen release (Fig. 1c,d). The expression of exogenous cyclin D1 accelerated the increase in E2F activity (Fig. 1e), and this increase was inhibited by palbociclib (also known as PD-0332991). Palbociclib is a selective inhibitor of cyclin D and CDK4–CDK6 (CDK4/6) and is used for the treatment of various cancers[2]. Treatment with the recently developed CDK2-selective inhibitor PF-07104091 (Extended Data Fig. 1m), which

is currently in clinical trials, also delayed the increase in E2F activity. Only simultaneous inhibition of CDK2 and CDK4/6 could fully suppress the mitogen-induced increase in E2F activity (Fig. 1f), which provided confirmation that either CDK2 or CDK4/6 can activate E2F. If these kinases are active, they are synergistically activating E2F[7,8].

We next added a component of the Fucci(CA) reporter system to also monitor the start of S phase in the same cells[9]. CDK4/6-inhibited cells activated E2F and entered S phase more slowly than in control cells (Fig. 1g), which confirmed that cyclin D–CDK4/6 activation shortened the length of the G1 phase[10]. Cultured cells generally have high CDK4/6 activity through optimized growth conditions or acquisition of mutations[2], which is useful property for experiments because it accelerates the cell division rate. Therefore, CDK4/6 inhibition has been used to slow cell division and mimic the typically longer G1 phases observed in vivo[7,11,12]. Thus, in most conditions, we included CDK4/6 inhibitors or low mitogen stimuli to slow cell cycle progression.

### Reversible, intermediate E2F activation

To monitor the activity of CDK2 and E2F in the same cell, we added a stably expressed cyclin E/A-CDK reporter[13] that measures the combined activity of cyclin E–CDK2 and cyclin A–CDK2/1. As cyclin A remains degraded until late G1, the reporter measures primarily cyclin E–CDK2 activity in G1[13,14].

[1]Department of Cell and Developmental Biology, Weill Cornell Medicine, New York, NY, USA. [2]Department of Chemical and Systems Biology, Stanford University School of Medicine, Stanford, CA, USA. [3]Laboratory for Quantitative Biology of Cell Fate Decision, RIKEN Center for Biosystems Dynamics Research, Kobe, Hyogo, Japan. [4]Department of Pathology and Center for Cancer Research, Massachusetts General Hospital and Harvard Medical School, Boston, MA, USA. ✉e-mail: yumi.konagaya@riken.jp; tom4003@med.cornell.edu

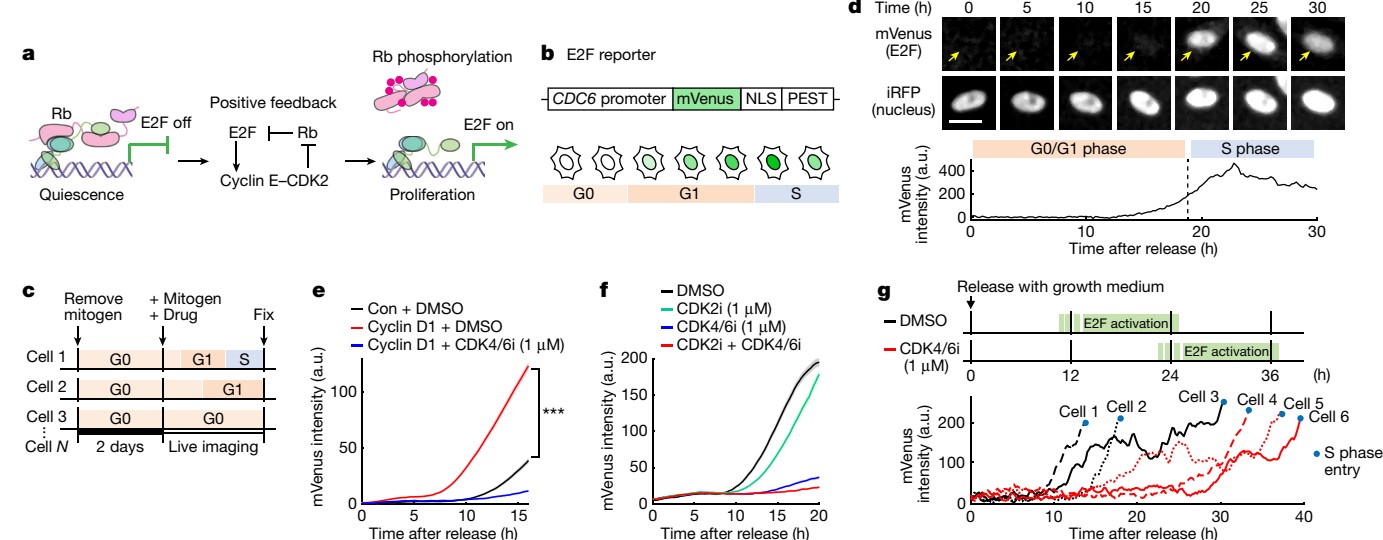

**Fig. 1 | A marked variation in the activation kinetics of E2F revealed by live cell E2F transcription analysis. a**, Model of the cell cycle decision process in G1 phase. **b**, Schematic of the E2F transcriptional reporter. **c**, Experimental design for mitogen release in MCF-10A cells. **d**, Top, time-course images of a cell expressing mVenus (E2F reporter) and H2B–iRFP (nucleus marker), starved for 2 days and released with growth medium. Arrows indicate cell nuclei. Scale bar, 20 μm. Bottom, mVenus intensity (in arbitrary units (a.u.)) in the cell shown in the top (1 out of 3 biological replicates). **e**, mVenus intensity traces in control (Con) or doxycycline-inducible HA-tagged cyclin D1-expressing cells. Doxycycline was added 5 h before release to induce cyclin D1. Cells were released with starvation medium + EGF (20 ng ml⁻¹). Cyclin D1-expressing cells were released with DMSO or CDK4/6 inhibitor (CDK4/6i; 1 μM palbociclib). HA > 2¹² was used to gate cyclin D1-overexpressing cells. $P = 2.8 \times 10^{-92}$ (mVenus intensity 16 h after release), calculated using two-sided, two-sample $t$-tests (mean ± s.e.). $n = 500, 500$ and 456 cells for Con + DMSO, cyclin D1 + DMSO and cyclin D1 + CDK4/6i, respectively; 1 out of 3 biological replicates. **f**, mVenus intensity traces after release with growth medium. Cells were released with DMSO, CDK2 inhibitor (CDK2i; 1 μM PF-07104091), CDK4/6i or CDK2i + CDK4/6i (mean ± s.e.). $n = 300$ cells per condition; 1 out of 4 biological replicates. **g**, Top, schematic of E2F activation in cells released with DMSO versus CDK4/6i. Bottom, single-cell traces of mVenus intensity after release with growth medium + DMSO or CDK4/6i.

CDK2 activity increased before E2F activity in the CDK4/6-inhibited condition. Both activities then fluctuated in G1 phase before they reached higher levels at the start of S phase[7] (Fig. 2a,b (with palbociclib) and Extended Data Fig. 3a–d (without palbociclib as a comparison and with or without CDK2 inhibition to confirm CDK2 contributes to the maintenance of E2F activity)). The induced exogenous expression of cyclin E1 was sufficient to activate CDK2 and E2F and to start S phase in the CDK4/6-inhibited condition[7,15] (Extended Data Fig. 3e–h).

To evaluate when the positive feedback mechanism engages, we compared the dynamics of E2F and CDK2 activity among cells that are variable in G1 length. This was achieved by computationally aligning hundreds of single-cell traces to the time of S start by grouping cells on the basis of their G1 length and averaging the activities in each group (Fig. 2c,d (yellow to red) and Extended Data Fig. 3i,j). This analysis showed that cells can stall for long periods in an intermediate E2F activity state but can then start a continued and parallel increase in both E2F and CDK2 activities shortly before S phase. This parallel increase had the characteristic of the proposed positive feedback between CDK2 and E2F[3,4]. However, this positive feedback fully engaged only when both E2F and CDK2 activity levels were already high late in G1 phase (Fig. 2d). In further support of a late-engaging positive feedback mechanism, the time it took for CDK2 activity to reach an intermediate level of 0.65 was highly variable, whereas the time from this intermediate level to the start of S phase was less variable (Fig. 2e).

We confirmed that the intermediate level of E2F activity corresponds to an intermediate level of expressed transcripts of the E2F target gene *CDC6* (Extended Data Fig. 3k). Interestingly, out of this state of intermediate E2F activation, a subset of cells decreased their E2F and CDK2 activity and returned to quiescence (termed E2F reverse; Fig. 2f,g). We observed such E2F reverse cells in all conditions and more frequently following stimulation with low concentrations of epidermal growth factor (EGF) (Fig. 2f, right). E2F reverse cells that stalled

in G1 were similarly observed in another cell line, RPE-1 cells, and in serum-released and asynchronously cycling MCF-10A cells without CDK4/6 inhibition (Fig. 2h and Extended Data Fig. 4a–e). Thus, variable intermediate E2F activation and reversal in G1 are probably general cellular behaviours.

To test whether cells in the state of intermediate E2F activation remain under the control of growth factor receptors, we inhibited EGFR signalling by treating cells with the EGFR inhibitor gefitinib (also known as ZD1839). Treatment led to a reduction in both CDK2 and E2F activities in G1 phase (Fig. 2i,j, Extended Data Fig. 5a–m (these figures explain a brief and slight increase in CDK2 activity after EGFR inhibition) and Supplementary Fig. 1 (original western blots)). However, after entering S phase, cells failed to reverse CDK2 activity after gefitinib addition (Fig. 2k), a result consistent with cells irreversibly committing to the cell cycle in late G1 (refs. 8,14,16). Thus, cells can spend time in a primed G1 state of intermediate E2F activity and of variable duration, during which external and internal signals keep being sensed. An appropriate trigger then ultimately induces cells to revert to quiescence or engage the positive feedback mechanism and commit to proliferate.

## Preferential Rb phosphorylation at T373

The finding of an intermediate E2F activity state was difficult to reconcile with the proposed all-or-none regulation of E2F and its inhibitor Rb (encoded by *RB1*)[3,4,17]. Biochemical studies have shown that the phosphorylation of Rb is highly cooperative in hyperphosphorylation (phosphorylation at 15 serine and threonine (Ser/Thr) sites in Rb), switching Rb from an active state to an inactive state. This inactivation of Rb in turn induces the activation of E2F[18,19]. It was also reported that CDK4/6 activity in quiescence or early G1 can stochastically monophosphorylate Rb at any one of these Ser/Thr sites, without activating E2F[20],

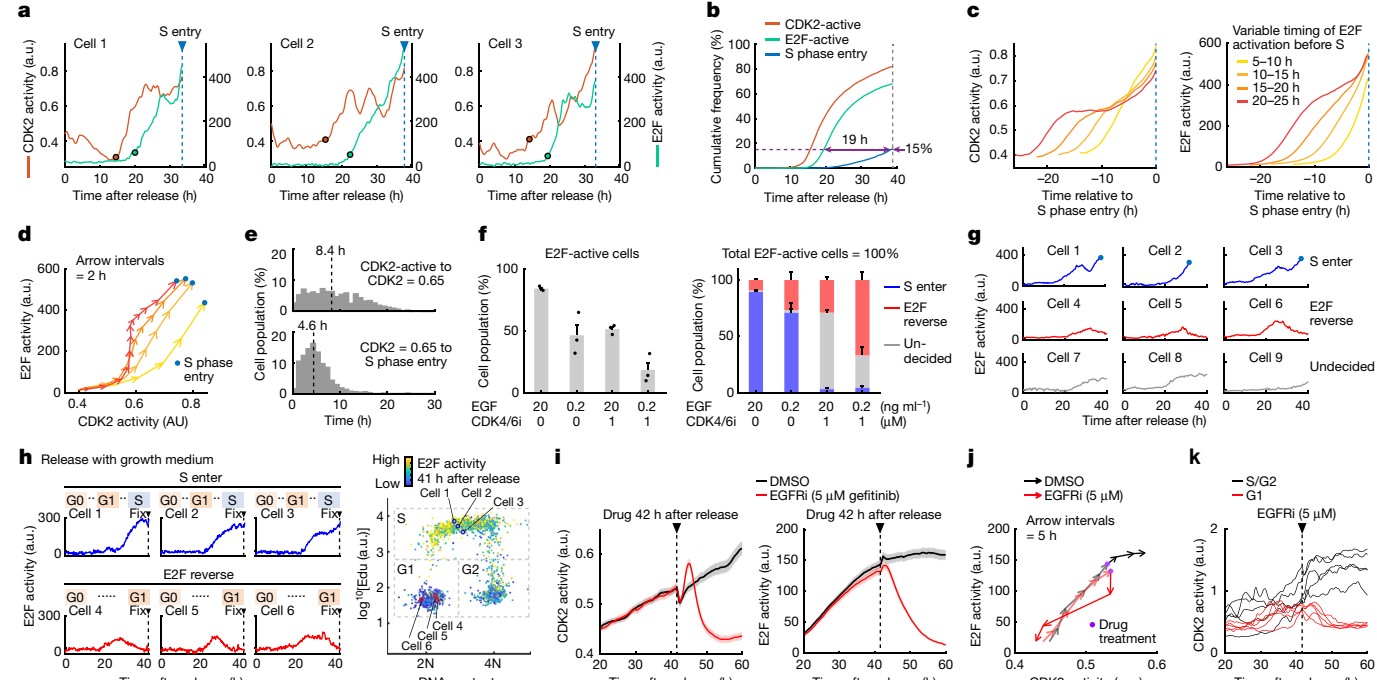

**Fig. 2 | E2F and CDK2 activities stay at intermediate levels and remain reversible before the positive feedback that starts S phase entry fully engages. a**, Single-cell traces of CDK2 and E2F activities. Circles, CDK2-activated or E2F-activated timing. **b**, Cumulative frequency of CDK2-activated, E2F-activated and S phase-entered cells. *n* = 5,969 cells; 1 out of 3 biological replicates. **c**, Cell traces were computationally aligned at S phase entry and stratified based on the variable time cells spend from E2F-active to S phase entry. CDK2 and E2F activity traces (mean per cell population). *n* = 244, 246, 226 and 125 cells for 5–10 h, 10–15 h, 15–20 h and 20–25 h, respectively; 1 out of 3 biological replicates. For **a–c**, conditions were release with growth medium + CDK4/6i (1 μM). **d**, Data in **c** plotted as a phase-plane trajectory. **e**, Data in **c** analysed for the variable time cells spend from CDK2-active to CDK2 activity = 0.65 (top), and from CDK2 activity = 0.65 to S phase entry (bottom). Dashed lines indicate the median. **f**, Left, percentage of cells with E2F activation by 40 h after release with starvation medium + EGF (20 or 0.2 ng ml⁻¹) ± CDK4/6i

(1 μM). Right, percentage of S enter, E2F reverse and undecided cells among E2F-activated cells (mean ± s.e.). Cells were categorized based on behaviours until 40 h after release (see Methods for more detail). *n* = 2,655, 3,611, 1,685 and 3,042 cells for EGF 20, EGF 0.2, EGF 20 + 4/6i and EGF 0.2 + 4/6i, respectively; 3 biological replicates. **g**, Single-cell traces of E2F activity in cells categorized in **f**. **h**, Single-cell traces of E2F activity in S enter and E2F reverse RPE-1 cells (left), determined based on DNA content versus 5-ethynyl-2′-deoxyuridine (EdU) incorporation at the end of live-cell imaging (right). One out of 3 biological replicates. **i**, CDK2 and E2F activity traces (mean ± s.e.) after release with starvation medium + EGF (20 ng ml⁻¹) + CDK4/6i (1 μM). *n* = 534 (DMSO) and 476 (EGFR inhibitor (EGFRi)) cells; 1 out of 3 biological replicates. **j**, Data in **i** plotted as a phase-plane trajectory. **k**, Single-cell traces of CDK2 activity before and after EGFRi. G1 and S/G2 cells in **i** were gated based on the CRL4^Cdt2 reporter signal. *n* = 5 cells each; 1 out of 3 biological replicates.

or regulate differential transcription programs[21]. Moreover, E2F is inhibited by two interactions between Rb and E2F: (1) the Rb pocket domain (RbP) interacts with the E2F transactivation domain (E2F(TD)), and (2) the Rb carboxy-terminal domain (RbC) interacts with the marked box (MB) domains of E2F and its dimerization partner DP (E2F(MB)–DP(MB)) (Fig. 3a). These interactions were identified in biochemical[22,23] and structural[24–29] studies and was observed in a computationally predicted structure that we generated using the deep-learning-based AlphaFold algorithm[30] (Fig. 3b). Mutagenesis analyses have shown that phosphorylation of Rb at T373, S608 or S612 selectively regulates the interaction between RbP and E2F(TD)[27,28], whereas phosphorylation of S788, S795, T821 or T826 regulates a different interaction between RbC and E2F(MB)–DP(MB)[26,29]. These structural findings motivated us to consider whether cells may initially phosphorylate a subset of Rb sites to disrupt one of these two inhibitory interactions and thereby increase E2F activity to an intermediate level.

To test for potential site-specific regulation of Rb phosphorylation in cells, we performed multiplexed single-cell immunofluorescence analysis[31] (Methods). To measure the relative phosphorylation of specific Rb sites, each phosphorylation signal was normalized by the total Rb antibody signal in the same cell. As a reference, we compared the partial phosphorylation of a given site to that of the C-terminal S807 and S811 (S807/S811) sites in the same cell. This analysis showed that the phosphorylation relationship in G1 phase is convex towards T373

over S807/S811, which implied that Rb proteins within each cell are preferentially phosphorylated at T373 over S807/S811 (Fig. 3c, top left).

We next quantitatively determined the preference between two phosphorylation sites using a phosphorylation site preference (PSP) analysis (see Methods for details on the PSP analysis and the PSP_coeff). The analysis showed that the T373 site was initially phosphorylated with a relative preference of about a factor of 5 over S807/S811 (Fig. 3c,d).

Several control experiments were performed to confirm results. We showed that T373 is phosphorylated before S807/S811 by analysing T373 and S807/S811 phosphorylation simultaneously in single cells in a time-course analysis following EGF stimulation (Fig. 3d). We validated the specificity of the phospho-T373 antibody by expressing a Rb that had T373 mutated to alanine and one that had all phosphorylation sites in Rb mutated except for T373 (Extended Data Fig. 6a,b). We also tested the reproducibility of the multiplexed imaging method by changing the order of antibody staining[31] (Extended Data Fig. 6c,d). Moreover, we tested different conditions and consistently observed that T373 is preferentially phosphorylated before C-terminal sites in asynchronously cycling MCF-10A cells with or without a DNA damage agent or with varied strengths of mitogen stimulation (Extended Data Fig. 7a–e). We also tested other cell types and found the same T373 preference in two non-transformed cell lines, RPE-1 and BJ-5ta cells, and in a transformed cell line, U2OS cells (Extended Data Fig. 8a–c).

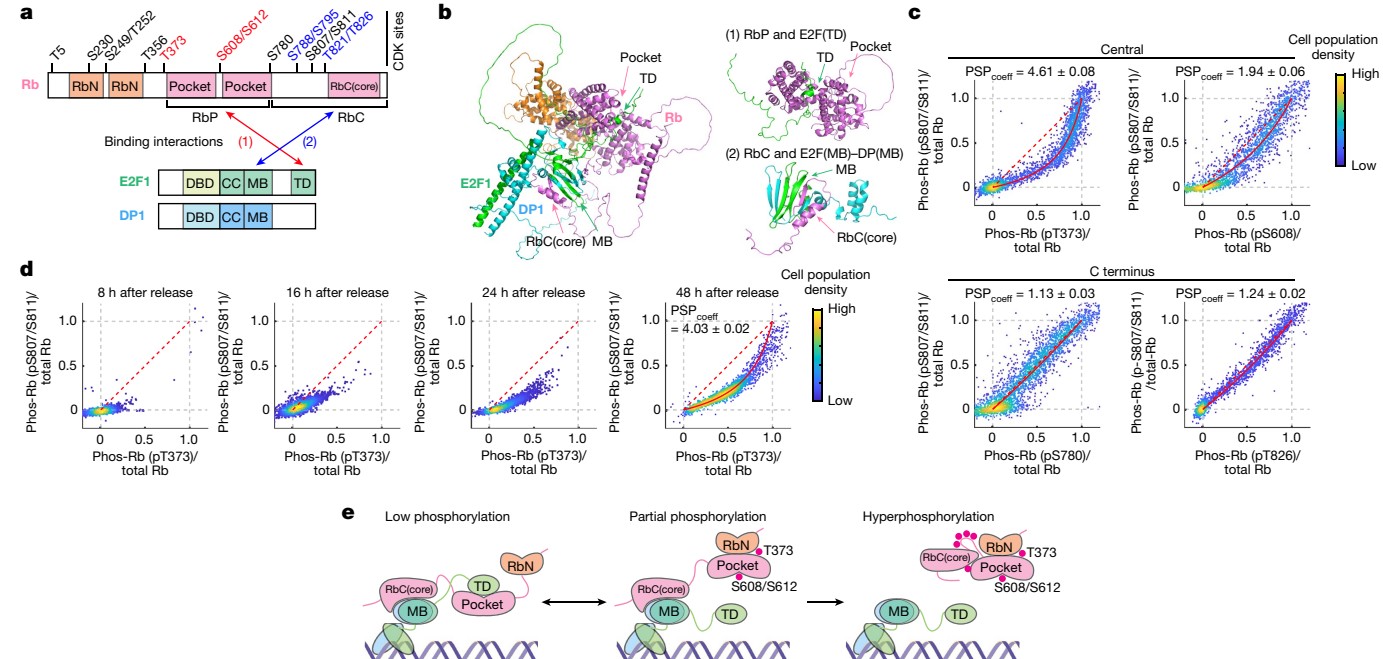

**Fig. 3 | Rb is first phosphorylated at T373 before its C-terminal sites.**
**a**, Domain architecture of Rb, E2F1 and DP1. Rb consists of a structured amino-terminal domain (RbN) and RbP. Its RbC is disordered except for a short sequence (RbC(core)) that adopts a structure after E2F binding. CC, coiled-coil domain; DBD, DNA-binding domain. **b**, Left, Rb–E2F1–DP1 complex prediction with AlphaFold. Rb(ΔCDK(43–928)) (CDK phosphorylation sites mutated to alanines to mimic unphosphorylated Rb), E2F1(200–437), and DP1(198–410) were used to predict the structure of the complex. Right, two interaction sites between Rb and E2F1–DP1: (1) RbP interacts with E2F(TD) and (2) RbC interacts with E2F(MB)–DP(MB). **c,d**, PSP plots showing single-cell correlation of Rb phosphorylation (phos-Rb) between two different sites 16 h after release with starvation medium + EGF (20 ng ml$^{-1}$) + CDK4/6i (20 nM) (**c**) or release after starvation medium + EGF (20 ng ml$^{-1}$) + CDK4/6i (1 µM) (**d**). Each phosphorylation signal was normalized by the total Rb antibody signal in the same cell and each axis was adjusted to the average phosphorylation signal in S phase of 1 (when Rb is hyperphosphorylated). A red line shows fitting with a preferential relative phosphorylation–dephosphorylation rate between the two sites (PSP$_{coeff}$) (see Methods for more details). **c**, Rb phosphorylation at T373 (pT373), S608 (pS608), S780 (pS780) and T826 (pT826) plotted against S807/S811 (pS807/S811). $n$ = 2,734, 2,159, 2,684 and 2203 cells for T373, S608, S780 and T826, respectively; 1 out of 3 biological replicates. **d**, Rb phosphorylation at T373 plotted against S807/S811. Cells were fixed 8, 16, 24 and 48 h after release. $n$ = 2,531, 2,655, 2,713 and 2,774 cells for 8 h, 16 h, 24 h and 48 h, respectively; 1 out of 2 biological replicates for 8 h, 3 biological replicates for 16 h, 4 biological replicates for 24 and 48 h. **e**, Model for phosphorylation and inactivation of Rb in a two-step process. First, Rb is phosphorylated at T373 and S608/S612, which probably disrupts (1) the RbP–E2F(TD) interaction. Second, Rb phosphorylation at C-terminal sites disrupts (2) the RbC and E2F(MB)–DP(MB) interaction, leading to full release of Rb from E2F.

In addition, Rb S608, which regulates the inhibitory interaction between RbP and E2F(TD), was phosphorylated earlier than S807/S811 (Fig. 3c, top right). By contrast, the C-terminal phosphorylation sites S780 and T826, which regulate the interaction between RbC and E2F(MB)–DP(MB), were phosphorylated along with the C-terminal S807/S811 sites (Fig. 3c, bottom two panels). Thus, the phosphorylation and inactivation of Rb is a two-step process (Fig. 3e). In the first step, Rb is phosphorylated at T373 and S608, which probably disrupts the RbP and E2F(TD) interaction without disrupting the RbC and E2F(MB)–DP(MB) interaction. The full release of Rb from E2F is therefore expected to occur only in the second step, when the C-terminal S780, S807/S811 and T826 sites become jointly phosphorylated as part of the hyperphosphorylation of Rb.

## Gradual phosphorylation at T373 by CDK2

To understand how CDK2 regulates Rb phosphorylation, we correlated CDK2 activity and Rb phosphorylation of T373 and S807/S811 in the same fixed cell. The analysis of thousands of single cells with different CDK2 activities showed that Rb S807/S811 is phosphorylated over a narrow range of already high CDK2 activity (Fig. 4a,b). This dependency could be fit to a sigmoidal distribution with a Hill coefficient of 5.81 ± 0.44. This result is consistent with earlier studies showing that phosphorylation of S807/S811 is part of a cooperative Rb hyperphosphorylation mechanism[32,33]. Similarly, another C-terminal phosphorylation site that we tested, T826, showed the same high Hill coefficient as S807/S811 (Extended Data Fig. 8d).

By contrast, T373 phosphorylation occurred at lower CDK2 activities and increased more gradually along with CDK2 activity (Fig. 4a,b). We confirmed that CDK2 phosphorylates T373 using the CDK2 inhibitor PF-07104091 (Fig. 4c).

## T373 link to intermediate E2F activity

To understand how Rb phosphorylation regulates E2F activity, we next correlated relative Rb phosphorylation and E2F activity by combining live E2F reporter analysis with multiplexed immunostaining. This approach can map the degree of Rb phosphorylation at a specific site to the corresponding E2F activity in the same cell. Analyses of thousands of cells showed that the phosphorylation of T373 and S608 initially gradually increased along with E2F activity (Fig. 4d, left two panels). By contrast, the phosphorylation of the C-terminal S807/S811 and T826 sites only substantially increased after E2F had already increased to an intermediate level (Fig. 4d, right two panels). Together, these single-cell measurements suggest that E2F is initially gradually activated by increasing CDK2 activity proportional to the fraction of Rb phosphorylated at T373 and S608. In the second step, E2F becomes

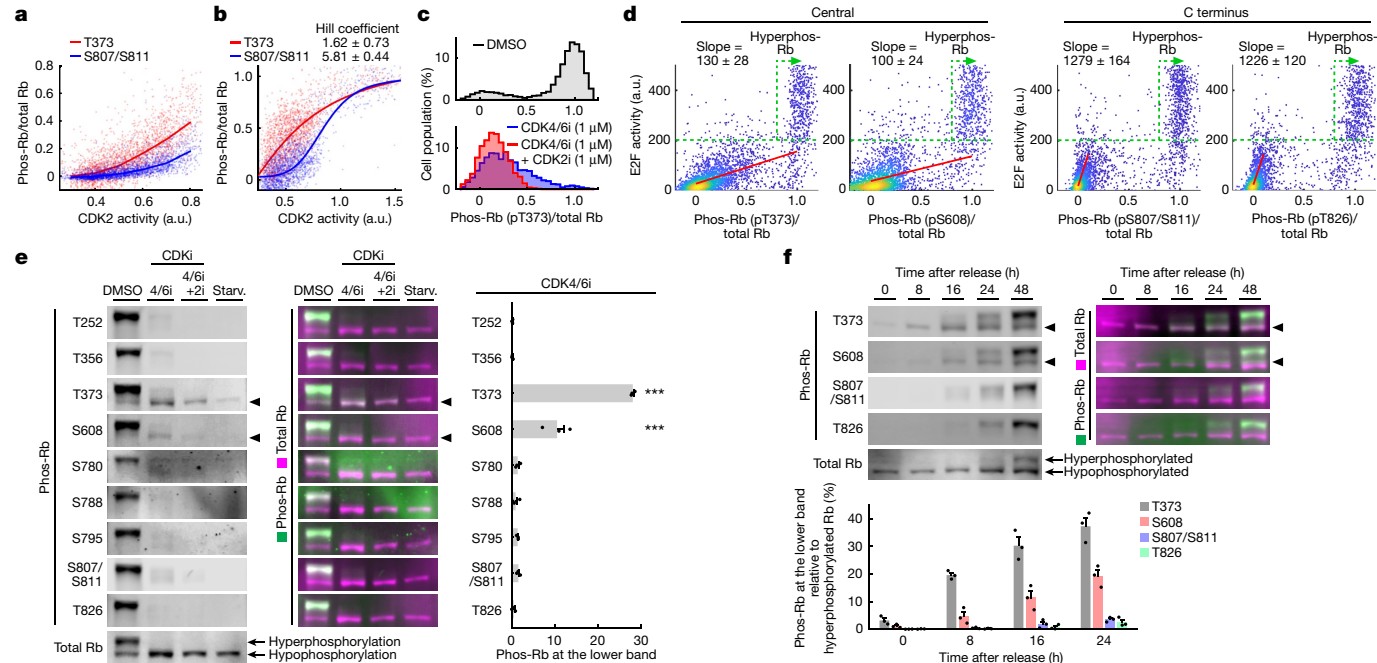

**Fig. 4 | The degree of Rb phosphorylation at T373 is proportional to E2F activity during the intermediate E2F activation state. a,b**, Single-cell correlation of CDK2 activity and Rb phosphorylation under starvation medium conditions + EGF (20 ng ml⁻¹) + CDK4/6i (1 µM), 24 h (**a**) or 48 h after release (**b**). **a**, Lines indicate median Rb phosphorylation. $n = 2,535$ cells; 1 out of 3 biological replicates. **b**, Lines indicate sigmoidal fit curves. $n = 1,967$ (T373) and 1,867 (S807/S811) cells; 1 out of 4 biological replicates. **c**, Histograms of Rb phosphorylation at T373 24 h after release with starvation medium + EGF (20 ng ml⁻¹). $n = 2,416$, 2,630 and 2,440 cells for DMSO, CDK4/6i and CDK4/6i + CDK2i, respectively; 1 out of 3 biological replicates. **d**, Single-cell correlation of Rb phosphorylation versus E2F activity 24 h after release with starvation medium + EGF (20 ng ml⁻¹) ± CDK4/6i (1 µM). Red lines indicate Deming regression lines. Slope indicates mean ± s.e. $n = 3,496$, 2,999, 3,197 and 2,881 cells for T373, S608, S807/S811 and T826 respectively; 1 out of 2 technical replicates and 2 biological replicates. **e,f**, Western blots of phospho-RB and total Rb, and quantification of the lower Rb phosphorylation band (mean ± s.e. from 3 biological replicates). Arrowheads indicate lower Rb phosphorylation

band (1 out of 3 biological replicates). **e**, Cells released with DMSO, CDK4/6i or CDK4/6i + CDK2i (10 µM) were assayed 16, 24 and 24 h after release with starvation (Starv.) medium + EGF (20 ng ml⁻¹), respectively. Starved cells were assayed before release. To account for the different phospho-antibody affinities, lower Rb phosphorylation bands in the 4/6i lane were normalized by the upper Rb phosphorylation bands in the DMSO lane. $P$ values were calculated using one-way analysis of variance (ANOVA) and Scheffé's post hoc comparison. T373: $P = 1.2 \times 10^{-13}$ (vs T252), $P = 1.2 \times 10^{-13}$ (vs T356), $P = 2.6 \times 10^{-13}$ (vs S780), $P = 1.9 \times 10^{-13}$ (vs S788), $P = 2.6 \times 10^{-13}$ (vs S795), $P = 2.9 \times 10^{-13}$ (vs S807/S811), $P = 1.5 \times 10^{-13}$ (vs T826). S608: $P = 1.6 \times 10^{-6}$ (vs T252), $P = 1.6 \times 10^{-6}$ (vs T356), $P = 1.0 \times 10^{-5}$ (vs S780), $P = 4.9 \times 10^{-6}$ (vs S788), $P = 1.0 \times 10^{-5}$ (vs S795), $P = 1.3 \times 10^{-5}$ (vs S807/S811), $P = 2.6 \times 10^{-6}$ (vs T826). **f**, Cells were assayed 0, 8, 16, 24 and 48 h after release with starvation medium + EGF (20 ng ml⁻¹) + CDK4/6i (1 µM), respectively. To account for the different phospho-antibody affinities, the lower Rb phosphorylation bands 0–24 h after release were normalized by the upper Rb phosphorylation bands 48 h after release.

maximally activated after Rb hyperphosphorylation once CDK2 activity has reached a high level in late G1.

## Sequential Rb phosphorylation

Rb undergoes a concerted conformational change in G1 phase that can be analysed by western blot analysis[8,18–20,32]. A hypophosphorylated state of Rb can be distinguished from a hyperphosphorylated state as a lower and an upper band of Rb, respectively (Fig. 4e). We used nine phosphosite-specific Rb antibodies suitable for western blot analysis and confirmed that phosphorylation at all sites, including T373, could be detected in the upper Rb band (in control cells released from serum arrest and rapidly entered G1; Fig. 4e, DMSO lane, and Supplementary Fig. 1). In cells that were released in the presence of a CDK4/6 inhibitor and entered more slowly (Fig. 4e, 4/6i lane), we also detected in the lower band phosphorylation at the T373 and S608 sites but not at the C-terminal S780, S788, S795, S807/S811 or T826 sites. This T373 and S608 phosphorylation in the lower Rb band was inhibited by CDK4/6 and CDK2 inhibitors (Fig. 4e, 4/6i + 2i lane), and was almost undetectable in cells starved in serum-free medium for 2 days (before release; Fig. 4e, Starv. lane).

Furthermore, results from the western blot time-course analysis supported that Rb is first phosphorylated at T373, followed by S608

and last by the C-terminal sites (Fig. 4f and Supplementary Fig. 1). This result strengthens the finding from the immunostaining analysis that the T373 and S608 sites of Rb are phosphorylated first, and further demonstrated that the large conformational change of Rb only occurs once Rb is hyperphosphorylated after the initial phosphorylation of T373 and S608.

## Slower dephosphorylation at Rb T373

The sequential phosphorylation of Rb could arise from CDK2 preferentially phosphorylating T373 or from phosphatase activity preferentially dephosphorylating S807/S811. Rb is dephosphorylated mostly by PP1 and PP2A[34]. To test for a potential phosphatase preference, we measured the time course of Rb dephosphorylation after inhibiting CDK2 activity through PF-07104091 addition to cells in G1 phase (under CDK4/6-inhibited conditions). Notably, this acute kinase inhibition resulted in a significantly slower dephosphorylation half-life of about 40 min for T373 and 15 min for S608 compared with about 6 min for the C-terminal S807/S811 and T826 sites (Fig. 5a). The total level of Rb did not significantly change after CDK2 inhibition (Fig. 5b).

To validate that the PP1 and PP2A phosphatases are mediating Rb dephosphorylation[34], we applied calyculin A, a competitive inhibitor for substrate binding[35] with a similar potency for PP1 and PP2A[36].

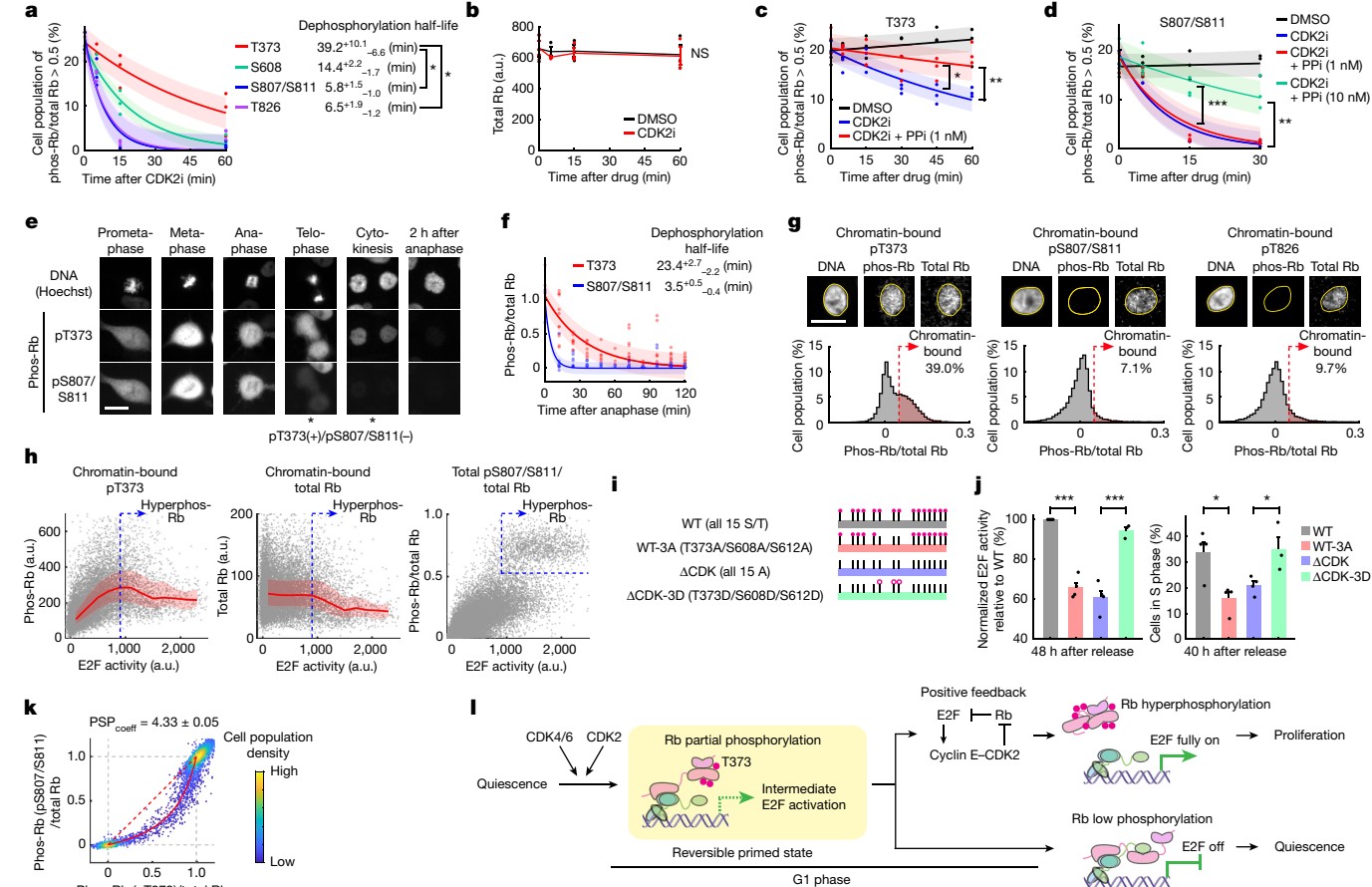

**Fig. 5 | Regulation and function of T373 phosphorylated Rb. a–d,** Exponential decay fitting (mean ± s.e. from 3 biological replicates). Cells were treated with CDK2i (20 μM) 42 h after release with starvation medium + EGF (20 ng ml⁻¹) + CDK4/6i (1 μM). One-way ANOVA and Scheffé's post hoc comparison (**a**) or two-sided, two-sample *t*-tests (**b–d**). **a**, *P* = 0.042 (vs S807/S811), *P* = 0.046 (vs T826). **b**, *P* = 0.91 (not significant (NS)). **c,d,** Cells were pre-treated with the PPi calyculin A (1 nM 60 min or 10 nM 30 min before CDK2i). T373: *P* = 0.039 (45 min), *P* = 4.5 × 10⁻³ (60 min). S807/S811: *P* = 7.1 × 10⁻⁴ (10 nM, 15 min), *P* = 6.3 × 10⁻³ (10 nM, 30 min). **e,f,** Asynchronously cycling cells in starvation medium + EGF (0.2 ng ml⁻¹) were fixed 16 h after 50 ng ml⁻¹ neocarzinostatin for 20 min. **e**, Images at different cell cycle stages during mitosis. Scale bar, 20 μm. **f**, Exponential decay fitting (mean ± s.e. from 3 biological replicates). Cells born 1–16 h after neocarzinostatin treatment were computationally aligned at anaphase. *n* = 1,160 cells. **g**, Representative images of MCF-10A cells stained for DNA (Hoechst), chromatin-bound phospho-Rb, and chromatin-bound total-Rb (top), and histograms of pre-extracted phospho-Rb/total-Rb (bottom). Soluble protein was pre-extracted 36 h after release with growth medium + CDK4/6i

(1 μM). Scale bar, 20 μm. *n* = 14,385, 13,341 and 14,578 cells for T373, S807/S811 and T826, respectively; 1 out of 3 biological replicates. **h**, Single-cell correlation of E2F activity and pre-extracted pT373 (left), pre-extracted total-Rb (middle) and total (without pre-extraction) pS807/S811/total Rb (right) 36 h after release with growth medium + CDK4/6i (1 μM). Red lines indicate median values ± 25th and 75th percentiles. *n* = 14,385 (left and middle) and 22,615 cells (right); 1 out of 3 biological replicates. **i**, Schematic of doxycycline-inducible HA-tagged Rb constructs. **j**, E2F activity (left) and cells in S phase (right) after release with growth medium + CDK4/6i (1 μM). Mean ± s.e. from 4 biological replicates, except ΔCDK-3D (3 biological replicates). Endogenous Rb was knocked down 1 day before release and exogenous Rb was doxycycline-induced 5 h before release. Cells with 2¹⁰ < HA < 2¹¹ were selected for analysis. Two-sided, two-sample *t*-tests. E2F activity: *P* = 1.8 × 10⁻⁵ (WT vs WT-3A), *P* = 9.2 × 10⁻⁴ (ΔCDK vs ΔCDK-3D). S phase cells: *P* = 0.013 (WT vs WT-3A), *P* = 0.036 (ΔCDK vs ΔCDK-3D). **k**, PSP plots of cells 16 h after release with starvation medium + EGF (20 ng ml⁻¹) + DMSO (without CDK4/6i). *n* = 3,580 cells; 1 out of 3 biological replicates. **l**, Model for a reversible primed G1 state.

The CDK2 inhibitor-induced dephosphorylation of T373 was blocked by 1 nM of calyculin A (Fig. 5c). Similarly, dephosphorylation at S807/S811 was blocked by calyculin A, but only at a higher dose (Fig. 5d), which potentially reflects the preference of the phosphatases for these sites. We conclude that the combined phosphatase activity has a 6.79 ± 2.15-fold preference in dephosphorylating S807/S811 over T373.

A kinetic model showed that the difference in the relative dephosphorylation rate (6.79 ± 2.15, s.e.) could explain the phosphorylation site preference from the PSP analysis (PSP_coeff of 4.61 ± 0.08, s.e.) (Fig. 3c; details in Methods). Although these results do not exclude the possibility that kinase selectivity could contribute to preferential phosphorylation, the simplest explanation is that the intermediate E2F activity state is the result of PP1 and PP2A preference for S807/S811 over T373.

We also measured the rate of Rb dephosphorylation after anaphase in mitosis when CDKs become rapidly inactivated. Notably, the

dephosphorylation of T373 was 6.65 ± 1.05-fold slower than that of S807/S811 (Fig. 5e,f). This result further supports that the T373 phosphorylation preference is based on preferential phosphatase activity for S807/S811 over T373.

## T373 phosphorylated Rb stays on chromatin

Rb phosphorylation at T373, which disrupts the RbP inhibitory interaction with E2F(TD), is induced earlier than C-terminal phosphorylation, which disrupts the RbC interaction with E2F(MB)–DP(MB). We therefore predicted that T373 phosphorylated Rb might remain bound to chromatin through the RbC and E2F(MB)–DP(MB) interaction. We tested this prediction by single-cell immunofluorescence analysis after pre-extracting the soluble fraction of Rb, which leaves behind the chromatin-bound Rb. The Rb T373 phosphorylation signal was still

detected in the nucleus after extraction of soluble proteins. By contrast, the signals from Rb with phosphorylated S807/S811 and T826 were lost after extraction, as has been previously shown for hyperphosphorylated Rb[37,38] (Fig. 5g). Thus, Rb can be phosphorylated at T373 and stay bound to chromatin as long as Rb is not hyperphosphorylated.

We next evaluated whether the amount of chromatin-bound T373 phosphorylated Rb correlates with E2F activity. We analysed the relationship by mapping the chromatin-bound T373 phosphorylation signal to the E2F activity measured by live-cell imaging in the same cell before extracting the soluble proteins. E2F activity initially increased to an intermediate level in proportion to the chromatin-bound Rb phosphorylated at T373 (Fig. 5h, left). We then used the E2F activity threshold above which Rb becomes hyperphosphorylated (marked by the S807/S811 phosphorylation of Rb; Fig. 5h, right, above the blue dashed line) and dissociates from chromatin (Fig. 5h, middle). Consistent with the prediction that Rb is chromatin-bound only during the intermediate E2F activity state during which Rb is not yet hyperphosphorylated, T373 phosphorylated Rb started to dissociate from chromatin at E2F activities above this threshold (Fig. 5h, left, above the blue dashed line). Together, these data suggest that E2F is initially activated up to intermediate levels in proportion to the degree of T373 phosphorylation of chromatin-bound Rb and later fully activated when Rb dissociates from chromatin after its hyperphosphorylation.

## T373 phosphorylation activates E2F

Phosphorylation of the central T373, S608 or S612 sites in Rb can disrupt the interaction between E2F(TD) and RbP[27,28]. We did not examine S612 in our experiments owing to the lack of a phospho-specific antibody. To determine the combined function of these sites, we developed an assay to measure the inhibitory capacity of expressed HA-tagged Rb mutants (Extended Data Fig. 9a,b validates *RB1* knockdown and Rb re-expression). We measured HA-tag staining at the end of the experiment to select and analyse cells that had the same range of expression of the respective Rb mutant. We used the culture condition and time point in which most of the unperturbed cells are in the intermediate E2F activity state. We first compared the effect of expressing wild-type (Rb(WT)) and a mutant in which all 15 CDK phosphophorylation sites were mutated to alanine residues (Rb(ΔCDK)) (Fig. 5i). As expected, the expression of Rb(ΔCDK), which cannot be phosphorylated, suppressed E2F activation and S phase entry more potently than Rb(WT) (Fig. 5j and Extended Data Fig. 9c).

To test whether T373, S608 and S612 (T373/S608/S612) phosphorylation is required for partial Rb inactivation and intermediate E2F activation, the central three residues in Rb were mutated to alanines (T373A/S608A/S612A; Rb(WT-3A)). Rb(WT-3A) suppressed E2F activation and S phase entry more than Rb(WT) and similarly to the Rb(ΔCDK) mutant (Fig. 5j and Extended Data Fig. 9c). We next tested the sufficiency of T373/S608/S612 phosphorylation by making an add-back Rb(ΔCDK) mutant, in which T373, S608 and S612 out of the 15 alanines were mutated to aspartic acid to mimic phosphorylation (T373D/S608D/S612D; Rb(ΔCDK-3D)). This phosphomimetic mutant lost its inhibitory capacity compared with the Rb(ΔCDK) mutant (Fig. 5j and Extended Data Fig. 9c). These results suggest that phosphorylation of the T373/S608/S612 sites in Rb is necessary and sufficient for intermediate E2F activation. Together, the different lines of evidence indicate that reversible phosphorylation of T373, S608 and probably S612 in the central region in Rb are rate-limiting steps for E2F activation, controlling how long cells stall in a primed state and whether cells ultimately commit to proliferate or reverse to quiescence.

## Both CDK2 and CDK4/6 target T373

We noted that most cultured cells have higher CDK4/6 activity and a shortened G1 phase compared with cells in vivo. We therefore used in most experiments CDK4/6 inhibition to mimic more physiological conditions. When we instead used bulk-cell western blot analysis of cells without CDK4/6 inhibition, we still detected T373 phosphorylation in the lower band, albeit at a reduced level (Fig. 4e, DMSO lane). Furthermore, multiplexed single-cell immunofluorescence analysis showed that Rb is also phosphorylated at T373 and S608 before S807/S811, S780 and T826 in cells without CDK4/6 inhibition (Fig. 5k and Extended Data Fig. 10a). The same order of phosphorylation was also observed when we inhibited CDK2 using PF-07104091 (Extended Data Fig. 10b,c). Thus, independent of whether CDK4/6 or CDK2 is active in G1, cells preferentially phosphorylate Rb at T373 and S608 over the C-terminal phosphorylation sites.

## Conclusions

Our study identified a reversible primed G1 state of partial Rb inactivation and intermediate E2F activation (Fig. 5l). Rb phosphorylation at T373 but not at S807/S811 can serve as a marker for this primed G1 state during which cells remain sensitive to fluctuating external and internal signals. Notably, the T373 site in Rb is conserved across vertebrates and some invertebrates (Extended Data Fig. 10d). Like the other Rb sites, T373 can be phosphorylated by CDK2 or CDK4/6. Our data suggest that the preferential phosphorylation of T373 and S608 in Rb is mostly mediated by the preferential dephosphorylation of the C-terminal Rb sites over that of these central sites. Having such a slowly turned over T373 site enables the averaging of CDK2 and CDK4/6 activities that frequently fluctuate in G1 phase[7,39] as a result of fluctuating externally and internally regulated upstream signals[5].

Moreover, our results indicated that the intermediate E2F activation in the primed G1 state is a consequence of Rb T373 and S608 phosphorylation mediating the release of E2F(TD) from RbP, one of the two interactions between E2F and Rb. The disruption of the E2F(TD)–RbP interaction probably activates E2F transcription by freeing up the E2F(TD), which recruits general transcription factors[40–42], and by liberating RbP, which releases the histone deacetylases that suppress transcription[43–45]. In this way, E2F can be activated to an intermediate level by T373 and S608 phosphorylation while Rb stays on chromatin. Our study further showed that full E2F activation is a consequence of Rb hyperphosphorylation disrupting the second inhibitory interaction, between RbC and E2F(MB)–DP(MB), which releases Rb from chromatin.

Previous work has shown that Rb can be monophosphorylated[20,21], which differs in the timing and context of the Rb T373 phosphorylation discussed here. In early G1, CDK4/6 initially monophosphorylates Rb stochastically and at low levels at 14 different sites without E2F activation[20]. The primed state we identified comprises a different state because a larger fraction of Rb is phosphorylated at T373. We further demonstrated that T373 phosphorylation is linked to intermediate E2F activation before the G1/S transition, and previous work has shown that the low level monophosphorylated Rb isoforms may have roles beyond cell cycle regulation[21].

In conclusion, we propose that cells integrate fluctuating external and internal signals[46] during a primed state of intermediate E2F activation. This primed G1 state may function as a safety period, allowing cells to reliably decide whether to prevent excessive proliferation by returning to quiescence or, if needed, initiate proliferation by engaging the positive feedback mechanism between E2F and CDK2.

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

## Methods

### Cell culture

All experiments were performed using MCF-10A human mammary epithelial cells (American Type Culture Collection (ATCC), CRL-10317, RRID: CVCL_0598) unless otherwise noted. MCF-10A cells were cultured in DMEM/F12 growth medium with HEPES (Gibco, 11039047), supplemented with 5% horse serum (Gibco, 16050122), 20 ng ml$^{-1}$ EGF (PeproTech, AF-100-15), 0.5 µg ml$^{-1}$ hydrocortisone (Sigma, H0888), 100 ng ml$^{-1}$ cholera toxin (Sigma, C8052) and 10 µg ml$^{-1}$ insulin (Sigma, I1882). Cells were passaged using trypsin-EDTA (0.05%, Gibco, 25300054), and trypsin was neutralized in DMEM/F12 supplemented with 20% horse serum. RPE-1 human retinal pigment epithelial cells (ATCC, CRL-4000, RRID: CVCL_4388) were cultured in DMEM/F12 with HEPES supplemented with 10% FBS (Sigma, TMS-013-B) and 0.01 mg ml$^{-1}$ hygromycin B (Invivogen, ant-hg-1). BJ-5ta human foreskin fibroblast cells (ATCC, CRL-4001, RRID: CVCL_6573) were cultured in DMEM growth medium (Gibco, 11995065) supplemented with 20% Medium 199 (Thermo Fisher, 11150059), 10% FBS and 0.01 mg ml$^{-1}$ hygromycin B. U2OS human osteosarcoma epithelial cells (ATCC, HTB-96, RRID: CVCL_0042) and Lenti-X 293T human embryonic kidney cells (Takara Bio, 632180, RRID: CVCL_4401) were cultured in DMEM growth medium with 10% FBS. For MCF-10A serum starvation, cells were cultured in starvation medium (growth medium without horse serum, EGF and insulin, but instead supplemented with 0.3% BSA) after two washes of starvation medium. For RPE-1 and BJ-5ta serum starvation, cells were cultured in starvation medium (growth medium without FBS but instead supplemented with 0.3% BSA) after two washes of starvation medium. For mitogen release, starvation medium was exchanged with starvation medium supplemented with EGF or growth medium. Cells were cultured at 37 °C and 5% $CO_2$. For microscopy experiments, 96-well glass-bottomed plates (Cellvis, P96-1.5H-N) were coated with collagen (Advanced Biomatrix, 5005-B, 30–60 µg ml$^{-1}$ for at least 1 h) for all cell lines (except RPE-1) or with bovine plasma fibronectin (Sigma-Aldrich, F1141, 10-20 µg ml$^{-1}$ for at least 1 h) for RPE-1 and cells were seeded into wells at least the night before performing experiments.

### Cell line generation

All constructs were introduced into cells using third-generation lentiviral transduction[47,48]. In brief, lentivirus was produced in HEK-293T cells co-transfected with packaging plasmids pMDLg/pRRE (Addgene, 12251), pRSV-rev (Addgene, 12253) and pCMV-VSV-G (Addgene, 8454) together with the lentiviral plasmid with Lipofectamine 2000 (Thermo, 11668019). At 72 h after transfection, virus was collected from the supernatant, filtered through a 0.22 µm filter (Millipore, SCGP00525) and concentrated using 100 kDa centrifugal filters (Millipore, UFC910024). Virus was then transduced into cells in growth medium. For constitutively expressed fluorescent constructs, positive fluorescent cells were sorted using a BD Influx cell sorter (performed at the Stanford Shared FACS Facility) or a BD Aria II cell sorter (performed at the Weill Cornell Medicine Shared FACS Facility). mVenus-positive MCF-10A cells were single-cell cloned to establish cell lines expressing the E2F reporter. Doxycycline-inducible constructs (TetOn in pCW backbone with puromycin selection marker) were selected with 1 µg ml$^{-1}$ puromycin until control cells died. TetOn cells were grown in the absence of doxycycline until the time of experiment. Doxycycline was added 5 h before release for induction. MCF-10A, RPE-1, BJ-5ta, U2OS cells were acquired directly from ATCC. Lenti-X 293T cells were acquired directly from Takara Bio. MCF-10A cells were validated by RNA-seq. All cell lines tested negative for mycoplasma contamination.

### Plasmid generation

Plasmids generated in this study were assembled using Gibson assembly of PCR-amplified inserts and restriction-enzyme-digested plasmid backbones. Genomic DNA was extracted using a DNeasy Blood & Tissue kit (Qiagen, 69504) according to the manufacturer's instructions. The nuclear marker H2B-miRFP670 was amplified from pLenti-PGK-CMV-H2B-miRFP670 (a gift from T. Kudo, M. Covert Laboratory, Stanford University) and inserted into the pLV backbone to generate pLV-EF1a-H2B-miRFP670. The CDK2 activity reporter was amplified from pCSII-hDHB(amino acids 994–1087)–mVenus[13] and inserted into pLV backbone to generate pLV-EF1a-DHB–mTurquoise. The CRL4$^{Cdt2}$ reporter was amplified from tFucci(CA)2/pCSII-EF[9] and inserted into the pLV backbone to generate pLV-mCherry–hCDT1(1–100)ΔCy. 12S-E1A was amplified from pBabe 12S E1A (Addgene, plasmid 18742) and inserted into the pCW backbone (derived from pCW-Cas9, Addgene, plasmid 50661) to generate pCW-12S-E1A-HA-puro. E2F1 was amplified from HA-E2F-1 wt-pRcCMV (Addgene, plasmid 21667) and inserted into the pCW backbone to generate pCW-HA-E2F1-puro. Cyclin E1, cyclin D1 and Rb were amplified out of cDNA from MCF-10A cells and inserted into the pCW backbone to generate pCW-CCNE1-HA-puro, pCW-CCND1-HA-puro and pCW-HA-hRB-WT-puro, respectively. Rb(ΔCDK) was amplified from pCMV HA hRb ΔCDK (Addgene, plasmid 58906) and inserted into the pCW backbone to generate pCW-HA-hRb-ΔCDK-puro. Rb T373A, S608A and S612A mutations were introduced by PCR on pCW-HA-hRb-WT-puro to generate pCW-HA-hRb-WT-T373A-puro, pCW-HA-hRb-WT-T373A/S608A-puro and pCW-HA-hRb-WT-T373A/S608A/S612A-puro, respectively. Rb T373 and S608 reverse mutations were introduced by PCR on pCW-HA-hRb-ΔCDK-puro to generate pCW-HA-hRb-ΔCDK-T373-puro and pCW-HA-hRb-ΔCDK-T373/S608-puro, respectively. Rb T373D/S608D/S612D mutations were introduced by PCR on pCW-HA-hRb-ΔCDK-puro to generate pCW-HA-hRb-ΔCDK-T373D/S608D/S612D-puro.

### E2F reporter development

Extended Data Fig. 1a explains the workflow for development of the E2F reporter. To optimize the promoter for the E2F transcriptional activity reporter, we selected 11 candidate genes that were upregulated by growth medium (containing serum, EGF and insulin) and downregulated by the CDK4/6 inhibitor (thus specific to the CDK–Rb–E2F pathway) in the RNA sequencing experiments (Extended Data Fig. 1b). We further validated the selected 11 genes by RT–qPCR for their responsiveness to growth medium and the CDK4/6 inhibitor (Extended Data Fig. 1c,d). We next selected promoter regions from each gene based on a genome database, Ensembl (https://www.ensembl.org), and inserted the promoter regions into the prototype reporter construct such that the promoter drives mVenus expression. We then evaluated the sensitivity and specificity of the reporter by live-cell imaging (Extended Data Fig. 1e). The fold change in DMSO indicated the sensitivity to growth medium, whereas the fold change in the CDK4/6 inhibitor indicated the specificity to the CDK–Rb–E2F pathway. Among the top three constructs (*FAM111B*, *DTL* and *CDC6*) for these criteria, we decided to use the *CDC6* promoter because it is free of cytotoxicity and has no background signals in quiescence.

Candidate constructs for an E2F reporter consisted of a E2F target gene promoter, mVenus, NLS (SV40), PEST (mouse ornithine decarboxylase) and a polyA signal (bGH). The PEST sequence was added to achieve a rapid turnover rate of the reporter by targeting the mVenus protein for degradation. The maturation half-time of mVenus is reported as 4.1 min[49,50]. The degradation half-life of eGFP with the PEST sequence used in the reporter is reported as 1–2 h[51]. E2F target gene promoter regions were either obtained as synthesized oligonucleotides (gBlocks) or amplified out of genomic DNA from MCF-10A cells. The following human E2F target gene promoter regions were used for the reporters (relative position from the transcription start site and the accession numbers in NCBI): *CDC6* (−1057 to 239, NM_001254.4); *FAM111B* (−545 to 1765, NM_198947.4); *DTL* (−1948 to 143, NM_016448.4); *MCM10* (−435 to 49, NM_018518.5); *RRM2* (−3317 to −8, NM_001034.4); *E2F1* (short) (−123 to 59, NM_005225.3) (the same promoter sequence as the previous E2F reporter[10]); *CDC45* (−457 to 49, NM_003504.5); *EXO1*

(−1004 to 1771, NM_130398.4); *E2F1* (long) (−720 to 1681, NM_005225.3); *CLSPN* (−446 to 263, NM_022111.4); *CDT1* (−1208 to 532, NM_030928.4); and *TCF19* (−1589 to 6, NM_007109.3).

As predicted, the mVenus signal increased as cells transitioned into S phase and then decreased[52,53] (Fig. 1c,d and Extended Data Fig. 1f), and the mVenus signal amplitude correlated in single cells with the mRNA abundance of known E2F targets (Extended Data Fig. 2f,g; non-linearity in the mRNA puncta area can be explained by the overlap of mRNA puncta, Extended Data Fig. 2h–k). Moreover, the mVenus signal was upregulated by knockdown of the E2F suppressor Rb (Extended Data Fig. 1g), overexpression of adenovirus E1A (which inactivates Rb) (Extended Data Fig. 1h) and induced expression of E2F1 (Extended Data Fig. 1i,j) or cyclin D1 (cyclinD1–CDK4/6 phosphorylates Rb) (Fig. 1e and Extended Data Fig. 1k). All these results provided validation that the reporter measures E2F transcriptional activity. The E2F reporter monitors the activity of both activating and repressing E2Fs and therefore can be used as a global E2F transcriptional activity reporter (Extended Data Fig. 2a–e). We note that a previous E2F reporter, which is based on the E2F1 promoter, is regulated by both E2F and MYC[17,54], but the E2F reporter developed here does not respond to MYC induction (Extended Data Fig. 1l).

The E2F reporter and other reporter constructs used in the study are available through the non-profit organization Addgene (plasmids 212665-212675) (https://www.addgene.org/Tobias_Meyer/).

### siRNA transfection
MCF-10A cells were transfected with siRNA using DharmaFECT 1 (Dharmacon, T-2001-03) according to the manufacturer's protocol using 20 nM siRNA and 1:500 diluted DharmaFECT 1 final concentration. Cells were incubated for 4–6 h in serum starvation medium containing transfection mixture, followed by a medium change. Pools of four siRNA oligonucleotides (ON-TARGETplus, Dharmacon) were used for siControl, si*RB1*, si*E2F1*, si*E2F2*, si*E2F3* and si*E2F7*. For *RB1* knockdown while overexpressing doxycycline-inducible Rb constructs, oligonucleotides that target the *RB1* 3′ untranslated region was selected to avoid knockdown the constructs.

### Chemicals
The following stock solutions of drugs were dissolved in DMSO (Sigma, D2650 or Santa Cruz, sc-358801): doxycycline hyclate (Sigma, D9891); the CDK4/6 inhibitor palbociclib (Selleck Chemicals, S1116); the CDK2 inhibitor PF-07104091 (ChemieTek, CT-PF0710); the CDK1 inhibitor RO-3306 (Cayman Chemical, 15149); the EGFR inhibitor gefitinib (Selleck Chemicals, S1025); and the PP1 and PP2A inhibitor calyculin A (Santa Cruz, sc-24000). Neocarzinostatin (Sigma, N9162) was dissolved in a solution containing 20 mM MES buffer, pH 5.5.

### RT–qPCR
RNA was extracted from cells using QIAshredder (Qiagen, 79656) and RNeasy Mini kits (Qiagen, 74106). cDNA was generated by reverse transcription using RevertAid reverse transcriptase (Thermo, EP0442) and Oligo(dT)18 primer (Thermo, SO132) or Random Hexamer primer (Thermo, SO142) according to the manufacturer's instructions. qPCR was performed using iTaq Universal SYBR Green supermix (Bio-Rad, 1725122) on a LightCycler 480 II (Roche) according to the manufacturer's instructions.

### RNA fluorescent in situ hybridization
Cells were fixed in 4% paraformaldehyde in PBS for 10 min at room temperature followed by a PBS wash. Cells were permeabilized in 0.2% Triton X-100 in PBS for 15 min followed by a PBS wash. RNA fluorescent in situ hybridization (FISH) was carried out using a ViewRNA ISH cell assay (Thermo, QVC0001) according to the manufacturer's instructions. Cells were washed in PBS and incubated with 1 μg ml⁻¹ Hoechst 33342 (Invitrogen, H3570) in PBS for 10 min, followed by a final PBS wash

before imaging. The following hybridization probes were used: E2F1 (Thermo, VA1-12108-VC); E2F2 (Thermo, VA4-16111-VC); E2F3 (Thermo, VA6-16112-VC); E2F7 (Thermo, VA6-3183369-VC); CCNE2 (Thermo, VA1-3005686-VC); and CDC6 (Thermo, VA4-3084153-VC).

### Western blotting
Cells were grown in 60 mm dishes. At the time of lysis, cells were washed in PBS, lysed in 2× Laemmli sample buffer with 200 mM DTT, passed through a 25 G needle 10 times and heated at 90 °C for 4 min. Samples were then separated by SDS–PAGE using 7.5% Mini-Protean TGX gels (Bio-Rad, 4561026) or 4–20% Mini-Protean TGX gels (Bio-Rad, 4561096) in Tris–glycine–SDS running buffer, followed by wet transfer onto Immun-Blot Low Fluorescence PVDF/filter paper (Bio-Rad, 1620261) in Tris–glycine buffer + 20% methanol. Membranes were washed in TBST (25 mM Tris, pH 7.5, 150 mM NaCl and 0.05% Tween 20), blocked for 1 h in 5% milk + 0.01% NaN₃ in TBST and incubated overnight with primary antibodies in 5% BSA + 0.01% NaN₃ in TBST. For fluorescence detection, membranes were incubated with the secondary antibodies goat anti-rabbit IgG (H+L) secondary antibody Alexa Fluor 680 (1:20,000, Thermo, A-21109) and goat anti-mouse IgG polyclonal antibody IRDye 800CW (1:20,000, Li-Cor, 925-32210). For chemiluminescence detection, membranes were incubated with a secondary HRP-linked goat anti-rabbit IgG secondary antibody (1:2,000, Cell Signaling Technology, 7074) and developed with SuperSignal West Pico PLUS chemiluminescent substrate (Thermo, 34580). Membranes were imaged using an Odyssey Infrared imaging system (Li-Cor). The following primary antibodies were used: mouse anti-Rb antibody (1:1,000; Cell Signaling Technology, 9309); rabbit anti-phospho-Rb (T252) antibody (1:1,000; Abcam, ab184797); rabbit anti-phospho-Rb (T356) antibody (1:1,000; Abcam, ab76298); rabbit anti-phospho-Rb (T373) antibody (1:1,000; Abcam, ab52975); rabbit anti-phospho-Rb (S608) antibody (1:1,000; Cell Signaling Technology, 2181); rabbit anti-phospho-Rb (S780) antibody (1:250; Thermo, 701272); rabbit anti-phospho-Rb (S788) antibody (1:250; Abcam, ab277775); rabbit anti-phospho-Rb (S795) antibody (1:1,000; Cell Signaling Technology, 9301); rabbit anti-phospho-Rb (S807/S811) antibody (1:20,000; Cell Signaling Technology, 8516); rabbit anti-phospho-Rb (T826) antibody (1:1,000, Abcam, ab133446); rabbit anti-EGFR antibody (1:1,000, Cell Signaling Technology, 4267); and rabbit anti-phospho-EGFR (Y1045) antibody (1:1,000; Cell Signaling Technology, 2237).

### Immunofluorescence
**General protocol.** Cells were fixed in 4% paraformaldehyde in PBS for 10 min at room temperature followed by a PBS wash. If cells expressed fluorescent proteins that spectrally overlapped with the fluorophores used in later steps, the fluorescent proteins were chemically bleached[55] in 3% H₂O₂ + 20 mM HCl in PBS for 1 h and washed in PBS. Cells were permeabilized and blocked in blocking buffer (10% FBS, 1% BSA, 0.1% Triton X-100 and 0.01% NaN₃ in PBS) for 1 h. Cells were then incubated with primary antibodies overnight in blocking buffer at 4 °C, washed in PBS and incubated with secondary antibodies in blocking buffer for 1 h at room temperature. Cells were washed in PBS and incubated with 1 μg ml⁻¹ Hoechst 33342 (Invitrogen, H3570) in PBS for 10 min, followed by a final PBS wash before imaging.

**Iterative immunofluorescence.** If simultaneously staining for targets with antibodies of the same species, the iterative indirect immunofluorescence imaging (4i) method[31] was used to sequentially image multiple antibodies. In brief, the first round of imaging was identical to the general immunofluorescence protocol, with the exception that cells after the post-Hoechst PBS wash were washed in ddH₂O and then placed in imaging buffer (700 mM *N*-acetyl cysteine in ddH₂O, pH 7.4, Sigma, A7250). Cells were imaged and then washed in ddH₂O. The previous-round antibodies were eluted by 3 × 10-min incubations in elution buffer, which consisted of 0.5 M glycine (Sigma), 3 M urea

(Sigma), 3 M guanidinium chloride (Sigma) and 70 mM TCEP-HCl (Goldbio, TCEP50) in ddH$_2$O, pH 2.5, followed by a PBS wash. Cells were then checked under a fluorescence microscope to ensure proper elution. Cells were then blocked with blocking buffer, consisting of 1% BSA and 150 mM maleimide (dissolved immediately before use, Sigma, 129585) in PBS for 1 h and then washed in PBS, followed by primary antibody incubation, and the subsequent steps were performed the same as in the first round and repeated as needed.

**Pre-extraction for chromatin-bound protein.** If chromatin-bound proteins were being stained, soluble proteins were extracted from cells. Immediately before fixation, medium was aspirated off from cells and the plate was placed on ice. Cells were incubated for 5 min in ice-cold pre-extraction buffer, consisting of 0.2% Triton X-100 (Sigma, X100) and 1× Halt Protease Inhibitor cocktail (Thermo, 78429) in PBS. After pre-extraction, 8% paraformaldehyde in H$_2$O was directly added to wells 1:1 with wide-orifice tips to minimize cell detachment, and cells were fixed for 1 h at room temperature, after which the sample was treated using the general staining protocol.

**EdU incorporation and labelling.** If measuring EdU incorporation, cells were pulsed with 10 µM EdU (Cayman Chemical, 20518) in growth medium for 15 min before fixation. EdU is incorporated throughout the EdU pulse, such that incorporated EdU reflects the average rate of DNA synthesis over the length of the pulse. Thus, 15 min of a short EdU pulse is more reflective of the instantaneous DNA synthesis rate compared with a longer pulse such as 1 h. After blocking cells (before primary antibodies), cells were washed once with PBS and then a click reaction was performed in 2 mM CuSO$_4$, 20 mg ml$^{-1}$ sodium ascorbate in TBS (Tris 50 mM and NaCl 150 mM pH 8.3) with 3 µM AFDye 568 picolyl azide (Click Chemistry Tools, 1292) for 30 min, followed by a PBS wash.

**Antibodies.** The following primary antibodies were used: mouse anti-Rb antibody (1:250; BD, 554136); rabbit anti-phospho-Rb (T373) antibody (1:100 to 1:1,000; Abcam, ab52975); rabbit anti-phospho-Rb (S608) antibody (1:100 to 1:250; Cell Signaling Technology, 2181); rabbit anti-phospho-Rb (S780) antibody (1:100 to 1:250, Thermo, 701272); rabbit anti-phospho-Rb (S807/S811) antibody (1:2,500; Cell Signaling Technology, 8516); rabbit anti-phospho-Rb (T826) antibody (1:100; Abcam, ab133446); rabbit anti-p21 antibody (1:2,500; Cell Signaling Technology, 2947); mouse anti-cyclin E antibody (1:400; Santa Cruz, sc-247); rabbit anti-c-Myc antibody (1:800; Cell Signaling Technology, 5605); rabbit anti-53BP1 antibody (1:500; Cell Signaling Technology, 4937); rabbit anti-HA tag antibody (1:1,000; Cell Signaling Technology, 3724); and mouse anti-HA tag antibody (1:1,000; BioLegend, 901501). The following secondary antibodies were used: goat anti-rabbit IgG Alexa Fluor 514 (1:2,000; Thermo, A-31558) and goat anti-mouse IgG Alexa Fluor 647 (1:2,000; Thermo, A-21235).

## Microscopy

For automated epifluorescence microscopy, cells were imaged using a Ti2-E inverted microscope (Nikon) or ImageXpress Micro XLS microscope (Molecular Devices). For imaging on the Ti2-E, multichannel fluorescent images were taken using a 89903-ET-BV421/BV480/AF488/AF568/AF647 Quinta Band set (Chroma Technology) with an LED light source (Lumencor Spectra X) and an ORCA-Flash4.0 V3 sCMOS camera (Hamamatsu). A ×20 (Nikon CFI Plan Apo Lambda, 0.75 NA) objective was used to acquire images. For imaging on the ImageXpress, images were taken with appropriate single-band filter sets with a white-light source, using a ×20 (Nikon CFI Plan Apo Lambda, 0.75 NA) and Zyla 4.2 sCMOS camera (Andor). All images were acquired in 16-bit mode with 2 × 2 or 4 × 4 binning, and acquisition settings were chosen to not saturate the signal.

For live-cell time-lapse imaging, 96-well plates were imaged within an enclosed 37 °C, 5% CO$_2$ environmental chamber in 200 µl of medium.

Around 4–9 sites were imaged in each well (with the number of wells imaged varying depending on the experiment) every 12 min. Light exposure to cells was limited by using the minimum exposure necessary to maintain an acceptable signal-to-noise ratio on a per-channel basis, and total light exposure was always limited to below 200 ms per site for each time point. When performing the live-cell imaging followed by fixed-cell imaging, cells were immediately taken off the microscope following the final time point and fixed. When matching fixed-cell imaging back to either live-cell imaging or previous rounds of fixed-cell imaging, the plate position (which can shift slightly when replacing the plate on the microscope) was aligned to approximately the same location and further aligned computationally during image analysis.

## Image analysis

Image analysis was performed using a custom Matlab pipeline as previously described[8,14]. In brief, images were first flatfield corrected (illumination bias determined by pooling background areas from multiple wells from the same imaging session), and then the background was subtracted locally. Cells were segmented for their nuclei based on either Hoechst staining or H2B signal. For the E2F and CRL4$^{Cdt2}$ reporters, the mean signal within the nucleus was then calculated. CDK2 activity was calculated by taking the ratio between the median cytoplasmic intensity and the mean nuclear intensity. The cytoplasm was sampled by expanding a ring outside the nucleus (with inner radius of 0.65 µm and outer radius of 3.25 µm) without overlapping with the cytoplasm from a neighbouring cell. For RNA FISH measurements, cells were segmented for their whole-cell regions by spatially approximating an area encompassing the nucleus and reaching as far as 15.6 µm outside the nuclear mask without overlapping other cell regions. A foreground mask of FISH puncta was generated by top hat-filtering the raw image with a circular kernel radius of 1.3 µm and thresholding on absolute intensity. RNA puncta count was calculated as the number of foreground pixels within a given whole-cell region. The image processing pipeline and code used to generate all figures in this study have been deposited into GitHub (https://github.com/MeyerLab/image-analysis-konagaya-2022).

The categorization of MCF-10A cells into S enter, E2F reverse and undecided was based on the following criteria: S enter cells are cells that entered S phase, which is detected by the CRL4$^{Cdt2}$ reporter signal; E2F reverse cells are cells with E2F activation, without S phase entry, and with an E2F activity decrease of more than half from the peak; undecided cells are cells with E2F activation, but without S phase entry or a decrease in E2F activity.

## Protein structural modelling

Structures were modelled using ColabFold[56], a simplified AlphaFold2 algorithm[30,57], without templates (https://colab.research.google.com/github/sokrypton/ColabFold/blob/main/beta/AlphaFAlp2_advanced.ipynb). Multiple sequence alignments (MSAs) were generated using MMseqs2 and unpaired (generates separate MSAs for each protein). The Rb–E2F1–DP1 complex was modelled as a heterotrimer with a 1:1:1 stoichiometric ratio. To reduce memory requirements, only a subset of the MSA was used as input to the model by subsampling the MSA to a maximum of 512 cluster centres and 1,024 extra sequences. Relaxation of the predicted structures using amber force fields was disabled because it barely moves the main-chain structure. Turbo setting (compiles once, swaps parameters and adjusts the maximum MSA) was used to speed-up and reduce memory requirements. The number of random seeds = 1, the number of ensembles = 1 and the threshold for tolerance = 0. Training setting (which activates the stochastic part of the model) was disabled. The predicted template modelling score was used to rank and assess the confidence of the predicted protein–protein interaction. Five models were computed through three recycles, and the one with the highest predicted template modelling score was visualized using PyMOL. The Rb–E2F1–DP1 complex predicted in the study is available through ModelArchive (https://modelarchive.org/doi/10.5452/ma-jcq2m).

## PSP analysis

PSP plots were used to show single-cell correlations of Rb phosphorylation between two different sites. Each phosphorylation signal was normalized by the total Rb antibody signal in the same cell, and each axis was adjusted to the average phosphorylation signal in S phase of 1 (when Rb is hyperphosphorylated). The colour bar indicates the relative cell population density. A red line shows fitting with a preferential relative phosphorylation or dephosphorylation rate between the two sites ($PSP_{coeff}$).

We constructed a kinetic phosphorylation–dephosphorylation model to ask whether the observed differences in dephosphorylation kinetics between T373 and S807/S811 (Fig. 5a) could explain the preferential phosphorylation at T373 over S807/S811 (Fig. 3c). Phosphorylated fractions at one site ($x$; such as T373) and another reference site ($y$; such as S807/S811) are described as follows:

$$\frac{dx}{dt} = \alpha_1(1-x) - \beta_1 x$$

$$\frac{dy}{dt} = \alpha_2(1-y) - \beta_2 y$$

where $\alpha_1$ and indicate $\alpha_2$ the phosphorylation rate for $x$ and $y$, respectively; and $\beta_1$ and $\beta_2$ indicate the dephosphorylation rate for $x$ and $y$, respectively. At equilibrium,

$$\alpha_1 = \frac{\beta_1 x}{(1-x)}$$

$$\alpha_2 = \frac{\beta_2 y}{(1-y)}$$

We assumed that the Rb-targeting kinases (CDK4/6 and CDK2) do not have selectivity to T373 over S807/S811 because the convex relationship in the PSP plot does not change with treatment with the CDK4/6 inhibitor or the CDK2 inhibitor (DMSO: Fig. 5k, CDK4/6i, Fig. 3c, the top left panel, CDK2i, and Extended Data Fig. 10b). With this assumption,

$$\alpha_1 = \alpha_2$$

Thus,

$$y = \frac{x}{(PSP_{coeff} + (1 - PSP_{coeff})x)}$$

where $PSP_{coeff} = \frac{\beta_2}{\beta_1}$ indicates the relative dephosphorylation rate between two sites. We used the final equation above to fit the measured phosphorylation fractions ($0.1 \le x \le 0.9$ and $0.1 \le y \le 0.9$) in the PSP plots and obtained $PSP_{coeff}$ values.

## ChIP–seq analysis

The ChIP–seq signals and peaks of E2F proteins on the *CDC6* promoter were downloaded from ENCODE[58] (https://www.encodeproject.org/). All the experiments were performed in duplicate and with an irreproducible discovery rate cut-off of 0.05. The following target E2F proteins and cell lines were used in the ChIP–seq analysis (the accession numbers of the fold change over control and irreproducible discovery rate thresholded-peaks in ENCODE): E2F1 in HepG2 (ENCFF846JMO.bigWig; ENCFF919WXY.bigBed); E2F1 in MCF-7 (ENCFF858GLM.bigWig; ENCFF692OYJ.bigBed); E2F2 in HepG2 (ENCFF826PYA.bigWig; ENCFF629CDJ.bigBed); E2F3 in K562 (ENCFF838PBU.bigWig; ENCFF922ILX.bigBed); E2F4 in HepG2 (ENCFF491MUP.bigWig; ENCFF311TOD.bigBed); E2F4 in MCF-7 (ENCFF232RTG.bigWig; ENCFF249IZG.bigBed); E2F5 in HepG2 (ENCFF518RZY.bigWig; ENCFF582JSQ.bigBed); E2F6 in A549 (ENCFF190MLK.bigWig; ENCFF550XVR.bigBed); E2F7 in K562 (ENCFF979YTG.bigWig; ENCFF212JSU.bigBed); E2F8 in HepG2 (ENCFF805BWM.bigWig; ENCFF320WJO.bigBed); and E2F8 in MCF-7 (ENCFF898YRB.bigWig; ENCFF641CMX.bigBed).

## MSA analyses

The amino acid sequences of human Rb orthologues were downloaded from the NCBI database (https://www.ncbi.nlm.nih.gov/). Rb orthologues used in the MSA analysis (accession numbers in NCBI): *Homo sapiens* (NP_000312.2), *Mus musculus* (NP_033055.2), *Gallus gallus* (NP_989750.2), *Xenopus tropicalis* (NP_001269454.1), *Danio rerio* (NP_001071248.1) and *Strongylocentrotus purpuratus* (XP_030838139.1). Full-length protein sequences were aligned using Clustal Omega (https://www.ebi.ac.uk/Tools/msa/clustalo/) with the default parameters as follows: output guide tree = true, output distance matrix = false, dealign input sequences = false, mBed-like clustering guide tree = true, mBed-like clustering iteration = true, number of iterations = 0, maximum guide tree iterations = −1, maximum HMM iterations = −1, output alignment format = clustal_num, output order = aligned, sequence type = protein

## Statistical analysis

Statistical analyses were performed using two-sided, two-sample *t*-test or one-way ANOVA and Scheffé's post hoc comparison. Quantifications are represented as the mean ± s.e. or median ± 25th and 75th percentiles as specified in the figure legends. *P* values are as follows: *$P < 0.05$, **$P < 0.01$ and ***$P < 0.001$. Further statistical details of experiments are reported in the figure legends. No statistical methods were used to predetermine sample size. The experiments were not randomized and investigators were not blinded to allocation during experiments and outcome assessment.

## Reporting summary

Further information on research design is available in the Nature Portfolio Reporting Summary linked to this article.

## Data availability

Uncropped western blot images are available in Supplementary Fig. 1. Example time-lapse imaging sequences of MCF-10A and RPE-1 cells expressing the E2F reporter are available at Mendeley Data (https://data.mendeley.com/preview/hfz5hg267n?a=5d36d373-3bbb-477b-a824-7704ff1df963). The complete raw imaging dataset is not provided owing to size limitations, but additional datasets are available upon reasonable request.

## Code availability

The custom image analysis code used in this study has been deposited into GitHub https://github.com/MeyerLab/image-analysis-konagaya-2022).

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

**Acknowledgements** We thank all members of the Meyer Laboratory and M.T. Laboratory for reagents and contributions to the planning of experiments and interpretation of the data; T. Kudo and members of the M.C. Laboratory for providing pLenti-PGK-CMV-H2B-miRFP670 plasmid; and staff at the Stanford and Weill Cornell Medicine Shared FACS Facilities for cell sorting. This work was supported by a fellowship from Astellas Foundation for Research on Metabolic Disorders (Y.K.), a Japan Society for the Promotion of Science (JSPS) Overseas Research Fellowship (Y.K.), a Medical Scientist Training Program grant from the National Institute of General Medical Sciences of the National Institutes of Health under award number T32GM007739 to the Weill Cornell/Rockefeller/Sloan Kettering Tri-Institutional MD-PhD Program (D.R.), a National Institute of Mental Health (NIMH) F30 grant F30MH132311-02 (D.R.), a NSF Graduate Research Fellowship DGE-1147470 (N.R.), a Stanford Graduate Fellowship (Y.F.), and a fellowship from Stanford Center for Systems Biology (Y.F.). All of the work was supported by National Institute of General Medical Sciences (NIGMS) R35 grant R35GM127026 (T.M.).

**Author contributions** The research plan, development of the E2F reporter, PSP analyses, data analyses approaches and the conceptualization of the results were based on ongoing discussions between Y.K. and T.M. Y.K. and T.M. wrote the manuscript. Y.K. performed all final experiments in the manuscript. D.R. made the initial observation that the activation of E2F can be transient. N.R. developed the software version used for the automated image analysis. N.R. and Y.F. made technical contributions that helped with the optimization of the imaging strategies and statistical analyses.

**Competing interests** The authors declare no competing interests.

**Additional information**
**Correspondence and requests for materials** should be addressed to Yumi Konagaya or Tobias Meyer.

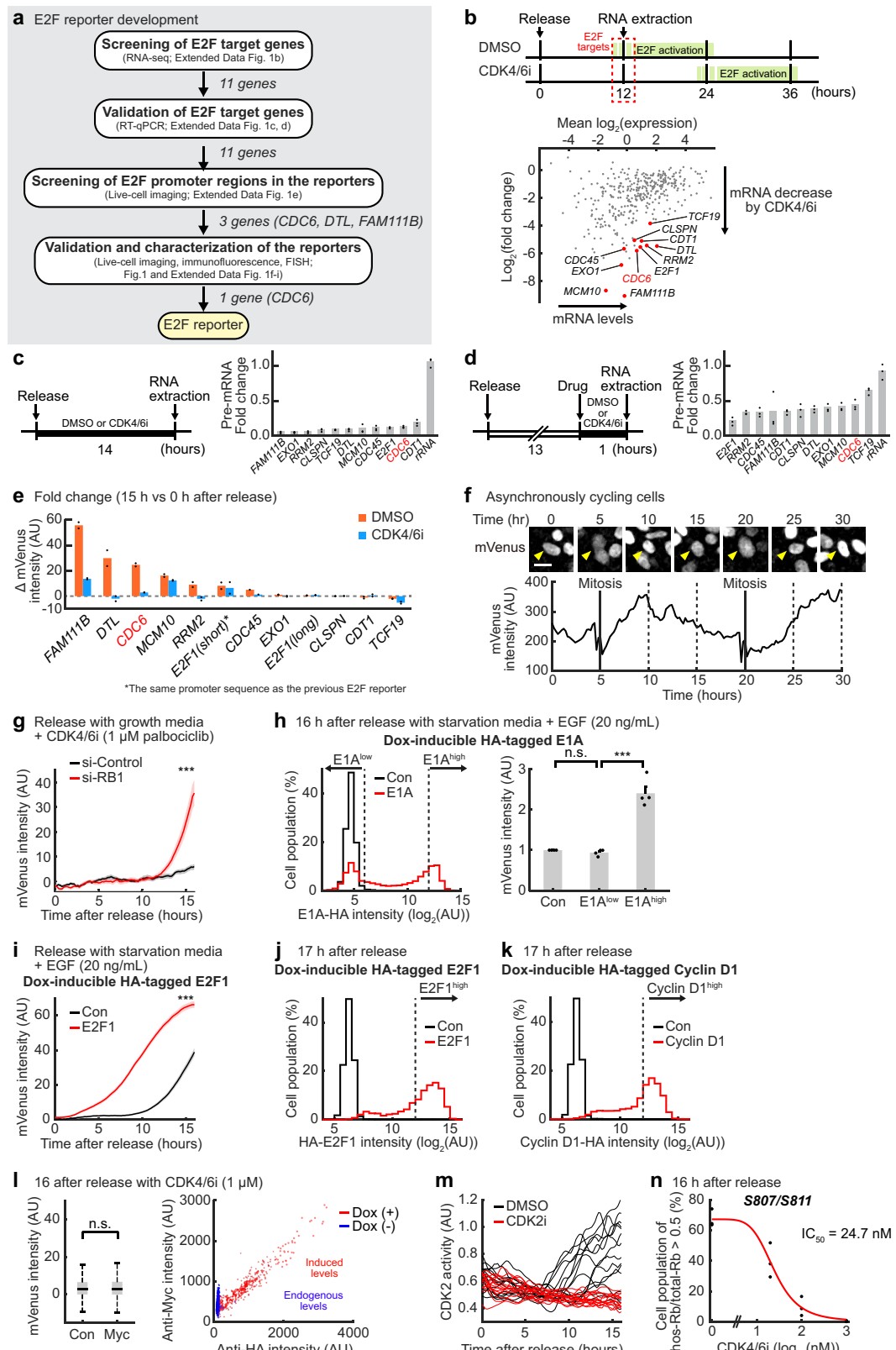

**Extended Data Fig. 1** | See next page for caption.

**Extended Data Fig. 1 | Development of an E2F activity reporter and validation of the E2F and CDK2 live-cell reporters.** Related to Fig. 1. **a**, Schematic of E2F reporter development workflow. **b**, Top, schematic of E2F activation in cells released with DMSO vs CDK4/6i. A red dotted square shows the timing of RNA extraction for RNA-Seq. Bottom, MA plot (log fold change of gene expression in DMSO vs CDK4/6i (1 μM) plotted against mean log gene expression). Cells were assayed for RNA-Seq 12 h after release with growth media + DMSO or CDK4/6i (1 μM). Marked genes are candidate E2F target genes that were used for E2F activity reporter development (1 of n = 3 biological replicates. Reanalysis of published data[59]). **c**, **d**, Fold change in pre-mRNA expression levels (measured by RT-qPCR) of the candidate E2F target genes as selected in **b**. In **c**, cells were released with growth media + DMSO or CDK4/6i (1 μM) for 14 h. In **d**, cells were released with growth media for 13 h and treated with DMSO or CDK4/6i (1 μM) for 1 h before RNA extraction. Relative pre-mRNA expression levels were calculated using EEF1A1 as a housekeeping gene. Expression levels with CDK4/6i were normalized by those with DMSO to calculate the fold change per gene per experiment (mean from n = 3 biological replicates). **e**, Fold change in mean mVenus intensity 15 h after release compared to 0 h after release. Cells expressing mVenus under the control of the E2F target gene promoters were released with growth media + DMSO or CDK4/6i (1 μM) for 15 h (mean from n = 2 biological replicates). **f**, Top, representative time-course images of MCF-10A cells expressing mVenus under the control of CDC6 promoter (the E2F reporter). Cells were asynchronously cycling in growth media. Scale bar = 20 μm. Bottom, single-cell trace of the mVenus intensity in the cell as shown with yellow arrowheads in Top. **g**, mVenus intensity (the E2F reporter) traces after release with growth media + CDK4/6i (1 μM). Cells were treated with si-Control or si-RB1 two days before release. A p-value for mVenus intensity 16 h after release was calculated using two-sided, two-sample t tests (mean ± SE. n = 100 cells for each condition. p-value = $1.7 \times 10^{-8}$. 1 of n = 3 biological replicates). **h**, Left, histograms of E1A-HA intensity in control or Dox-inducible HA-tagged E1A-expressing cells. Dox was added 5 h before release to induce E1A. Cells were fixed 16 h after release with starvation media + EGF (20 ng/mL) to stain HA. Thresholds for E1A^{low} and E1A^{high} are HA < $2^6$ and HA > $2^{12}$, respectively. Right, mVenus intensity (the E2F reporter) in control, E1A^{low} and E1A^{high} cells 16 h after release. p-values were calculated using two-sided, two-sample t tests (mean ± SE. n = 4 biological replicates. n = 19649, 19983, 6668 cells in total for Con, E1A^{low}, and E1A^{high}, respectively. p-value (Con vs E1A^{low}) = 0.16 (n.s.), p-value (E1A^{low} vs E1A^{high}) =

$1.9 \times 10^{-4}$). **i**, mVenus intensity (the E2F reporter) traces in control or Dox-inducible HA-tagged E2F1-expressing cells. Dox was added 5 h before release to induce E2F1. Cells were released with starvation media + EGF (20 ng/mL). Similar as **h**, HA > $2^{12}$ was used to gate E2F1 overexpressed cells. A p-value for mVenus intensity 16 h after release was calculated using two-sided, two-sample t tests (mean ± SE. n = 500, 484 cells for Con, E2F1, respectively. p-value = $1.3 \times 10^{-23}$. 1 of n = 3 biological replicates). **j**, Histograms of HA-E2F1 intensity in control or Dox-inducible HA-tagged E2F1-expressing cells. Dox was added 5 h before release to induce E2F1. Cells were fixed 17 h after release with starvation media + EGF (20 ng/mL) to stain HA. A threshold for E2F^{high} (used in **i**) is HA > $2^{12}$ (n = 8075, 3398 cells for Con and E2F^{high}, respectively. 1 of n = 3 biological replicates). **k**, Histograms of cyclin D1-HA intensity in control or Dox-inducible HA-tagged cyclin D1-expressing cells. Dox was added 5 h before release to induce cyclin D1. Cells were fixed 17 h after release with starvation media + EGF (20 ng/mL) to stain HA. A threshold for cyclin D1^{high} (used in Fig. 1e) is HA > $2^{12}$ (n = 8075, 5621 cells for Con and Cyclin D1^{high}, respectively. 1 of n = 3 biological replicates). **l**, Left, box plots of mVenus intensity (the E2F reporter) in control or Dox-inducible HA-tagged Myc-expressing cells. Dox was added 5 h before release to induce Myc. Cells were released with starvation media + EGF (20 ng/mL) + CDK4/6i (1 μM). HA > $2^{10}$ was used to gate Myc overexpressed cells. Box centers are median values, box edges are the 25th and 75th percentiles, and whiskers are minimum and maximum values. A p-value for mVenus intensity 16 h after release was calculated using two-sided, two-sample t tests (n = 500 cells each. p-value = 0.23 (n.s.). 1 of n = 3 biological replicates). Right, single-cell correlation of anti-HA intensity and anti-Myc intensity (measured by immunofluorescence). Cells were released with starvation media + EGF (20 ng/mL) + CDK4/6i (1 μM) for 16 h. Dox treatment induces HA-tagged Myc overexpression (n = 500 cells each. 1 of n = 2 biological replicates). **m**, Single-cell traces of CDK2 activity after release with growth media for 16 h. Cells were released with DMSO or CDK2i (1 μM) (n = 15 cells each. 1 of n = 4 biological replicates). **n**, Dose-response curve of the suppression of Rb phosphorylation at S807/S811 by CDK4/6i. Cells were fixed 16 h after release with starvation media + EGF (20 ng/mL) + DMSO or CDK4/6i (20 nM, 100 nM, 1 μM). Percentage of cells with Rb phosphorylation levels > 0.5 are plotted as a function of CDK4/6i concentration. A sigmoidal fit was used to derive the half maximal inhibitory concentration, IC_{50}. (n = 3 biological replicates. n = 9001, 9578, 9932, 8641 cells in total for 0, 20, 100, 1000 nM, respectively).

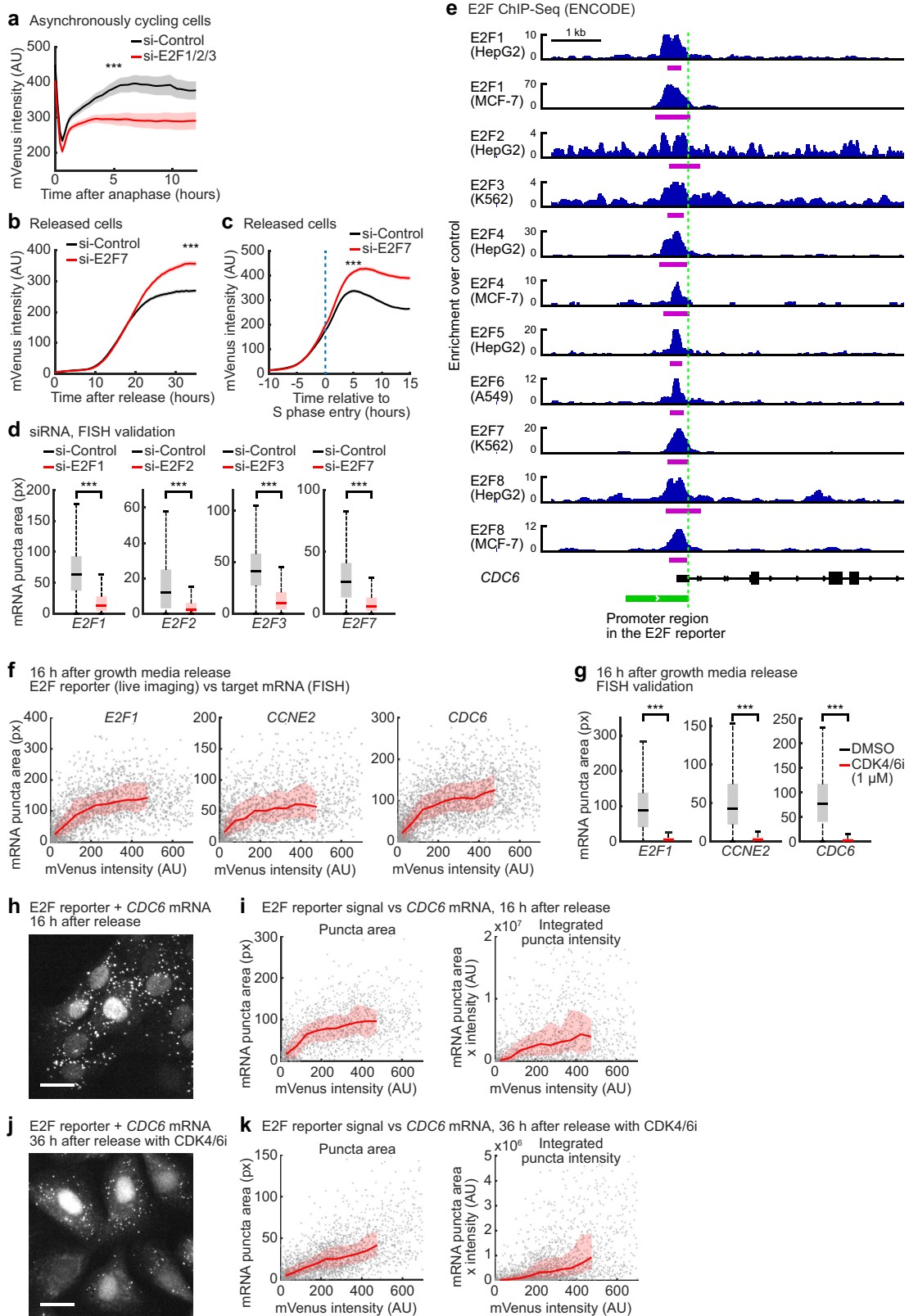

**Extended Data Fig. 2 |** See next page for caption.

**Extended Data Fig. 2 | The E2F reporter is a global E2F transcriptional activity reporter, monitoring both activating and repressing E2Fs.** Related to Fig. 1. **a**, mVenus intensity (the E2F reporter) traces of MCF−10A cells asynchronously cycling in starvation media + EGF (20 ng/mL). Cells were treated with si-Control or si-E2F1/2/3 four hours before starting live-cell imaging. Cells that were born 22 h to 30 h after the start of live-cell imaging were analyzed. Cell traces were computationally aligned at anaphase. A p-value for mVenus intensity 5 h after anaphase was calculated using two-sided, two-sample t tests (mean ± SE. n = 125, 120 cells for si-Control, si-E2F1/2/3, respectively. p-value = 4.3 × 10⁻⁴. 1 of n = 3 biological replicates). **b**, mVenus intensity (the E2F reporter) traces after release with starvation media + EGF (20 ng/mL). Cells were treated with si-Control or si-E2F7 two days before release. A p-value for mVenus intensity 35 h after release was calculated using two-sided, two-sample t tests (mean ± SE. n = 913, 1026 cells for si-Control, si-E2F7, respectively. p-value = 9.0 × 10⁻²⁰. 1 of n = 3 biological replicates). **c**, Cell traces in **b** were computationally aligned at S phase entry. A p-value for mVenus intensity 5 h after S phase entry was calculated using two-sided, two-sample t tests (mean ± SE. n = 796, 872 cells for si-Control, si-E2F7, respectively. p-value = 1.1 × 10⁻¹⁴. 1 of n = 3 biological replicates). **d**, Box plots of mRNA puncta area of *E2F1*, *E2F2*, *E2F3*, *E2F7*. Cells were treated with si-Control, si-E2F1, si-E2F2, or si-E2F3 one day before release with growth media and fixed 16 h after release. Cells were treated with si-Control or si-E2F7 for four hours, incubated in starvation media + EGF (20 ng/mL) for 14 h, and fixed. Box centers are median values, box edges are the 25th and 75th percentiles, and whiskers are minimum and maximum values. p-values were calculated using two-sided, two-sample t tests (*E2F1*: n = 721 cells for si-Control, 729 cells for si-E2F1, p-value = 3.8 × 10⁻¹¹⁹. *E2F2*: n = 721 cells for si-Control, 777 cells for si-E2F2, p-value = 1.1 × 10⁻⁶⁶. *E2F3*: n = 721 cells for si-Control, 731 cells for si-E2F3, p-value = 5.6 × 10⁻¹⁴¹. *E2F7*:

n = 1000 cells for si-Control, 1000 cells for si-E2F7, p-value = 3.0 × 10⁻¹⁴⁵). **e**, E2F ChIP-Seq signals (blue plots) over *CDC6* promoter region used in the E2F reporter (a green box), and detected peaks (magenta lines) with irreproducible discovery rate (IDR) cutoff = 0.05 (publicly available datasets from ENCODE). **f**, Single-cell correlation of mVenus intensity (the E2F reporter) and E2F target gene mRNA puncta area. Cells were fixed 16 h after growth media release. Red lines are median values and light red ranges are the 25th and 75th percentiles for E2F activity bins (n = 2192, 1833, 2836 cells for *E2F1*, *CCNE2*, *CDC6*, respectively. 1 of n = 4 biological replicates). **g**, Box plots of mRNA puncta area of E2F target genes. Cells were released with growth media + DMSO or CDK4/6i (1 μM palbociclib) and fixed 16 h after release. Box centers are median values, box edges are the 25th and 75th percentiles, and whiskers are minimum and maximum values. p-values were calculated using two-sided, two-sample t tests (*E2F1*: n = 806 cells for DMSO, 778 cells for 4/6i, p-value = 1.6 × 10⁻²¹³. *CCNE2*: n = 656 cells for DMSO, 765 cells for 4/6i, p-value = 3.9 × 10⁻¹⁸⁶. *CDC6*: n = 760 cells for DMSO, 768 cells for 4/6i, p-value = 2.8 × 10⁻²⁴⁰. 1 of n = 3 biological replicates). **h**, **j**, Representative images of MCF-10A cells expressing mVenus (the E2F reporter, in the nucleus) and FISH staining for *CDC6* mRNA (puncta in the cytoplasm). Cells were fixed 16 h after release with growth media (**h**) or 36 h after release with growth media + CDK4/6i (1 μM) (**j**). Scale bar = 20 μm. **i**, **k**, Single-cell correlation of mVenus intensity (the E2F reporter) and *CDC6* mRNA puncta area (left) or *CDC6* mRNA puncta area multiplied by its intensity (right). Cells were fixed 16 h after release with growth media (**i**) or 36 h after release with growth media + CDK4/6i (1 μM) (**k**). Red lines are median values and light red ranges are the 25th and 75th percentiles for E2F activity bins (**i**: n = 1637 cells. 1 of n = 4 biological replicates. **k**: n = 2499 cells. 1 of n = 3 biological replicates).

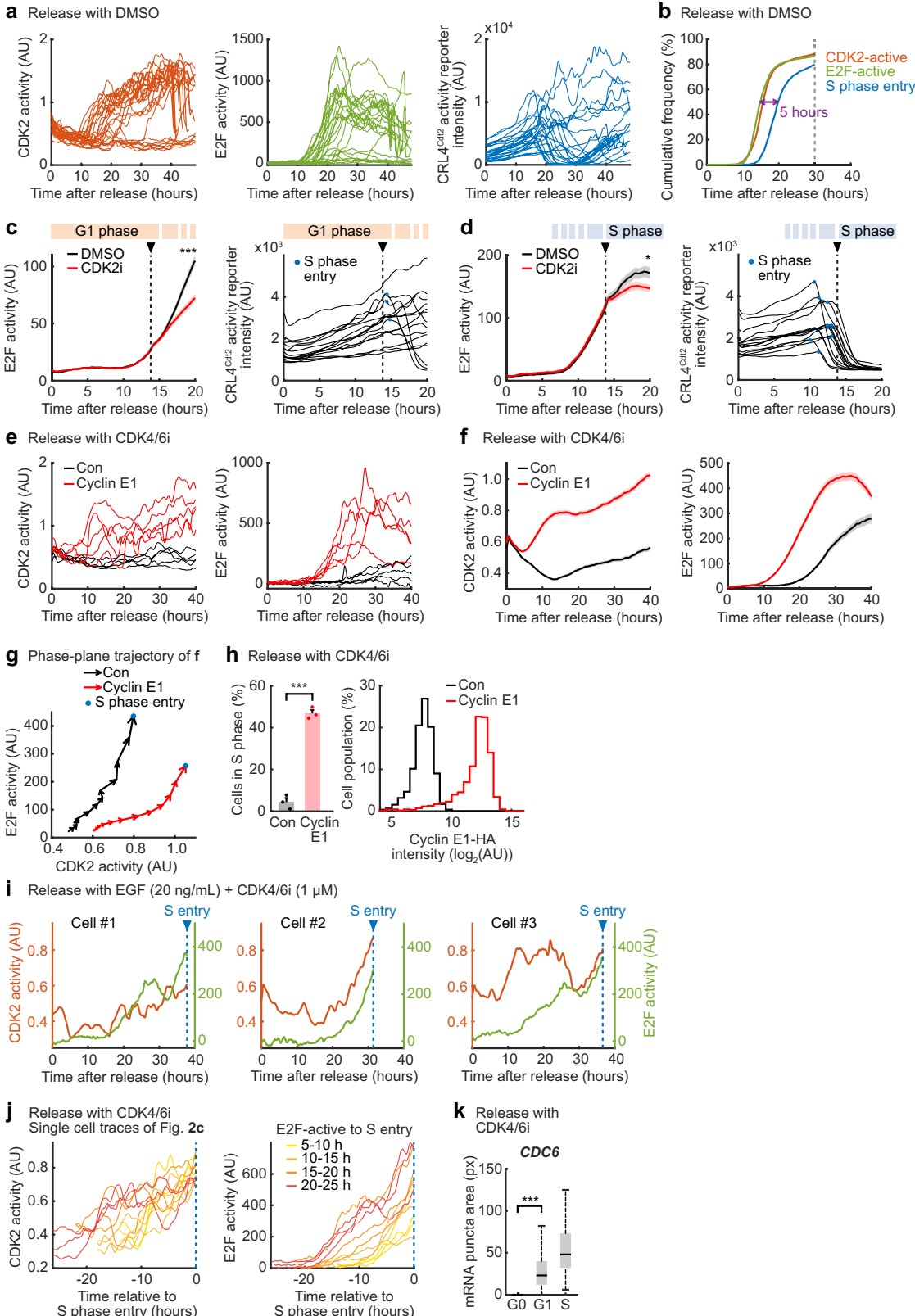

**Extended Data Fig. 3 |** See next page for caption.

**Extended Data Fig. 3 | Cyclin E induces CDK2 and E2F activation in CDK4/6 inhibited cells.** Related to Fig. 2. **a**, Single-cell traces of CDK2 activity, E2F activity, and CRL4$^{Cdt2}$ activity reporter intensity after release with growth media + DMSO for 48 h (n = 25 cells each. 1 of n = 3 biological replicates). **b**, Cumulative frequency of CDK2-activated, E2F-activated, and S phase-entered cells after release with growth media + DMSO. A time gap between E2F-active and S phase entry was calculated at the 50% line (n = 1941 cells. 1 of n = 3 biological replicates). **c, d**, Left, E2F activity traces in cells treated with DMSO and CDK2i (1 μM) 14 h after release with starvation media + EGF (20 ng/mL). Right, Representative CRL4$^{Cdt2}$ reporter traces in cells treated with DMSO 14 h after release with starvation media + EGF (20 ng/mL) to show examples of computational gating. Cells that were in G1 phase (**c**) or S phase (**d**) upon the drug treatment were computationally gated based on the CRL4$^{Cdt2}$ reporter signal. p-values for E2F activity 6 h after the drug treatment (20 h after release) were calculated using two-sided, two-sample t tests (E2F in **c**: mean ± SE. n = 465, 454 cells for DMSO, CDK2i, respectively. p-value = 9.2 × 10$^{-10}$. E2F in **d**: mean ± SE. n = 66, 88 cells for DMSO, CDK2i, respectively. p-value = 0.014. CRL4$^{Cdt2}$: single-cell traces. n = 15 cells each. 1 of n = 3 biological replicates). **e, f**, CDK2 (left) and E2F (right) activity traces in control or Dox-inducible HA-tagged cyclin E1-expressing cells. Dox was added 5 h before release to induce cyclin E1. Cells were released with growth media + CDK4/6i (1 μM) (**e**: single-cell traces. n = 5 cells each. **f**: mean ± SE. n = 205, 363 cells for Con, Cyclin E1, respectively. 1 of n = 3 biological replicates). **g**, Same data in **f** was gated for cells entered S phase and plotted as a phase-plane trajectory. Intervals between arrows are one hour. Blue circles mark the time points for S phase entry, which were determined by the CRL4$^{Cdt2}$ reporter signal (n = 6, 175 cells for Con, Cyclin E1, respectively. 1 of n = 3 biological replicates). **h**, Left, percentage of cells entered S phase by 40 h after release in control or Dox-inducible HA-tagged cyclin E1-expressing cells. Dox was added 5 h before release to induce cyclin E1. Cells were released with growth media + CDK4/6i (1 μM). A p-value was calculated using two-sided, two-sample t tests (mean ± SE. n = 3 biological replicates. n = 703, 834 cells in total for Con, Cyclin E1, respectively. p-value = 7.2 × 10$^{-5}$). Right, histograms of cyclin E1-HA intensity in control or Dox-inducible HA-tagged cyclin E1-expressing cells. Cells were fixed 48 h after release with growth media + CDK4/6i (1 μM) to stain HA and confirm cyclin E1 induction. (n = 1420, 708 cells for Con, Cyclin E1, respectively. 1 of n = 3 biological replicates). **i**, Same data in Fig. 2g (S enter) was shown with CDK2 activity traces. Cells were released with starvation media + EGF (20 ng/mL) + CDK4/6i (1 μM). Dashed lines mark the time points for S phase entry, which were determined by the CRL4$^{Cdt2}$ reporter signal (n = 3 cells. 1 of n = 3 biological replicates). **j**, Same data in Fig. 2c was shown as representative single-cell traces. CDK2 (top) and E2F (bottom) activity traces after release with growth media + CDK4/6i (1 μM). Cells were stratified and color-coded based on the time cells spend from E2F-active to S phase entry. Cell traces were computationally aligned at S phase entry (n = 3 cells each. 1 of n = 3 biological replicates). **k**, Box plots of *CDC6* mRNA puncta area multiplied by its intensity. Cells were starved for two days for measuring *CDC6* mRNA levels in G0. Cells were fixed 36 h after release with growth media + CDK4/6i (1 μM) for measuring *CDC6* mRNA levels in G1 and S. G1 and S cells were gated based on the CRL4$^{Cdt2}$ reporter signal. Box centers are median values, box edges are the 25th and 75th percentiles, and whiskers are minimum and maximum values. A p-value was calculated using two-sided, two-sample t tests (n = 1410, 1124, 102 cells for G0, G1, S cells, respectively. p-value (G0 vs G1) = 2.5 × 10$^{-298}$. 1 of n = 3 biological replicates).

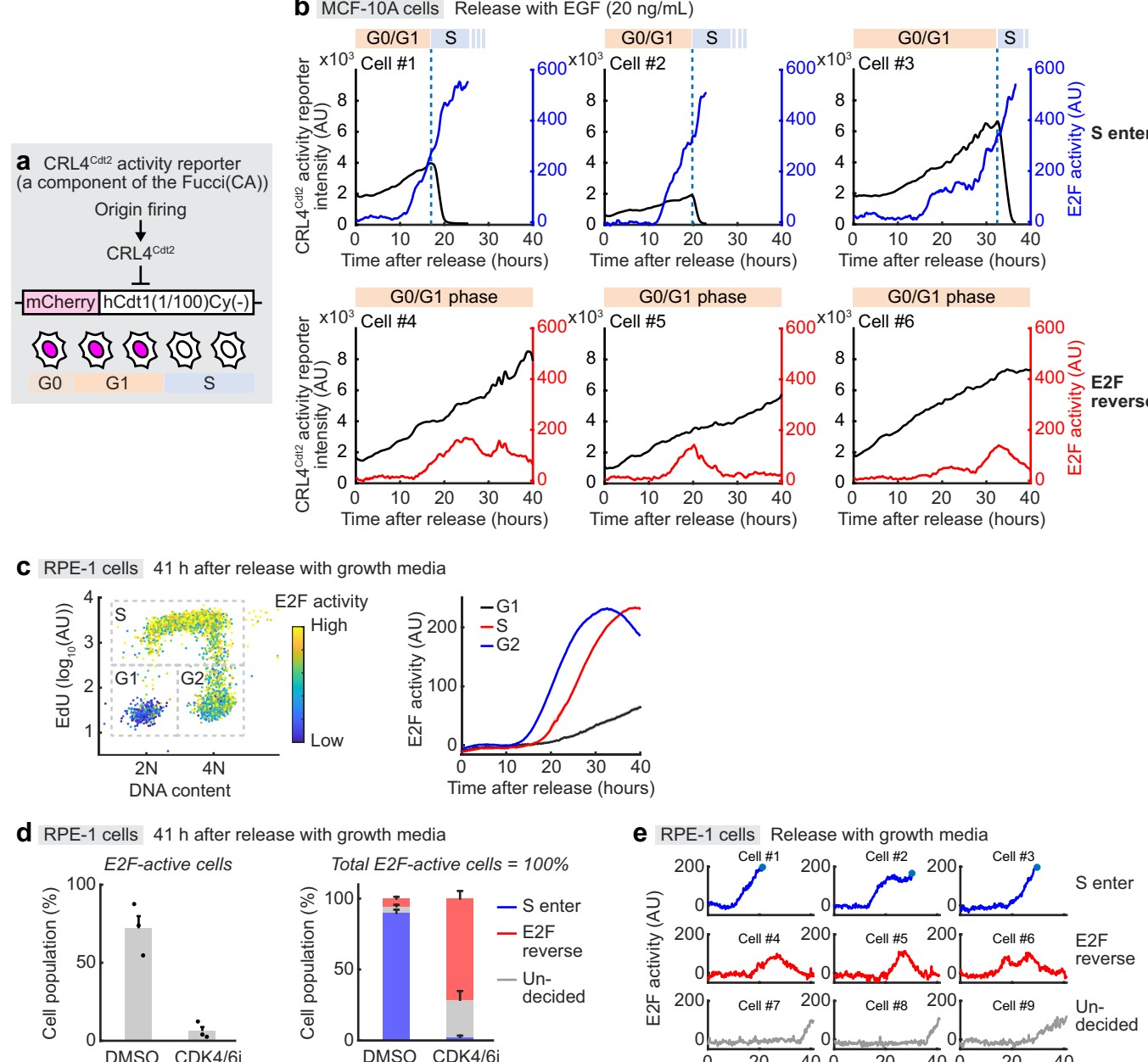

**Extended Data Fig. 4 | Reversible intermediate E2F activation is generalizable across non-transformed cells.** Related to Fig. 2. **a**, Schematic of the CRL4$^{Cdt2}$ reporter, a component of the Fucci(CA) reporter system. **b**, Single-cell traces of E2F and CRL4$^{Cdt2}$ activity as an example for "S enter" and "E2F reverse" cells categorized in Fig. 2f. MCF-10A cells were released with starvation media + EGF (20 ng/mL) (1 of $n$ = 3 biological replicates). **c**, RPE-1 cells expressing the E2F reporter. Left, single-cell correlation of DNA content plotted against EdU incorporation. Dots are color-coded based on E2F activity 41 h after release with growth media. Right, E2F activity traces in cells released with growth media. G1, S, G2 cells 41 h after release are gated based on DNA content and EdU incorporation (mean ± SE. $n$ = 431, 1865, 1466 cells for G1, S, G2 cells, respectively. 1 of $n$ = 3 biological replicates). **d**, RPE-1 cells expressing the E2F reporter. Left, percentage of cells with E2F activation by 41 h after release.

Cells were released with growth media + DMSO or CDK4/6i (1 μM palbociclib). Right, percentage of "S enter", "E2F reverse", "undecided" cells among E2F-activated cells per each condition. Cells were categorized into different groups based on the behaviors by 41 h after release. "S enter" cells are cells that entered S phase, which is determined by DNA content and EdU incorporation. "E2F reverse" cells are cells with E2F activation, without S phase entry, and with E2F activity decrease more than half from the peak. "Undecided" cells are cells with E2F activation, but without S phase entry or E2F activity decrease. (mean ± SE. $n$ = 3 biological replicates. $n$ = 11492, 18886 cells in total for DMSO, CDK4/6i, respectively). **e**, RPE-1 cells expressing the E2F reporter. Single-cell traces of E2F activity as an example for "S enter", "E2F reverse", "undecided" cells categorized in **d**. Cells were released with growth media + DMSO.

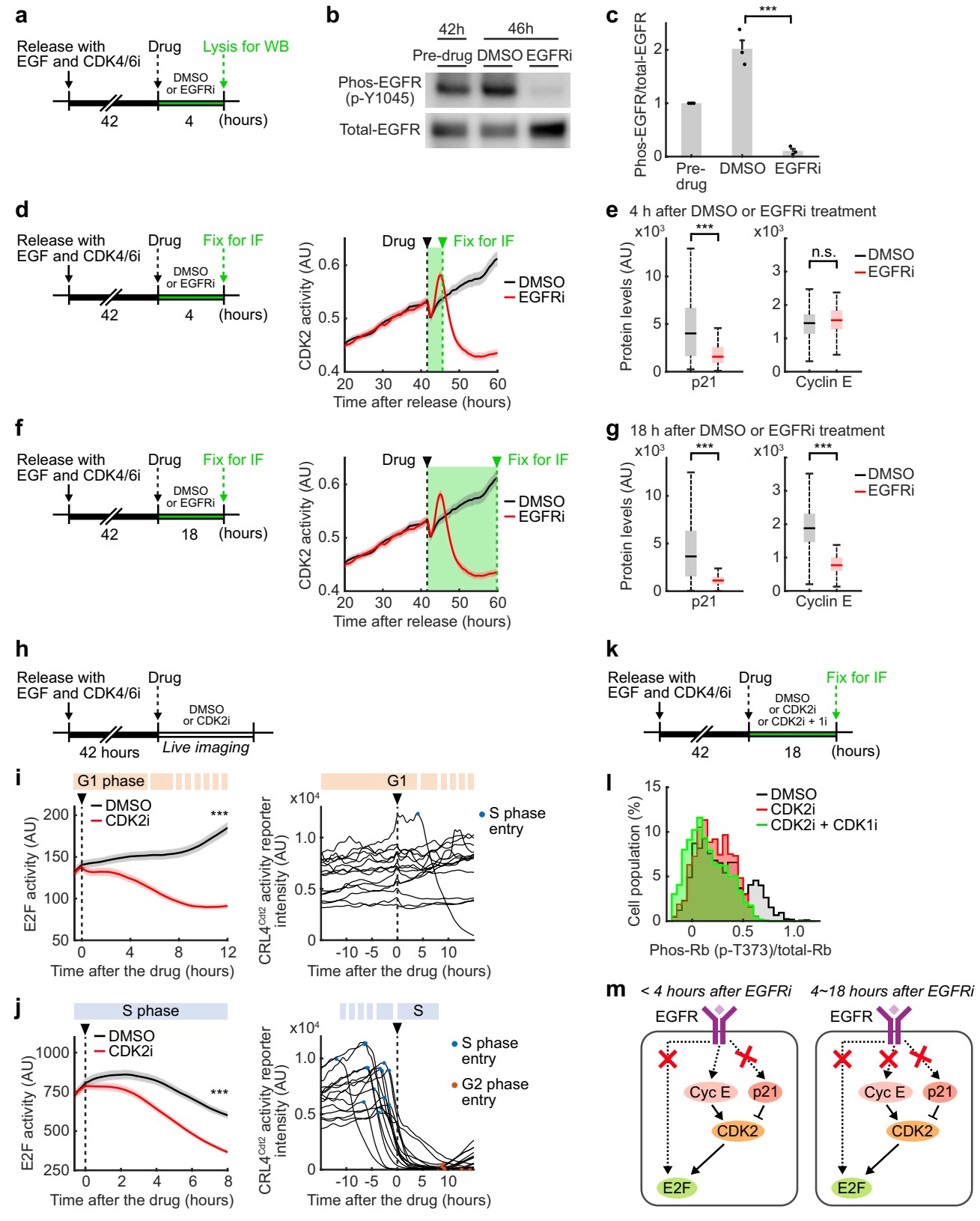

**Extended Data Fig. 5** | See next page for caption.

**Extended Data Fig. 5 | CDK2 and E2F are activated via EGFR signaling in CDK4/6 inhibited cells.** Related to Fig. 2. **a**, Experimental design for **b** and **c**. Cells were treated with DMSO or EGFRi (5 μM gefitinib) 42 h after release with starvation media + EGF (20 ng/mL) + CDK4/6i. Cells were lysed 4 h after DMSO or EGFRi treatment for western blots (Supplementary Fig. 1). **b**, **c**, Western blots of phospho-EGFR (p-Y1045) and total-EGFR (**b**) and quantification (**c**). A p-value was calculated using two-sided, two-sample t tests (**b**: 1 of $n = 3$ biological replicates. **c**: mean ± SE. $n = 3$ biological replicates. p-value (DMSO vs EGFRi) = $6.3 \times 10^{-4}$). **d**, Experimental design for **e**. Cells were treated with DMSO or EGFRi (5 μM gefitinib) 42 h after release with starvation media + EGF (20 ng/mL) + CDK4/6i. Cells were fixed 4 h after DMSO or EGFRi treatment for immunofluorescence (mean ± SE). **e**, Box plots of p21 or cyclin E protein levels. Box centers are median values, box edges are the 25th and 75th percentiles, and whiskers are minimum and maximum values. p-values were calculated using two-sided, two-sample t tests (p21: $n = 100$ cells for DMSO, 100 cells for EGFRi, p-value = $2.5 \times 10^{-10}$. Cyclin E: $n = 100$ cells for DMSO, 100 cells for EGFRi, p-value = 0.39 (n.s.). 1 of $n = 2$ biological replicates). **f**, Experimental design for **g**. Cells were treated with DMSO or EGFRi (5 μM gefitinib) 42 h after release with starvation media + EGF (20 ng/mL) + CDK4/6i. Cells were fixed 18 h after DMSO or EGFRi treatment for immunofluorescence (mean ± SE). **g**, Box plots of p21 or cyclin E protein levels. Box centers are median values, box edges are the 25th and 75th percentiles, and whiskers are minimum and maximum values. p-values were calculated using two-sided, two-sample t tests (p21: n = 100 cells for DMSO, 100 cells for EGFRi, p-value = $1.2 \times 10^{-14}$. Cyclin E: n = 100 cells for DMSO, 100 cells for EGFRi, p-value = $1.6 \times 10^{-30}$. 1 of $n = 3$ biological replicates). **h**, Experimental design for **i** and **j**. Cells were treated with DMSO or CDK2i (1 μM) 42 h after release with starvation media + EGF (20 ng/mL) + CDK4/6i for live-cell imaging. **i**, **j**, Left, E2F activity traces in cells treated with DMSO and CDK2i (1 μM). Right, Representative CRL4$^{Cdt2}$ reporter traces in cells treated with DMSO to show examples of computational gating. Cells that were in G1 phase (**i**) or S phase (**j**) upon the drug treatment were computationally gated based on the CRL4$^{Cdt2}$ reporter signal. p-values for E2F activity 12 h (**i**) or 8 h (**j**) after the drug treatment was calculated using two-sided, two-sample t tests (E2F in **i**: mean ± SE. $n = 892$, 876 cells for DMSO, CDK2i, respectively. p-value = $1.1 \times 10^{-38}$. E2F in **j**: mean ± SE. $n = 128, 145$ cells for DMSO, CDK2i, respectively. p-value = $1.3 \times 10^{-22}$. CRL4$^{Cdt2}$: single-cell traces. $n = 15$ cells each. 1 of $n = 3$ biological replicates). **k**, Experimental design for **i**. Cells were treated with DMSO or CDK2i or CDK2i + CDK1i (10 μM RO-3306) 42 h after release with starvation media + EGF (20 ng/mL) + CDK4/6i. Cells were fixed 18 h after the drug treatment for immunofluorescence. **l**, Histograms of Rb phosphorylation at T373 ($n = 1825, 1492, 1725$ cells for DMSO, CDK2i, CDK2i + CDK1i, respectively. 1 of $n = 3$ biological replicates). **m**, Model for the effect of EGFRi (gefitinib) on cell cycle regulators in cells released with EGF and CDK4/6i. During the first 4 h after EGFRi, E2F is inactivated whereas CDK2 is temporarily activated due to p21 downregulation. In contrast, 4 to 18 h after EGFRi, both E2F and CDK2 are inactivated because cyclin E is also downregulated.

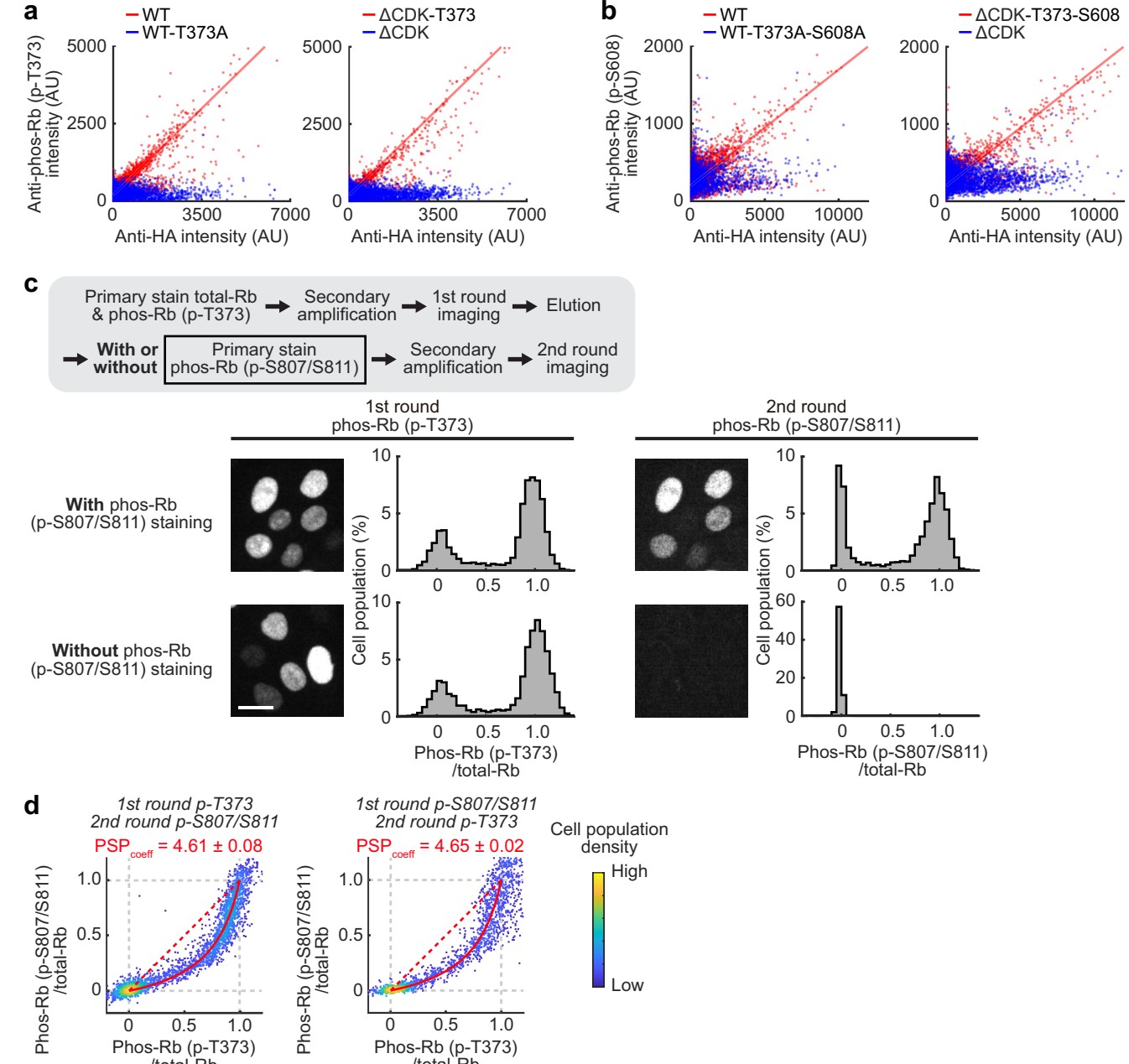

**Extended Data Fig. 6 | Validation of the antibodies and the iterative indirect immunofluorescence imaging (4i).** Related to Fig. 3. **a**, **b**, Single-cell correlation of anti-HA intensity and anti-phospho-Rb (p-T373 for **a**, p-S608 for **b**) intensity in cells expressing the Dox-inducible HA-tagged Rb constructs (measured by immunofluorescence). Cells were treated with si-RB1 one day before release to knockdown endogenous Rb, treated with Dox 5 h before release to induce Rb constructs, and fixed 36 h after release with growth media. Cells with the efficient RB1 knockdown were gated based on the correlation between anti-HA and anti-total-Rb. In **a**, Rb-WT-T373A and ΔCDK-T373 were compared to Rb-WT and ΔCDK, respectively and red reference lines are $y = 0.8x + 200$. In **b**, Rb-WT-T373A-S608A and ΔCDK-T373-S608 were compared to Rb-WT and ΔCDK, respectively and red reference lines are $y = 0.15x + 200$ ($n = 3803, 3463, 2857, 3607$ cells for Rb-WT, Rb-WT-T373A, ΔCDK-T373, ΔCDK, respectively (**a**). $n = 3863, 2687, 3518, 3499$ cells for Rb-WT, Rb-WT-T373A-S608A, ΔCDK-T373-S608, ΔCDK, respectively (**b**)). **c**, Workflow of the iterative indirect immunofluorescence imaging (4i) protocol. Representative images and histograms of Rb phosphorylation at T373 and S807/S811. Cells were fixed 14 h after release with starvation media + EGF (20 ng/mL). To validate the signal in the second round of 4i is due to anti-phospho-Rb (p-S807/S811) antibody,

4i was performed with (top) or without (bottom) phospho-Rb (p-S807/S811) staining, while keeping the rest same. The same cells across two rounds of 4i are shown for each protocol (top or bottom). Scale bar = 20 μm (Histograms: $n = 2960, 2765$ cells for with or without phospho-Rb (p-S807/S811) staining, respectively). **d**, Phosphorylation Site Preference (PSP) plots showing single-cell correlation of Rb phosphorylation at T373 plotted against S807/S811 or vice versa. Cells were fixed 16 h after release with starvation media + EGF (20 ng/mL) + CDK4/6i (20 nM). Left, cells were stained with anti-phospho-Rb (p-T373) antibody in the first round and anti-phospho-Rb (p-S807/S811) antibody in the second round of 4i. Right, cells were stained with anti-phospho-Rb (p-S807/S811) antibody in the first round and anti-phospho-Rb (p-T373) antibody in the second round of 4i. Each phosphorylation signal was normalized by the total Rb antibody signal in the same cell and each axis was adjusted to the average phosphorylation signal in S phase of 1. Color indicates relative cell population density. A red line shows fitting with a preferential relative phosphorylation or dephosphorylation rate between the two sites ($PSP_{coeff}$) (see more details in Methods) ($n = 2734, 1727$ cells for left and right, respectively. 1 of $n = 2$ biological replicates).

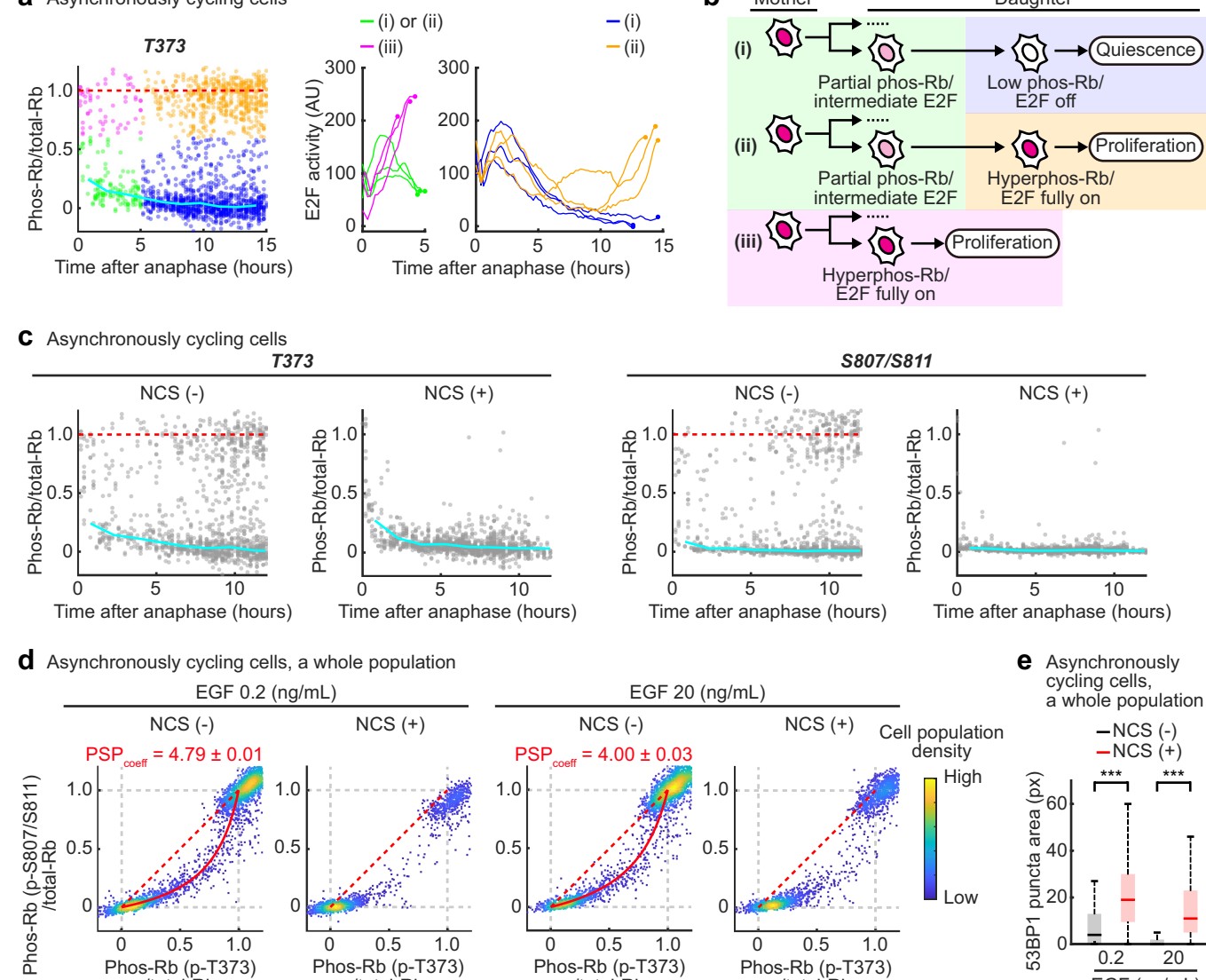

**Extended Data Fig. 7 | Preferential phosphorylation of Rb at T373 is generalizable in cycling cells, and slower dephosphorylation at T373 in Rb compared to other sites creates partially phosphorylated Rb/intermediate E2F activity states in cycling cells.** Related to Figs. 3 and 5. **a**, Left, Rb phosphorylation at T373 after anaphase measured by immunofluorescence. Cells were asynchronously cycling in starvation media + EGF (0.2 ng/mL). Pseudo-time course immunofluorescence was measured by computationally aligning cells at anaphase based on live-cell images. Cells were color-coded based on Rb phosphorylation at T373 (green: Rb phosphorylation at T373 0.6, 0-5 h after anaphase, blue: Rb phosphorylation at T373 0.6, 5-15 h). A cyan line shows median levels of Rb phosphorylation at T373 = 1.0 (n = 3 biological replicates. n = 1719 cells in total). Right, representative single-cell traces of E2F activity categorized in Left (n = 3 cells each. 1 of n = 3 biological replicates). **b**, Model for three different cell fates based on Rb phosphorylation across cell generations. The background color code is matched with the categorization in **a**. (i) Daughter cells are born with partially phosphorylated Rb involving p-T373 (green) and dephosphorylate Rb completely to undergo quiescence (blue). (ii) Daughter cells are born with partially phosphorylated Rb involving p-T373 (green) and hyperphosphorylate Rb to enter S phase (orange). (iii) Daughter cells are born with hyperphosphorylated Rb and enter S phase quickly (magenta). **c**, Rb dephosphorylation kinetics after anaphase at T373 (left) and S807/S811 (right) measured by immunofluorescence. Cells were asynchronously cycling in starvation media + EGF (0.2 ng/mL). Cells were treated with or without 50 ng/mL neocarzinostatin (NCS) for 20 min, and incubated in starvation media + EGF (0.2 ng/mL) again for 16 h before fixation.

Pseudo-time course immunofluorescence was measured by computationally aligning cells at anaphase based on live-cell images. Cells that were born 1 h to 16 h after NCS or mock treatment were analyzed. A cyan line shows median levels of Rb phosphorylation <0.6 and a red dotted line shows a reference line of Rb phosphorylation = 1.0 (n = 3 biological replicates. n = 1719, 1160 cells in total for NCS (−), NCS (+), respectively. **d**, Phosphorylation Site Preference (PSP) plots showing single-cell correlation of Rb phosphorylation at T373 plotted against S807/S811. Each phosphorylation signal was normalized by the total Rb antibody signal in the same cell and each axis was adjusted to the average phosphorylation signal in S phase of 1. Color indicates relative cell population density. A red line shows fitting with a preferential relative phosphorylation or dephosphorylation rate between the two sites ($PSP_{coeff}$) (see more details in Methods). Cells asynchronously cycling in starvation media + EGF (0.2 or 20 ng/mL) were fixed 16 h after 50 ng/mL neocarzinostatin (NCS) treatment for 20 min or mock treatment (n = 3393, 1904, 3079, 2247 cells for NCS (−) in EGF (0.2 ng/mL), NCS (+) in EGF (0.2 ng/mL), NCS (−) in EGF (20 ng/mL), NCS (+) in EGF (20 ng/mL), respectively. 1 of n = 3 biological replicates). **e**, Box plots of 53BP1 puncta area. Box centers are median values, box edges are the 25th and 75th percentiles, and whiskers are minimum and maximum values. Cells asynchronously cycling in starvation media + EGF (0.2 or 20 ng/mL) were fixed 16 h after 50 ng/mL neocarzinostatin (NCS) treatment for 20 min or mock treatment. p-values were calculated using two-sided, two-sample t tests (n = 100 cells each, p-value (NCS (−) vs (+) in EGF (0.2 ng/mL)) = $2.0 \times 10^{-14}$. p-value (NCS (−) vs (+) in EGF (20 ng/mL)) = $2.5 \times 10^{-18}$. 1 of n = 2 biological replicates).

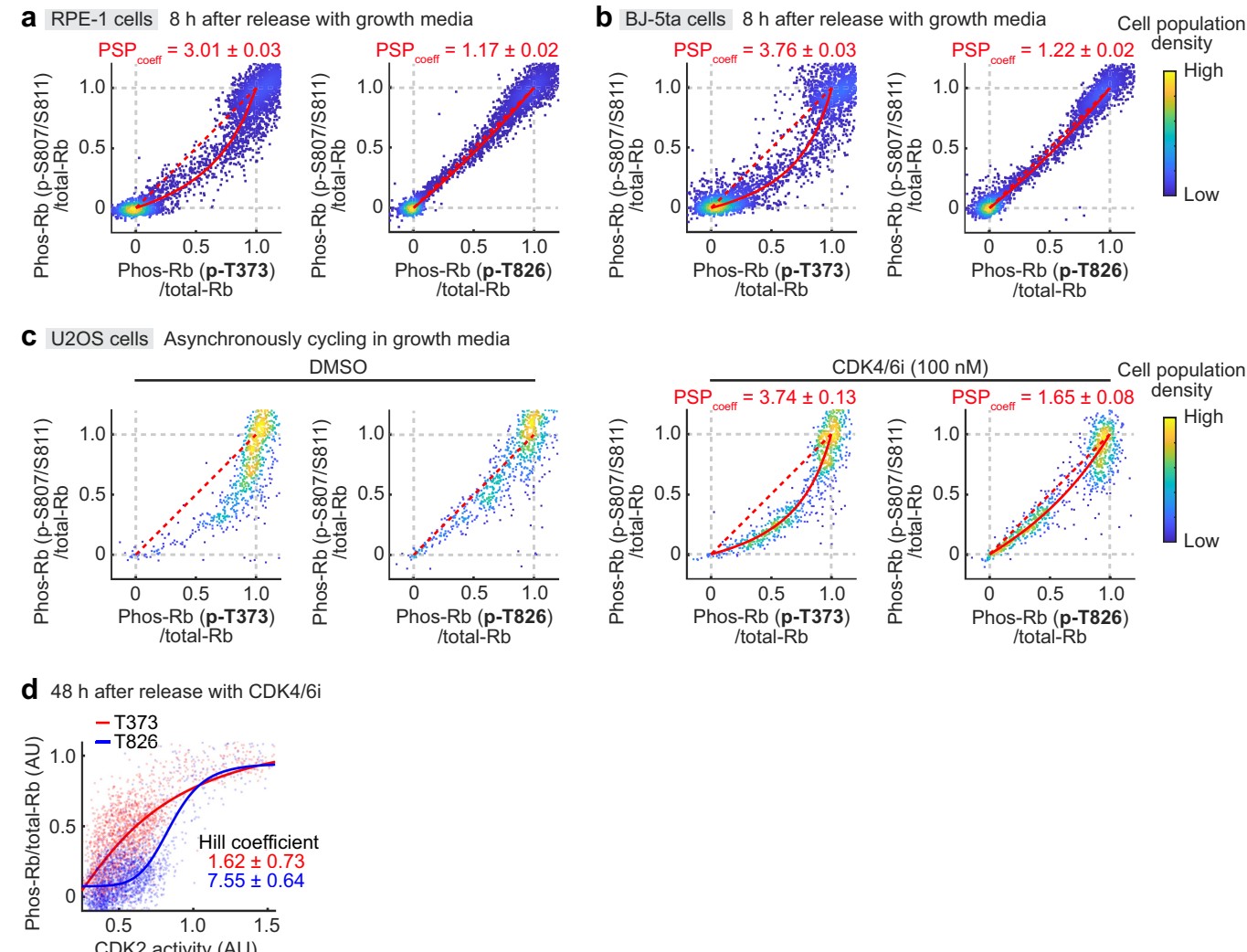

**Extended Data Fig. 8 | Preferential phosphorylation of Rb at T373 is generalizable across non-transformed cells, and phosphorylation of the C-terminal sites of Rb is part of the cooperative Rb hyperphosphorylation mechanism.** Related to Figs. 3 and 4. **a**–**c**, Phosphorylation Site Preference (PSP) plots showing single-cell correlation of Rb phosphorylation at different sites in RPE-1 cells (**a**), BJ-5ta cells (**b**), or U2OS cells (**c**). Each phosphorylation signal was normalized by the total Rb antibody signal in the same cell and each axis was adjusted to the average phosphorylation signal in S phase of 1. Color indicates relative cell population density. A red line shows fitting with a preferential relative phosphorylation or dephosphorylation rate between the two sites (PSP$_{coeff}$) (see more details in Methods). Rb phosphorylation

at T373 or T826 was plotted against S807/S811. **a**,**b**, Cells were fixed 8 h after release with growth media (**a**: $n$ = 5019, 4900 cells for T373, T826, respectively. **b**: $n$ = 3409, 3502 cells for T373, T826, respectively. 1 of $n$ = 3 biological replicates). **c**, Cells asynchronously cycling in growth media were fixed 6 h after DMSO or CDK4/6i (100 nM) treatment (DMSO: $n$ = 704, 700 cells for T373, T826, respectively. CDK4/6i: $n$ = 850, 878 cells for T373, T826, respectively. 1 of $n$ = 3 biological replicates). **d**, Single-cell correlation of CDK2 activity and Rb phosphorylation at T373 and T826. Cells were fixed 48 h after release with starvation media + EGF (20 ng/mL) + CDK4/6i (1 μM). Rb phosphorylation levels as a function CDK2 activity were fitted by sigmoidal curves (sigmoidal curves with Hill coefficients. $n$ = 1967, 2050 cells for T373, T826, respectively).

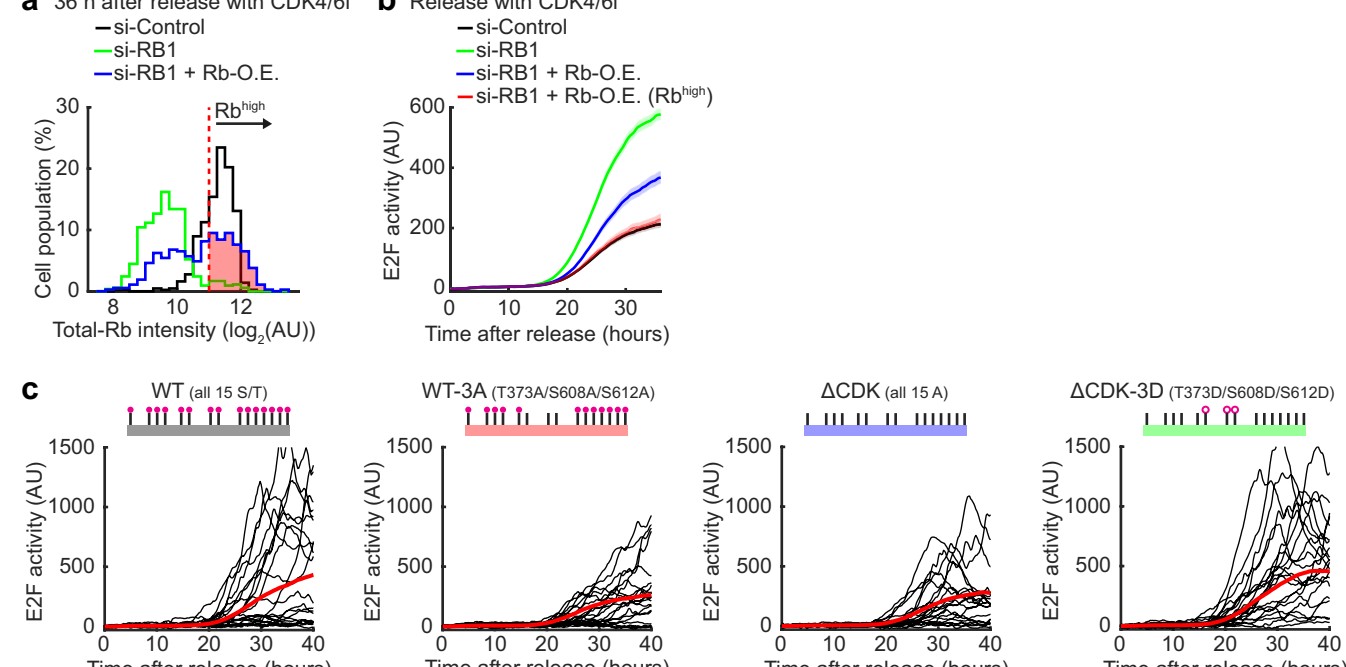

**Extended Data Fig. 9 | Functions of T373 site in Rb are tested in the Rb mutant re-expression assay.** Related to Fig. 5. **a**, **b**, Histograms of total-Rb intensity (**a**) and E2F activity traces (**b**) in cells with or without RB1 knockdown or with RB1 knockdown + exogenous Rb expression. Cells were fixed 36 h after release with growth media + CDK4/6i. Cells were treated with si-RB1 one day before release to knockdown endogenous Rb and treated with Dox 5 h before release to induce exogenous Rb. A threshold for $Rb^{high}$ is $Rb > 2^{11}$ (n = 435, 537, 368, 173 cells for si-Control, si-RB1, si-RB1 + Rb-O.E., si-RB1 + $Rb^{high}$, respectively. E2F: mean ± SE). **c**, single-cell traces of E2F activity in cells expressing the

Dox-inducible HA-tagged Rb constructs. Cells were released with growth media + CDK4/6i. Cells were treated with si-RB1 one day before release to knockdown endogenous Rb and treated with Dox 5 h before release to induce exogenous Rb. To compare the same expression levels of Rb constructs, cells with $2^{10} <$ HA $< 2^{11}$ were gated for analysis (black lines denote the example single-cell traces of 20 cells from one representative experiment. red lines denote the mean of a whole population. n = 323, 288, 383, 246 cells for Rb-WT, Rb-WT-3A, Rb-ΔCDK, Rb-ΔCDK-3D, respectively. n = 4 biological replicates for Rb-WT, Rb-WT-3A, Rb-ΔCDK, and 3 biological replicates for Rb-ΔCDK-3D).

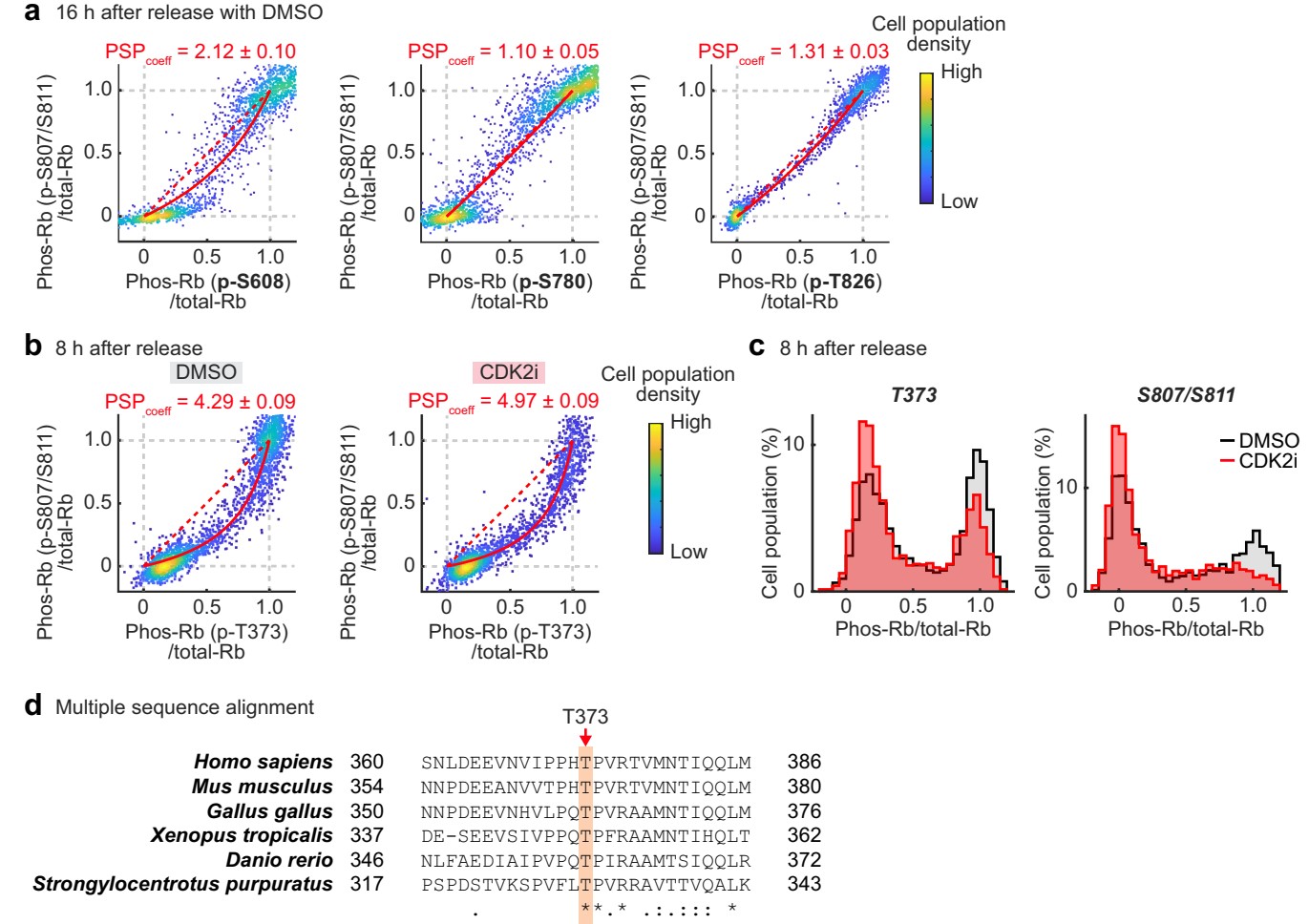

# Reporting Summary

## Statistics

For all statistical analyses, confirm that the following items are present in the figure legend, table legend, main text, or Methods section.

| n/a | Confirmed | |
|---|---|---|
| ☐ | ☒ | The exact sample size (*n*) for each experimental group/condition, given as a discrete number and unit of measurement |
| ☐ | ☒ | A statement on whether measurements were taken from distinct samples or whether the same sample was measured repeatedly |
| ☐ | ☒ | The statistical test(s) used AND whether they are one- or two-sided *Only common tests should be described solely by name; describe more complex techniques in the Methods section.* |
| ☒ | ☐ | A description of all covariates tested |
| ☐ | ☒ | A description of any assumptions or corrections, such as tests of normality and adjustment for multiple comparisons |
| ☐ | ☒ | A full description of the statistical parameters including central tendency (e.g. means) or other basic estimates (e.g. regression coefficient) AND variation (e.g. standard deviation) or associated estimates of uncertainty (e.g. confidence intervals) |
| ☐ | ☒ | For null hypothesis testing, the test statistic (e.g. *F*, *t*, *r*) with confidence intervals, effect sizes, degrees of freedom and *P* value noted *Give P values as exact values whenever suitable.* |
| ☒ | ☐ | For Bayesian analysis, information on the choice of priors and Markov chain Monte Carlo settings |
| ☒ | ☐ | For hierarchical and complex designs, identification of the appropriate level for tests and full reporting of outcomes |
| ☒ | ☐ | Estimates of effect sizes (e.g. Cohen's *d*, Pearson's *r*), indicating how they were calculated |

*Our web collection on statistics for biologists contains articles on many of the points above.*

## Software and code

Policy information about availability of computer code

| Data collection | MetaExpress version 6.1, a software provided by Molecular Devices, was used for acquisition of microscopy images on ImageXpress Micro. NIS-Elements AR version 5.21.03, a software provided by Nikon, was used for acquisition of microscopy images on Ti2-E. |
|---|---|
| Data analysis | A repository of Matlab code used for cell segmentation, tracking, and analysis is cited in the methods section. |

For manuscripts utilizing custom algorithms or software that are central to the research but not yet described in published literature, software must be made available to editors and reviewers. We strongly encourage code deposition in a community repository (e.g. GitHub). See the Nature Portfolio guidelines for submitting code & software for further information.

## Data

Policy information about availability of data

All manuscripts must include a data availability statement. This statement should provide the following information, where applicable:

- Accession codes, unique identifiers, or web links for publicly available datasets
- A description of any restrictions on data availability
- For clinical datasets or third party data, please ensure that the statement adheres to our policy

Raw imaging data acquired during this study have not been deposited in a public repository due to storage limitations but are available from the lead contact upon request.

# Research involving human participants, their data, or biological material

Policy information about studies with [human participants or human data](). See also policy information about [sex, gender (identity/presentation), and sexual orientation]() and [race, ethnicity and racism]().

| | |
|---|---|
| Reporting on sex and gender | *Use the terms sex (biological attribute) and gender (shaped by social and cultural circumstances) carefully in order to avoid confusing both terms. Indicate if findings apply to only one sex or gender; describe whether sex and gender were considered in study design; whether sex and/or gender was determined based on self-reporting or assigned and methods used.*<br>*Provide in the source data disaggregated sex and gender data, where this information has been collected, and if consent has been obtained for sharing of individual-level data; provide overall numbers in this Reporting Summary. Please state if this information has not been collected.*<br>*Report sex- and gender-based analyses where performed, justify reasons for lack of sex- and gender-based analysis.* |
| Reporting on race, ethnicity, or other socially relevant groupings | *Please specify the socially constructed or socially relevant categorization variable(s) used in your manuscript and explain why they were used. Please note that such variables should not be used as proxies for other socially constructed/relevant variables (for example, race or ethnicity should not be used as a proxy for socioeconomic status).*<br>*Provide clear definitions of the relevant terms used, how they were provided (by the participants/respondents, the researchers, or third parties), and the method(s) used to classify people into the different categories (e.g. self-report, census or administrative data, social media data, etc.)*<br>*Please provide details about how you controlled for confounding variables in your analyses.* |
| Population characteristics | *Describe the covariate-relevant population characteristics of the human research participants (e.g. age, genotypic information, past and current diagnosis and treatment categories). If you filled out the behavioural & social sciences study design questions and have nothing to add here, write "See above."* |
| Recruitment | *Describe how participants were recruited. Outline any potential self-selection bias or other biases that may be present and how these are likely to impact results.* |
| Ethics oversight | *Identify the organization(s) that approved the study protocol.* |

Note that full information on the approval of the study protocol must also be provided in the manuscript.

# Field-specific reporting

Please select the one below that is the best fit for your research. If you are not sure, read the appropriate sections before making your selection.

☒ Life sciences ☐ Behavioural & social sciences ☐ Ecological, evolutionary & environmental sciences

For a reference copy of the document with all sections, see [nature.com/documents/nr-reporting-summary-flat.pdf]()

# Life sciences study design

All studies must disclose on these points even when the disclosure is negative.

| | |
|---|---|
| Sample size | No sample-size calculation was performed. Sample size was determined to be adequate based on the magnitude and consistency of the measurable differences between groups. All experiments were repeated on multiple days to ensure reproducibility. For experiments where significance was calculated, n of at least 3 was predetermined to enable adequate statistical testing. |
| Data exclusions | Any failed experiments where cells were at suboptimal confluency, CO2, or temperature were excluded. For live-cell imaging analysis, cells were excluded if they could not be distinguished from a neighboring cell or were not tracked for a significant period of time in the experiment. |
| Replication | Attempts at replication were successful, and all experiments were repeated on multiple days. Number of times experiments were repeated can be found in the figure legends. Experiments where significance were derived have at least n=3. |
| Randomization | In cell line experiments, no randomization was required because the experimental cells came from the same source and were tested identically with the variable treatment. Cell lines were used for the entirety of this study; no other sample type that would require randomization was used in this study. |
| Blinding | Blinding was not done. The same automated analysis pipeline was applied to all conditions. Blinding was not considered based upon experience and similar published studies. |

# Reporting for specific materials, systems and methods

We require information from authors about some types of materials, experimental systems and methods used in many studies. Here, indicate whether each material, system or method listed is relevant to your study. If you are not sure if a list item applies to your research, read the appropriate section before selecting a response.

## Materials & experimental systems

| n/a | Involved in the study |
|---|---|
| ☐ | ☒ Antibodies |
| ☐ | ☒ Eukaryotic cell lines |
| ☒ | ☐ Palaeontology and archaeology |
| ☒ | ☐ Animals and other organisms |
| ☒ | ☐ Clinical data |
| ☒ | ☐ Dual use research of concern |
| ☒ | ☐ Plants |

## Methods

| n/a | Involved in the study |
|---|---|
| ☐ | ☒ ChIP-seq |
| ☒ | ☐ Flow cytometry |
| ☒ | ☐ MRI-based neuroimaging |

# Antibodies

| | |
|---|---|
| Antibodies used | All antibody catalog numbers are provided in the methods section. |
| Validation | Antibodies validated through siRNA knockdown and cellular localization using immunofluorescence: mouse anti-Rb antibody (BD Cat# 554136), rabbit anti-phospho-Rb (S807/S811) antibody (Cell Signaling Technology Cat# 8516), rabbit anti-p21 antibody (Cell Signaling Technology Cat# 2947), mouse anti-cyclin E antibody (Santa Cruz Cat# sc-247), rabbit anti-c-Myc antibody (Cell Signaling Technology #5605), rabbit anti-53BP1 antibody (Cell Signaling Technology Cat# 4937)<br><br>Antibodies validated through overexpression using immunofluorescence: rabbit anti-phospho-Rb (T373) antibody (Abcam Cat# ab52975), rabbit anti-phospho-Rb (S608) antibody (Cell Signaling Technology Cat# 2181), rabbit anti-HA tag antibody (Cell Signaling Technology Cat# 3724), mouse anti-HA tag antibody (BioLegend Cat# 901501)<br><br>Antibodies validated through band size using Western blots: mouse anti-Rb antibody (Cell Signaling Technology Cat# 9309), all anti-phospho-Rb antibodies were further validated with CDK4/6 inhibitor treatment.<br><br>Statements on manufacturer's website or comments about citations where relevant:<br>mouse anti-Rb antibody (BD Cat# 554136): "G3-245 was made using a Trp-E-Rb fusion protein as immunogen and recognizes an epitope between amino acids 332-344 (DARLFDHDKTLQ) of the human retinoblastoma protein (pp110-114 Rb)."<br><br>mouse anti-Rb antibody (Cell Signaling Technology Cat# 9309): "Rb (4H1) Mouse mAb detects endogenous levels of total Rb protein. The antibody does not cross-react with the Rb homologues p107 or p130, or with other proteins."<br><br>rabbit anti-phospho-Rb (S807/S811) antibody (Cell Signaling Technology Cat# 8516): "Phospho-Rb (Ser807/811) (D20B12) XP® Rabbit mAb recognizes endogenous levels of Rb protein only when phosphorylated at Ser807, Ser811, or at both sites. This antibody does not cross-react with Rb phosphorylated at Ser608."<br><br>rabbit anti-p21 antibody (Cell Signaling Technology Cat# 2947): "p21 Waf1/Cip1 (12D1) Rabbit mAb detects endogenous levels of total p21 protein. The antibody does not cross-react with other CDK inhibitors." |

# Eukaryotic cell lines

Policy information about cell lines and Sex and Gender in Research

| | |
|---|---|
| Cell line source(s) | MCF-10A, RPE-1, BJ-5ta, U2OS cells were acquired directly from ATCC. Lenti-X 293T cells were acquired directly from Takara Bio. |
| Authentication | MCF-10A cells were validated by RNA-seq. Other cell lines were not validated. |
| Mycoplasma contamination | All cell lines tested negative for mycoplasma contamination. Our laboratory routilinely tests for myoplasma by a PCR test and by Hoechst 33342 DNA staining. |
| Commonly misidentified lines<br>(See ICLAC register) | No commonly misidentified lines were used in this study. |

# Plants

| | |
|---|---|
| Seed stocks | *Report on the source of all seed stocks or other plant material used. If applicable, state the seed stock centre and catalogue number. If plant specimens were collected from the field, describe the collection location, date and sampling procedures.* |
| Novel plant genotypes | *Describe the methods by which all novel plant genotypes were produced. This includes those generated by transgenic approaches, gene editing, chemical/radiation-based mutagenesis and hybridization. For transgenic lines, describe the transformation method, the number of independent lines analyzed and the generation upon which experiments were performed. For gene-edited lines, describe the editor used, the endogenous sequence targeted for editing, the targeting guide RNA sequence (if applicable) and how the editor was applied.* |
| Authentication | *Describe any authentication procedures for each seed stock used or novel genotype generated. Describe any experiments used to assess the effect of a mutation and, where applicable, how potential secondary effects (e.g. second site T-DNA insertions, mosiacism, off-target gene editing) were examined.* |

# ChIP-seq

## Data deposition

☒ Confirm that both raw and final processed data have been deposited in a public database such as GEO.

☐ Confirm that you have deposited or provided access to graph files (e.g. BED files) for the called peaks.

| | |
|---|---|
| Data access links<br>*May remain private before publication.* | Publicly available datasets downloaded from ENCODE are presented in our study. |
| Files in database submission | E2F1 in HepG2: https://www.encodeproject.org/experiments/ENCSR717ZZW/<br>E2F1 in MCF-7: https://www.encodeproject.org/experiments/ENCSR000EWX/<br>E2F2 in HepG2: https://www.encodeproject.org/experiments/ENCSR013RNH/<br>E2F3 in K562: https://www.encodeproject.org/experiments/ENCSR036QIR/<br>E2F4 in HepG2: https://www.encodeproject.org/experiments/ENCSR924LSO/<br>E2F4 in MCF-7: https://www.encodeproject.org/experiments/ENCSR505NMN/<br>E2F5 in HepG2: https://www.encodeproject.org/experiments/ENCSR486JYI/<br>E2F6 in A549: https://www.encodeproject.org/experiments/ENCSR000BTC/<br>E2F7 in K562: https://www.encodeproject.org/experiments/ENCSR171CAY/<br>E2F8 in HepG2: https://www.encodeproject.org/experiments/ENCSR634GEO/<br>E2F8 in MCF-7: https://www.encodeproject.org/experiments/ENCSR897ZXU/ |
| Genome browser session<br>(e.g. UCSC) | *Provide a link to an anonymized genome browser session for "Initial submission" and "Revised version" documents only, to enable peer review.  Write "no longer applicable" for "Final submission" documents.* |

## Methodology

| | |
|---|---|
| Replicates | *Describe the experimental replicates, specifying number, type and replicate agreement.* |
| Sequencing depth | *Describe the sequencing depth for each experiment, providing the total number of reads, uniquely mapped reads, length of reads and whether they were paired- or single-end.* |
| Antibodies | *Describe the antibodies used for the ChIP-seq experiments; as applicable, provide supplier name, catalog number, clone name, and lot number.* |
| Peak calling parameters | *Specify the command line program and parameters used for read mapping and peak calling, including the ChIP, control and index files used.* |
| Data quality | *Describe the methods used to ensure data quality in full detail, including how many peaks are at FDR 5% and above 5-fold enrichment.* |
| Software | *Describe the software used to collect and analyze the ChIP-seq data. For custom code that has been deposited into a community repository, provide accession details.* |

