## [Peer Review file · Nature]

Manuscript Title: A state of partial Rb inactivation and E2F activation safeguards proliferation commitment

Reviewer Comments & Author Rebuttals

Reviewer Reports on the Initial Version:

Referees' comments:

Referee #1 (Remarks to the Author):

Overview

In this work, Konagaya and colleagues aim to address an important question – how do cells only trigger cell cycle entry when needed and not in response to small fluctuations? A particular challenge when, as described by the authors, that cell cycle switch is thought to be controlled by a positive feedback loop between E2F and CDK2. Using a simple trick of slowing down S-phase entry by using a low dose of CDK4/6 inhibitor, the authors discover that phosphorylation of T373 in the transcriptional repressor protein, Rb, may underpin a range of intermediate E2F activity states. This trick enables the authors to investigate molecular events that may happen at fast timescales in cell culture models, but perhaps more slowly in vivo (presumably). Through a number of live and fixed single-cell imaging experiments, generation of a new E2F activity reporter (that is shown to be more E2F-specific than the previously published reporter), structural modelling of E2F-DP1-Rb interactions and characterisation of Rb mutants, the authors propose a model whereby Rb phosphorylation by CDKs in the Rb pocket region generates an intermediate state of E2F activation where cells are not yet committed to S-phase entry since Rb is still bound to chromatin. Only upon phosphorylation of C-terminal residues in Rb, is Rb released from chromatin, permitting “full” E2F activation through initiating the positive feedback loop between E2F and CDK. The authors suggest that these intermediate states of E2F activity persist since it appears that T373 (and S608) may be more slowly dephosphorylated than C-terminal Rb residues.

Overall, the paper is nicely written, hypotheses are clearly developed, experiments well executed, data nicely presented and well explained and the conclusions novel. There are a few places where extra controls need to be shown/performed to be confident in the results and these are outlined below. More could also be done to make a convincing case that it is slow dephosphorylation that allows T373 phosphorylation to persist. I also think it is essential that the authors address the work of Narasimha et al., eLife 2014 which first showed the existence of relatively long-lived Rb mono-phosphorylated species and called into question the simple E2F-CDK2 positive feedback model presented as the only model here. This paper is referenced in the middle of the paper but needs to be brought in much earlier since it represents existing data that already starts to refute the simple positive feedback model, yet the authors don't discuss this in detail at all.

Major points:

- I think it is still not clear, from the data shown, that CDK2 plays a major role in E2F activation, despite being mentioned as being crucial to the positive feedback in the text and the model. It is also not clear to me the importance of CDK2 in phosphorylating T373, particularly in a context where CDK4/6 is active. My reasoning is as follows:

o In Figure 1k, treatment with CDK2 inhibitor had only a very mild effect on the rate of E2F activation so how can CDK2 be driving the positive feedback that leads to high E2F activity?

o What happens to T373 phosphorylation if only CDK2 inhibitor is added?

o Is CDK2 required to maintain the high level of E2F activity? Presumably if the C-terminal Rb sites can be dephosphorylated quickly, then, in theory, Rb could rebind to E2F in S-phase if CDK2 activity was turned off. What happens to E2F activity after S-phase entry in the presence of CDK2i? The authors show EGFRi treatment post S-phase entry and see no effect (Fig. 2j). But in that same experiment where cells are still in G1, E2F activity decreases before CDK2 activity when EGFRi is added, suggesting that EGFRi could be acting directly at E2F and not via CDK2?

- The decrease in PRb/TotRb in Figures 4f and g is attributed to dephosphorylation of the residue. However, this is not formally shown. If phosphatase inhibitors are added to cells, does the level of phosphorylated Rb stay constant? Also, what do the total Rb levels look like? Do they change after CDK2i addition? It could be that phosphorylation of Rb on a particular residue changes Rb protein turnover and so the total Rb level changes.

- The model assumes that phosphatase activity towards Rb is constant (or not rate-limiting) during G1. The authors can (and should) test this by depleting Rb-targeting phosphatases which would, if they are correct, abolish the intermediate Rb phosphorylation states.

- I'm struggling to completely understand Figure 5a. If the argument is that when Rb is phosphorylated on S807/811 this kick starts the positive feedback loop to hyperphosphorylate Rb and evict Rb from chromatin, why is total-Rb still on chromatin (middle panels)? Presumably the total Rb bound to chromatin will be heterogeneous across the population (and may negatively correlate with the amount of chromatin-bound P-S807/811)?

- There is an important control missing in Figure 5c to show that they are overexpressing functional Rb here. The authors should deplete Rb1 and show that expressing Rb-WT restores Rb function (i.e. in this assay, E2F activity will increase after Rb1 siRNA (as seen in Figure 1g), but restoring Rb-WT should restore normal E2F kinetics (or, at least, decrease E2F activity again)).

- There is currently no validation that siRNA depletions, (many of the) inducible overexpressions or inhibitor treatments are working as expected. These must be shown by western blotting or immunostaining.

Minor points:

- The rationale of releasing cells into CDK4/6i to slow down G1 is given at the start of Figure 2, but similar experiments are already being performed in Figure 1 and Extended Data Figure 1 which makes those figures confusing to understand at first. This concept should be introduced earlier and it would be helpful to have a schematic on Extended Data Figure 1a as to how this experiment was

done and why it was done this way.

- Similarly, I am confused by the lack of heterogeneity in Extended Figure 2c. Do the authors have data in a similar format as in Extended Figure 2a for cells released with CDK4/6i? Presumably at least the Control conditions (when cells are released into CDK4/6i) are heterogeneous (especially since only ~45% of cells make it to S-phase by 40h post-release) yet there seems to be very little heterogeneity in timing in these figures? Have these cells been gated in some way?

- How was the Cdc6 promoter sequence deemed to be the optimal one? It isn't clear to me from the experiments shown in Extended Figure 1 how this conclusion was reached.

- In Figure 1e, would you not expect the E2F reporter (that is based on the Cdc6 promoter sequence) and Cdc6 mRNA to linearly correlate?

- Why is mRNA quantified as puncta area rather than, say, number of spots (Figure 1e)?

- It is mentioned that site T373 in Rb is conserved from Arabidopsis to human and a reference is given that also states this site is conserved. However, there is no evidence here or in that reference to show this. The authors should either reference work that shows conservation of T373 or show the conservation in this work.

- When was Doxycycline added in the inducible overexpression experiments?

- What is the rationale behind using Standard Error over Standard Deviation?

- In line 124 describing the mutagenesis analysis please put a space in the text between TD/MB and the references. I almost missed the references as they look like residue numbers.

- In Figure 2h, do the authors know why CDK2 activity initially increases after EGFRi addition?

- In Extended Figure 3c showing the robustness of the 4i technique, different cells are shown for the second round vs the first round (upper images). Shouldn't images of the same cells be shown to make their point?

- In Figure 4e, T373 and S608 phospho-antibodies have the strongest signal and they are also the ones where a signal at the hypo-phosphorylated Rb size is visible. Can the authors be certain that if they overexpose the other antibodies there is no signal at this smaller size?

Referee #2 (Remarks to the Author):

The authors developed a live cell reporter of E2F activity and use it with complimentary experiments to study the dynamics of E2F activation prior to S phase entry, which marks the cell commitment to division. They identify an “intermediate” E2F activity under a set of conditions designed to extend the G1 phase, and they show that this intermediate E2F activity is correlated with a population of the E2F inhibitor RB1 that is phosphorylated on a specific subset of CDK sites. The potential impact of this study is from the high significance of the CDK-RB1-E2F pathway in cell biology and cancer. In particular, the idea that E2F and RB1 can exist in an intermediate state of activity is a novel finding that would change current thinking about how the switch at the G1-S transition of the cell cycle operates. The study is technically sound, and the manuscript is well written. However, as described below, several key claims are not sufficiently supported by the data, and some conclusions are overstated. Another criticism is that the function of the “intermediate” activity discovered in this study is not sufficiently demonstrated. It is speculated that a “safe” period, in which E2F activity slowly rises or fluctuates, could be used to integrate signals and delay commitment to S phase. However, such signals and how they are integrated are not characterized, and the system is not perturbed to understand the impact on cell cycle progression. While the direct observation of fluctuating E2F activity is interesting, especially because it contrasts the field’s current thinking of switch-like behavior, the function and relevance are not established, limiting the impact of the work.

1) The authors state conclusions in terms that imply general and broad mechanisms of cell cycle and proliferation control, yet the key experiments are only performed in one cancer cell type and under conditions of chemical CDK4-6 inhibition. To increase the rigor of the study, several experiments, particularly those observing intermediate E2F activity and an order to RB1 phosphorylation, should be extended to other cell lines. The rationale for using cell culture with CDK4-6 inhibition as an experimental system and that the results under those conditions are more “physiological relevant” is not sufficiently justified. Behavior of MCF10A cells in the absence of CDK4-6 activity may be distinct from behavior of other nontransformed cell types that have low CDK4-6 activity and other cancer cells that are dependent on CDK2 activity for G1/S transition. The key experiments could be repeated in other cells, including primary cells or nontransformed cells such as fibroblasts or RPE cells (an examples common in cell cycle field that are thought of as “normal” cells and have “normal” CDK4-6 activity). The significance of the fact that the results are specific to conditions of CDK4-6 inhibition should be more fully addressed. In sum, while it is clearly established that intermediate E2F activity is possible, the argument that the intermediate state is broadly relevant for cell cycle control is extrapolated from observations made in one cell type and under conditions (low CDK4-6 activity and growth factor) that are designed to impede proliferation and cell cycle progression; this evidence alone is not convincing.

2) A concluding distinction is made from the data in Fig. 2 that during “intermediate” E2F activity the positive CDK-RB-E2F feedback loop is not engaged, while during times of parallel increasing CDK2 and E2F activity the positive feedback is engaged. This conclusion is not well justified and the significance unclear. Although not well described, it seems the rationale for the author’s connection of the data to feedback loop activation is that once positive feedback is engaged there would be no fluctuation of CDK2 and E2F activity. However, simultaneous fluctuation and/or intermediate activity of a positive feedback loop are possible if there are other inputs that could modulate the loop. A lack

of correlation of E2F and CDK2 would be some evidence that the positive feedback loop is not engaged during periods of intermediate activity but this is not sufficiently demonstrated. In contrast, it appears in Fig. 2C that the overall trend is that E2F activity tracks with and follows CDK2 activity, whether there is a delay in the increase (red lines, long period between E2F activation and S phase) or not. This also seems to be the case for two of the three individual cells in Fig. 2A. The CDK2 activity of the cells in Fig. 2G should be shown to observe more examples of how the activities correlate. The authors should explain the rationale for this conclusion better and offer more justification from the analysis.

3) The structural model in Fig. 3B is extraneous and should probably be removed. It is not used to make predictions or hypotheses, design experiments, or interpret data, and therefore does not add to the study. As a visual aid to understanding the two different previously identified E2F-RB1 interactions, it is distracting and hard to parse compared to panels 3A and 3E.

4) It is not certain whether the authors conclusion that “E2F is gradually activated by increasing CDK2 activity proportional to the fraction of RB1 phosphorylated at T373” is meant to point out a correlation or if there is further implication that T373 phosphorylation is necessary and/or sufficient for this activity. A correlation is a rather weak conclusion, and there is not enough evidence to support the latter claim. At a minimum, it would be useful to examine the correlation between E2F activity and phosphorylation at other specific sites (i.e. the data in Fig. 4D plotted for other S807/S811 phosphorylation, S608, other C-terminal sites). Ideally, one would observe the impact of RB1 phosphorylation site mutations in this experiment, for example using an RB1 knock-down re-expression system (as in PMID: 30711375) or isogenic knock-in.

5) The conclusion that “the western blot analysis supports the result from the single cell analysis that RB1 is phosphorylated at T373 and S608 before RB1 is hyperphosphorylated” is not well justified. The data show that under conditions of CDK4-6 inhibition, the population of hypophosphorylated RB1 containing T373 and S608 phosphorylation are minor yet significantly greater than the population of other phosphorylated species. The connection of this observation to a temporal ordering of phosphorylation events is tenuous. The Western blot analysis could be performed in a time course as was done in Fig. 3D. It should also be noted that similar analysis of diverse phosphorylated species in a hypophosphorylated RB1 population has been previously reported (PMID: 30711375 and PMID: 24876129); this previous work should be acknowledged and referenced here. As a minor point, the middle panel in Fig. 4E showing total RB1 is not explained in the caption.

6) The conclusion that “a reversible safety state of partial RB1 inactivation” and partial phosphorylation has been identified appears to be in contrast to this group’s conclusion from a previous study that RB1 is either hyperphosphorylated or not (PMID: 31543423). The different conclusions and the different experimental conditions (if any) should be addressed. Regardless, it is a shortcoming of this study that the consequences of phosphorylation of T373 and S608 to achieve partial RB1 inactivation prior to the other phosphorylation events is not explored.

7) In the analysis in Figs. 4A, 4B, and 4D more explanation of how the modeling was done and the rationale is needed. It is not clear why data in 4A and 4B were fit to a sigmoidal model, while the data in Fig. 4D were fit to a line with a rather poor R-squared value.

Referee #3 (Remarks to the Author):

In this study, Konagaya and colleagues try to understand how cells control when the positive feedback (E2F->Cdk2 -|RB -|E2F or E2F->Cdk2->E2F) that underlies G1/S-phase transition and commitment to proliferation is triggered. To do this, the authors developed a new activity sensor for the transcription factor E2F and measured the combined activities of Cdk2 and E2F in single cells. The authors propose that single cells maintain an intermediate level of E2F activity in which they can integrate proliferative and other cues and trigger positive feedback only when time is right. They show that this intermediate activity of E2F correlates with mono-phosphorylation of the E2F regulator RB at Threonine 373. The authors claim this site is phosphorylated by G1 CDKs (Cdk4/6 and Cdk2) and is slowly dephosphorylated. When T373 on RB is de-phosphorylated, cells undergo quiescence. The authors propose the slow dephosphorylation of T373 allows for E2F intermediate activity and to trigger the E2F->Cdk2->E2F positive feedback only when cells are committed to divide.

I was excited to read this manuscript. The study addresses an important and fundamental question in cell biology - how do cells avoid unruly cell division. The authors use single cell imaging approaches and biosensors and propose a new mechanism that explains why the classical E2F->Cdk2->E2F positive feedback isn't triggered in an unscheduled manner. The study is timely, the findings novel and of general interest. I think developing an E2F reporter is useful for the community.

I have, however, a few concerns that I believe are important to address in order to make sure the data fully supports the conclusions. Unfortunately, in some instances, the experimental design was not adequate. There are also gaps about the mechanistic details, how specific and how conserved the findings are and the functional importance of the intermediate E2F activity.

Please see below the detailed comments. It is my hope that these help improve the manuscript.

Major concerns

1. Experiments in the context of CDK4/6 inhibition

Many of the experiments are done in the context of Cdk4/6 inhibition (e.g. Figure 1, Figure 2, Figure 4, Figure 5). The logic behind this is a bit unclear and the explanation seems to be that the "activity of Cdk4/6 in MCF10A is too high comparatively to in vivo". There are several problems with doing experiments under Cdk4/6 inhibition but one of the main issues here is showed in Figure 2f where one can appreciate that CDK4/6 inhibition results in the appearance of the E2F "reverse" state. In other words, intermediate E2F. This is problematic since the novel finding of the study is that E2F activity can be in an intermediate state. It will be important to both a) perform these experiments in the context of asynchronous, cycling cells and b) perform these experiments in different cell lines. The former can be done with cell cycle sensors such as PCNA or the FUCCI sensor. It would be helpful if Cdk4/6 activity is shown for the different lines in b).

2. Specificity of intermediate E2F and RB T373 phosphorylation

Related to the point above, because experiments were not performed in unsynchronous cells it is not known whether the intermediate E2F and RB T373 mono-phosphorylation are specific to MCF10A cells coming out of serum starvation, under Cdk4/6 inhibition. What happens in normal cycling cells and importantly, what happens under stress and DNA damage? Understanding the latter

would be ideal since cells would not want to trigger the E2F → Cdk2 → E2F positive feedback under stress/DNA damage conditions

3. The mechanism underlying RB mono-phosphorylation of T373

Sanidas et al have a beautiful paper showing that RB is mono-phosphorylated in a variety of residues depending on context and that these mono-phosphorylations affect association with between RB and other proteins (Sanidas et al Mol Cell 2019). However, in Konagaya et al, the authors propose that RB is first phosphorylated only at T373 (Figure 3). This contradicts the results from Sanidas et al.

a) How do the authors reconcile the two studies?

b) Konagaya et al also propose that T373 is slowly dephosphorylated but no mechanism is shown for how this would happen. In a dynamic view of kinase:phosphatase inside the cell, we know that no phosphorylation stays “static” but is instead a result of distinct kinetics of phosphorylation and dephosphorylation reactions. What is the mechanism that gives rise to slow dephosphorylation? c) Finally, the authors nicely show that Cdk4/6 and Cdk2 phosphorylate T373. What about Cdk1? This site could be phosphorylated during the previous mitosis and the mechanism could potentially involve both kinase and phosphatase dynamics (and perhaps spatial localization).

4. Correlation, not causality, between E2F intermediate activity and RB T373 phosphorylation

The authors showed nice experiments using RB mutant to make T373 unable to be phosphorylated or mimic phosphorylation. However, these experiments were done with ectopically expressed constructs, where wt RB is present. It would be important to do these in the context of endogenous RB. If the model is true, what would happen if cells had constitutive endogenous RB mono-phosphorylation (T373EE/DD) or RB was unable to be phosphorylated (T373A) at T373? This would show some causality between the T373 and the buffering of positive feedback by intermediate levels of E2F (and entry into S-phase). Knock-ins are pretty straightforward in mammalian cells and could be a great way of testing causality.

5. Specificity of the E2F sensor. It isn't clear whether the E2F reporter is specific to any E2F isoform in particular. Different E2Fs can have distinct (even opposing) roles and it will be important to explore which isoforms the sensor reports activity of.

6. Does the intermediate E2F activity actually result in transcription? It will be important to understand what is the functional consequence of this reduced activation.

7. How are the intermediate levels of E2F maintained?

8. Figure 2 a. In the context of Cdk4/6 inhibition, E2F activity is supposed to rely on Cdk2 only. If this is the case why fluctuations of Cdk2 activity (e.g. middle panel) don't have any impact on the E2F sensor activity?

Minor concerns

9. Multiplexing IF is great but sometimes difficult to achieve unless antibodies are very good. It will be important to see 1 example of the raw images of the 4i protocol with the different RB phospho-antibodies used. Extended figure 3 seems to try and address this but unfortunately doesn't show the same field of view i.e. the same cells.

10. it would be good to expand on the reasons why choosing the Cdc6 promoter as readout for the E2F reporter. The authors show (extended figure 1) that several genes could in principle be used. It would be good to expand on the rationale beyond saying cdc6 does not report on Myc activity.

11. Sanidas et al Molecular Cell 2019 on “A code of mono-phosphorylation modulates the function of RB” should be cited.

Author Rebuttals to First Revision:

We clarify our new changes with yellow highlights in our revised manuscripts.

Referee #1 (Remarks to the Author):

Overview

In this work, Konagaya and colleagues aim to address an important question – how do cells only trigger cell cycle entry when needed and not in response to small fluctuations? A particular challenge when, as described by the authors, that cell cycle switch is thought to be controlled by a positive feedback loop between E2F and CDK2. Using a simple trick of slowing down S-phase entry by using a low dose of CDK4/6 inhibitor, the authors discover that phosphorylation of T373 in the transcriptional repressor protein, Rb, may underpin a range of intermediate E2F activity states. This trick enables the authors to investigate molecular events that may happen at fast timescales in cell culture models, but perhaps more slowly in vivo (presumably). Through a number of live and fixed single-cell imaging experiments, generation of a new E2F activity reporter (that is shown to be more E2F-specific than the previously published reporter), structural modelling of E2F-DP1-Rb interactions and characterisation of Rb mutants, the authors propose a model whereby Rb phosphorylation by CDKs in the Rb pocket region generates an intermediate state of E2F activation where cells are not yet committed to S-phase entry since Rb is still bound to chromatin. Only upon phosphorylation of C-terminal residues in Rb, is Rb released from chromatin, permitting “full” E2F activation through initiating the positive feedback loop between E2F and CDK. The authors suggest that these intermediate states of E2F activity persist since it appears that T373 (and S608) may be more slowly dephosphorylated than C-terminal Rb residues.

Overall, the paper is nicely written, hypotheses are clearly developed, experiments well executed, data nicely presented and well explained and the conclusions novel. There are a few places where extra controls need to be shown/performed to be confident in the results and these are outlined below. More could also be done to make a convincing case that it is slow dephosphorylation that allows T373 phosphorylation to persist. I also think it is essential that the authors address the work of Narasimha et al., eLife 2014 which first showed the existence of relatively long-lived Rb mono-phosphorylated species and called into question the simple E2F-CDK2 positive feedback model presented as the only model here. This paper is referenced in the middle of the paper but needs to be brought in much earlier since it represents existing data that already starts to refute the simple positive feedback model, yet the authors don't discuss this in detail at all.

Major points:

- I think it is still not clear, from the data shown, that CDK2 plays a major role in E2F activation, despite being mentioned as being crucial to the positive feedback in the text and the model. It is also not clear to me the importance of CDK2 in phosphorylating T373, particularly in a context where CDK4/6 is active. My reasoning is as follows:

o In Figure 1k, treatment with CDK2 inhibitor had only a very mild effect on the rate of E2F activation so how can CDK2 be driving the positive feedback that leads to high E2F activity?

We appreciate the reviewer bringing up the concern about the importance of CDK2 in activating E2F in the positive feedback. The reason why the CDK2 inhibitor has only a mild effect on E2F activation is that CDK4/6 can compensate for loss of CDK2 activity. We confirmed that CDK4/6 alone is capable of hyperphosphorylating Rb in the absence of CDK2 activity (added in Extended Data Fig. 10b, c) and thereby activating E2F, albeit less efficient and delayed (Fig. 1k), in the absence of CDK2 activity. The caveat here is that cultured cells often have hyperactive CDK4/6, due to p16 loss in the case of MCF-10A for instance. The importance of CDK2 is likely more pronounced in physiological settings, where CDK4/6 is less active compared to cultured cells: our previous *in vivo* study indicated that mouse intestinal epithelial cells use CDK2-dependent S phase entry, which is marked by APC/C^{Cdh1} inactivation before Rb hyperphosphorylation, even without CDK4/6 inhibition (Liu, et al. *Nature Communications*, 2020). Thus, we speculate that the importance of CDK2 is lower relative to CDK4/6 in cultured cell models and that CDK2 plays a more prominent role in activating E2F in many conditions *in vivo*, especially in adult tissue maintenance and renewal. We added the clarification in the Main text, line 62-63 in page 4.

o What happens to T373 phosphorylation if only CDK2 inhibitor is added?

Rb T373 phosphorylation levels were slightly reduced by the CDK2 inhibitor alone without the CDK4/6 inhibitor (added in Extended Data Fig. 10c). This is consistent with Figure 1k, where E2F activity was modestly suppressed by the CDK2 inhibitor alone. These results confirm that CDK2 contributes to Rb T373 phosphorylation and E2F activation even in the absence of CDK4/6 inhibition.

o Is CDK2 required to maintain the high level of E2F activity? Presumably if the C-terminal Rb sites can be dephosphorylated quickly, then, in theory, Rb could rebind to E2F in S-phase if CDK2 activity was turned off. What happens to E2F activity after S-phase entry in the presence of CDK2i?

We treated cells with the CDK2 inhibitor and computationally gated for the cells either in G1 phase or S phase based on the CRL4^{Cdt2} activity reporter signal that marks the start of S (the CRL4^{Cdt2} reporter is the latest version of the Fucci reporter that best marks the start of S; Sakaue-

Sawano, et al. *Molecular Cell*, 2017). We used this approach to test if CDK2 activity is required to maintain the high level of E2F activity in G1 and S. E2F activity decreased with the CDK2 inhibitor regardless of cell cycle phase, or CDK4/6i treatment (*without CDK4/6i*: added in Extended Data Fig. 3c, d. *with CDK4/6i*: added in Extended Data Fig. 4h-j), suggesting that CDK2 activity contributes to the maintenance of high E2F activity both in G1 phase and S phase. The suppression of E2F activity by CDK2i is not striking because CDK4/6 can compensate for CDK2 in G1, and CDK1 can compensate for CDK2 in late G1 to S. Please also note that E2F activity decreases after mid-S phase due to inhibitory E2Fs even without the CDK2 inhibitor treatment (Fig. 1d, bottom). We included these results in the Main text, line 58-59 in page 3.

The authors show EGFRi treatment post S-phase entry and see no effect (Fig. 2j). But in that same experiment where cells are still in G1, E2F activity decreases before CDK2 activity when EGFRi is added, suggesting that EGFRi could be acting directly at E2F and not via CDK2?

We first validated that EGFRi indeed suppresses EGFR phosphorylation (added in Extended Data Fig. 4a-c). As the reviewer pointed out, E2F is first suppressed before CDK2 by EGFRi likely through direct transcriptional regulation (Dong, et al. *Nature Communications*, 2014), in the condition where E2F is already activated to an intermediate level. We also explored a reason why CDK2 activity decrease is initially delayed after EGFRi addition and included our explanation in Extended Data Fig. 4d-g, m. Regardless, we confirmed that E2F is still under the control of CDK2 through EGFR signaling because CDK2i decreases Rb phosphorylation at T373 (added in Extended Data Fig. 4k, l) and E2F activity eventually (added in Extended Data Fig. 4h-j).

- The decrease in PRb/TotRb in Figures 4f and g is attributed to dephosphorylation of the residue. However, this is not formally shown. If phosphatase inhibitors are added to cells, does the level of phosphorylated Rb stay constant?

*The former Fig. 4f, g are now Fig. 5a, b.

We thank the reviewer for a great suggestion on Rb dephosphorylation. Given that Rb is dephosphorylated by PP1 and PP2A, we used calyculin A, a phosphatase inhibitor for PP1 and PP2A. The CDK2 inhibitor-induced dephosphorylation at T373 in Rb was indeed blocked by 1 nM of calyculin A treatment (added in Fig. 5d). Similarly, the dephosphorylation at S807/S811 was blocked by calyculin A treatment, but only at a higher dose (10 nM of calyculin A was required for S807/S811) (added in Fig. 5e). This result confirms that PP1 and PP2A dephosphorylate Rb at T373 and S807/S811 with a preference for S807/S811 over T373.

We also used a kinetic phosphorylation-dephosphorylation model to ask whether the observed differences in dephosphorylation kinetics between T373 versus S807/S811 could explain the preferential phosphorylation at T373 over S807/S811 (added in Fig. 5f). We assumed that the kinases for Rb (CDK4/6 and CDK2) do not have selectivity to T373 over S807/S811, since the marked convex relationship in the Phosphorylation Site Preference (PSP) plot does not change with the CDK4/6 inhibitor or the CDK2 inhibitor (DMSO: Fig. 6e, CDK4/6i: Fig. 3c, the leftmost panel, CDK2i: Extended Data Fig. 10b). Markedly, when we used this assumption and estimated the relative dephosphorylation coefficient (PSP_{coeff}) from the PSP plot, we found a range of PSP_{coeff} is 4.40 ± 0.31 , which is within the PSP_{coeff} range measured from the dephosphorylation kinetics assay, 6.79 ± 4.21 . Thus, a slower dephosphorylation rate at T373 relative to S807/S811 can explain why we are observing preferential phosphorylation at T373 over S807/S811.

We included these new data in the Main text, line 162-165 in page7, and line 238-251 in page 10.

Also, what do the total Rb levels look like? Do they change after CDK2i addition? It could be that phosphorylation of Rb on a particular residue changes Rb protein turnover and so the total Rb level changes.

To test whether phosphorylation of Rb on a particular residue changes Rb protein turnover, we measured the total-Rb levels after CDK2 inhibitor addition. The total-Rb levels remained unchanged for up to 1 hour after CDK2 inhibitor addition (added in Fig. 5c). Thus, we conclude that the differential decrease in phospho-Rb/total-Rb between T373 versus S807/S811 in Fig. 5a, b (formerly Fig. 4f, g) is mainly explained by dephosphorylation selectivity by PP1/PP2A, rather than a potential Rb protein turnover control mediated by phosphorylation at a certain residue. We included this result in the Main text, line 236-238 in page 10.

- The model assumes that phosphatase activity towards Rb is constant (or not rate-limiting) during G1. The authors can (and should) test this by depleting Rb-targeting phosphatases which would, if they are correct, abolish the intermediate Rb phosphorylation states.

We truly appreciated the reviewer's insightful suggestion and tried to see whether we could abolish the intermediate Rb phosphorylation state, but we were not able to prove this point because of technical difficulties. We took multiple approaches to deplete Rb-targeting phosphatases such as various timing and dose of PP1/PP2A-targeting siRNA and inhibitors. However, we observed no change in the intermediate Rb phosphorylation states with minor/shorter perturbations and cell death with stronger/longer perturbations due to cell

detachment and apoptosis. Even though we were not able to show it experimentally, our alternative approach using a kinetic phosphorylation-dephosphorylation model suggests that the intermediate Rb phosphorylation states may primarily result from a slower dephosphorylation rate at T373 relative to S807/S811 (added in Fig. 5f).

- I'm struggling to completely understand Figure 5a. If the argument is that when Rb is phosphorylated on S807/811 this kick starts the positive feedback loop to hyperphosphorylate Rb and evict Rb from chromatin, why is total-Rb still on chromatin (middle panels)? Presumably the total Rb bound to chromatin will be heterogeneous across the population (and may negatively correlate with the amount of chromatin-bound P-S807/811)?

*The former Fig. 5a is now Fig. 6a.

We apologize for not being clearer. The timepoint we selected, 36 hours after release with the CDK4/6 inhibitor, is when most of the cells are either in the intermediate state or in quiescence. We are identifying cells in this intermediate state if they still have chromatin-bound Rb (as marked by the remaining total-Rb stain after extraction) and are negative for p-S807/S811 (the middle panels in Fig. 6a) but still have T373 partially phosphorylated. In these experiments, we can stain for either p-T373 or p-S807/S811, due to technical difficulties in performing iterative indirect immunofluorescence imaging (4i) after pre-extraction. We added the clarification in the Main text, line 265-274 in page 11.

- There is an important control missing in Figure 5c to show that they are overexpressing functional Rb here. The authors should deplete Rb1 and show that expressing Rb-WT restores Rb function (i.e. in this assay, E2F activity will increase after Rb1 siRNA (as seen in Figure 1g), but restoring Rb-WT should restore normal E2F kinetics (or, at least, decrease E2F activity again)).

*The former Fig. 5c is now Fig. 6c.

We first compared cells with or without efficient Rb1 knockdown and confirmed that Rb1 knockdown increases E2F activity (added in Extended Data Fig. 9a, b; black versus green), consistent with Fig. 1g. We next induced Rb-WT expression in Rb1 knockdown cells and observed that E2F activity was suppressed (added in Extended Data Fig. 9a, b; green versus blue). Among those Rb-restored cells, we gated cells with the endogenous level of Rb expression (Rb^{high}; red) because Rb re-expression levels are heterogeneous. The gated cells with the endogenous level of Rb expression showed further suppressed E2F activity (added in Extended Data Fig. 9a, b; blue versus red), with the kinetics almost identical to the unperturbed control

cells (added in Extended Data Fig. 9a, b; red versus black). These results confirm that Rb-WT used in Fig. 6c, d (formerly Fig. 5c, d) is functional and that Rb re-expression levels are in the physiological range of the parental MCF-10A cells. We included these new data in the Main text, line 293-297 in page 12.

- There is currently no validation that siRNA depletions, (many of the) inducible overexpressions or inhibitor treatments are working as expected. These must be shown by western blotting or immunostaining.

We added the requested control experiments as follows:

1. Rb knockdown and Rb re-expression were confirmed by immunofluorescence for Rb (added in Extended Data Fig. 9a, b).
2. E2F1, E2F2, E2F3, and E2F7 knockdowns were confirmed by fluorescence in situ hybridization (FISH) (added in Extended Data Fig. 2d).
3. HA-tagged E1A, E2F1, Cyclin D1, Myc, Cyclin E1 overexpressions were confirmed by immunofluorescence for HA (E1A: Fig. 1h, E2F: added in Extended Data Fig. 1g, Cyclin D1: added in Extended Data Fig. 1h, Myc: Extended Data Fig. 1i, Cyclin E1: added in Extended Data Fig. 3h).
4. CDK2 inhibitor was confirmed by the CDK2 activity reporter (Extended Data Fig. 1j).
5. CDK4/6 inhibitor was confirmed by immunofluorescence for Rb phosphorylation at S807/S811 (added in Extended Data Fig. 1k).
6. EGFR inhibitor was confirmed by western blots for EGFR phosphorylation at Y1045 (added in Extended Data Fig. 4a-c).
7. The effectiveness of Neocarzinostatin (NCS), which induces DNA damage, was confirmed by 53BP1 puncta (added in Extended Data Fig. 8e).

Minor points:

- The rationale of releasing cells into CDK4/6i to slow down G1 is given at the start of Figure 2, but similar experiments are already being performed in Figure 1 and Extended Data Figure 1 which makes those figures confusing to understand at first. This concept should be introduced earlier and it would be helpful to have a schematic on Extended Data Figure 1a as to how this experiment was done and why it was done this way.

*The former Extended Data Fig. 1a is now Extended Data Fig. 1b.

We appreciate the useful suggestion from the reviewer. As the reviewer suggested, we are now introducing the rationale for the use of the CDK4/6 inhibitor earlier for better readability in the Main text, line 68-73 in page 4. Additionally, we now include a schematic of the rationale and experimental design for live-cell imaging (added in Fig. 1I) and RNA-seq (added in Extended Data Fig. 1b).

- Similarly, I am confused by the lack of heterogeneity in Extended Figure 2c. Do the authors have data in a similar format as in Extended Figure 2a for cells released with CDK4/6i? Presumably at least the Control conditions (when cells are released into CDK4/6i) are heterogeneous (especially since only ~45% of cells make it to S-phase by 40h post-release) yet there seems to be very little heterogeneity in timing in these figures? Have these cells been gated in some way?

*The former Extended Data Fig. 2a, c is now Extended Data Fig. 3a. f.

We appreciate the reviewer's comment on the lack of heterogeneity in Extended Data Fig. 3f (formerly Extended Data Fig. 2c). To show the heterogeneity in the cell population, we included representative single-cell traces (n = 5 cells per each condition) as in Extended Data Fig. 3e. Cells were not gated based on S phase entry in Extended Data Fig. 3e, f, but cells that entered S phase were gated in Extended Data Fig. 3g.

- How was the Cdc6 promoter sequence deemed to be the optimal one? It isn't clear to me from the experiments shown in Extended Figure 1 how this conclusion was reached.

To explain the rationale for the CDC6 promoter, we included the workflow for the E2F reporter development in Extended Data Fig. 1a. To optimize the promoter for the E2F transcriptional activity reporter, we selected 11 candidate genes that were upregulated by growth media (containing serum, EGF, and insulin) and downregulated by the CDK4/6 inhibitor (thus specific to the CDK-Rb-E2F pathway) in the RNA-Seq (in Extended Data Fig. 1b). We further validated the selected eleven genes by RT-qPCR for their responsiveness to growth media and the CDK4/6 inhibitor in Extended Data Fig. 1c, d. We next selected promoter regions from each gene based on a genome database, Ensembl (<https://www.ensembl.org>), and inserted the promoter regions into the prototype reporter construct such that the promoter drives mVenus expression. We evaluated the sensitivity and specificity of the reporter by live-cell imaging in Extended Data Fig. 1e. The fold change in DMSO indicates the sensitivity to growth media whereas the fold change in the CDK4/6 inhibitor shows the specificity to the CDK-Rb-E2F pathway. Among the

top three constructs (FAM111B, DTL, and CDC6) in these criteria, we decided to use the CDC6 promoter because it is free of cytotoxicity and has no background signals in quiescence. We added the explanation about the E2F reporter development in the Methods section.

- In Figure 1e, would you not expect the E2F reporter (that is based on the Cdc6 promoter sequence) and Cdc6 mRNA to linearly correlate?

The reviewer brought up a good point about the linearity of the E2F reporter signal in relation to mRNA puncta area in FISH. CDC6 is one of the strongest E2F target genes. Thus, CDC6 mRNA puncta area saturates quickly, especially in growth media, because the mRNA puncta overlap each other while the E2F reporter signal has a wider dynamic range. To account for mRNA puncta being overlapped, we now show CDC6 mRNA puncta area multiplied by puncta intensity versus mVenus intensity from the E2F reporter. The correlation at the population level is now linear (added in Extended Data Fig. 2f, g) although the correlation at the single-cell level is noisy, probably due to two parameters, mRNA puncta area and intensity. Additionally, we observed a linear correlation in CDC6 mRNA puncta area versus mVenus intensity from the E2F reporter by slowing down CDC6 induction by the CDK4/6 inhibitor and preventing CDC6 mRNA from overlapping each other (added in Extended Data Fig. 2h, i).

We included these new data in the Main text, line 46-47 in page 3.

- Why is mRNA quantified as puncta area rather than, say, number of spots (Figure 1e)?

The reason why we use mRNA puncta area rather than the number of spots is that we cannot reliably resolve single spots using our microscope setting with an x20 objective lens (used to get large cell numbers for several parallel conditions). Thus, we use puncta area and integrated intensity to better estimate mRNA expression levels in individual cells, where multiple mRNA puncta could happen to be near each other or overlap.

- It is mentioned that site T373 in Rb is conserved from Arabidopsis to human and a reference is given that also states this site is conserved. However, there is no evidence here or in that reference to show this. The authors should either reference work that shows conservation of T373 or show the conservation in this work.

We included the Multiple sequence alignment (MSA) to show the conservation of T373 in Extended Data Fig. 9c. Even though the conservation of T373 in Arabidopsis was implicated in

the previous literature, the MSA analysis made us realize that we can only confirm a clear conservation of the T373 flanking sequences in Rb across vertebrates and certain invertebrates (*Strongylocentrotus purpuratus*, the purple sea urchin). Thus, we excluded our statement about Arabidopsis from the Main text and revised line 355 in page 14. We again thank the reviewer for the comments, which led to the critical correction.

- When was Doxycycline added in the inducible overexpression experiments?

We apologize for not being clear about when Doxycycline was added. Dox was added 5 hours before release throughout the paper. We included the Dox addition timing information in the Figure Legends and Methods sections.

- What is the rationale behind using Standard Error over Standard Deviation?

We think it is useful to show both statistical significance as well as the variability in the cell population. We have now also included more single-cell traces to show the variability in Extended Data Fig. 3a, e, j, k.

- In line 124 describing the mutagenesis analysis please put a space in the text between TD/MB and the references. I almost missed the references as they look like residue numbers.

We thank the reviewer for pointing this out. We added spaces and put the references in brackets according to the Nature References guideline (in the Main text, line 144-145 in page 7).

- In Figure 2h, do the authors know why CDK2 activity initially increases after EGFRi addition?

We first validated that EGFRi indeed suppresses EGFR phosphorylation (added in Extended Data Fig. 4a-c), and explored a reason why CDK2 activity initially increases after EGFRi addition. CDK2 activity slightly increases briefly ~ 4 hours after EGFRi. We found that EGFRi suppresses p21 levels, while cyclin E levels remain unaffected during the first 4 h after EGFRi (added in Extended Data Fig. 4d, e). These results suggest that the transient CDK2 activation after the EGFRi treatment may be explained by the p21 decrease. In the long term (4 to 18 h after EGFRi), both E2F and CDK2 are inactivated because Cyclin E is also downregulated eventually (added in Extended Data Fig. 4f, g). We added our model explanation in Extended Data Fig. 4m

and mentioned in the Main text, line 117-118 in page 6.

- In Extended Figure 3c showing the robustness of the 4i technique, different cells are shown for the second round vs the first round (upper images). Shouldn't images of the same cells be shown to make their point?

*The former Extended Data Fig. 3c is now Extended Data Fig. 5c.

We appreciate the valid point brought up by the reviewer. We corrected it by showing the same cells for the first round and second round of 4i in Extended Data Fig. 5c.

- In Figure 4e, T373 and S608 phospho-antibodies have the strongest signal and they are also the ones where a signal at the hypo-phosphorylated Rb size is visible. Can the authors be certain that if they overexpose the other antibodies there is no signal at this smaller size?

To account for differences in each antibody affinity, we normalize the phospho-Rb intensity at the lower band by the matched phospho-Rb intensity at the upper band for each antibody. The quantification is shown in the bar graph on the right in Fig. 4e. Additionally, we adjusted the exposure of western images (more overexposed for T252, S780, and S788) so that the phospho-Rb intensity at the upper band is similar across different antibodies (Fig. 4e, left).

Referee #2 (Remarks to the Author):

The authors developed a live cell reporter of E2F activity and use it with complimentary experiments to study the dynamics of E2F activation prior to S phase entry, which marks the cell commitment to division. They identify an “intermediate” E2F activity under a set of conditions designed to extend the G1 phase, and they show that this intermediate E2F activity is correlated with a population of the E2F inhibitor RB1 that is phosphorylated on a specific subset of CDK sites. The potential impact of this study is from the high significance of the CDK-RB1-E2F pathway in cell biology and cancer. In particular, the idea that E2F and RB1 can exist in an intermediate state of activity is a novel finding that would change current thinking about how the switch at the G1-S transition of the cell cycle operates. The study is technically sound, and the manuscript is well written. However, as described below, several key claims are not sufficiently supported by the data, and some conclusions are overstated. Another criticism is that the function of the “intermediate” activity discovered in this study is not sufficiently demonstrated. It is speculated that a “safe” period, in which E2F activity slowly rises or fluctuates, could be used to integrate signals and delay commitment to S phase. However, such signals and how they are integrated are not characterized, and the system is not perturbed to understand the impact on cell cycle progression. While the direct observation of fluctuating E2F activity is interesting, especially because it contrasts the field’s current thinking of switch-like behavior, the function and relevance are not established, limiting the impact of the work.

1) The authors state conclusions in terms that imply general and broad mechanisms of cell cycle and proliferation control, yet the key experiments are only performed in one cancer cell type and under conditions of chemical CDK4-6 inhibition. To increase the rigor of the study, several experiments, particularly those observing intermediate E2F activity and an order to RB1 phosphorylation, should be extended to other cell lines. The rationale for using cell culture with CDK4-6 inhibition as an experimental system and that the results under those conditions are more “physiological relevant” is not sufficiently justified. Behavior of MCF10A cells in the absence of CDK4-6 activity may be distinct from behavior of other nontransformed cell types that have low CDK4-6 activity and other cancer cells that are dependent on CDK2 activity for G1/S transition. The key experiments could be repeated in other cells, including primary cells or nontransformed cells such as fibroblasts or RPE cells (an examples common in cell cycle field that are thought of as “normal” cells and have “normal” CDK4-6 activity). The significance of the fact that the results are specific to conditions of CDK4-6 inhibition should be more fully addressed. In sum, while it is clearly established that intermediate E2F activity is possible, the argument that the intermediate state is broadly relevant for cell cycle control is extrapolated from observations made in one cell type and under conditions (low CDK4-6 activity and growth factor) that are designed to impede proliferation and cell cycle progression; this evidence alone is not convincing.

We appreciate the reviewer for bringing up the concern regarding the generalizability of our work. To confirm the generalizability of Rb partial phosphorylation, we first performed immunofluorescence for Rb phosphorylation using two non-transformed human cell lines: retinal pigment epithelial cells, RPE-1 cells, and foreskin fibroblasts, BJ-5ta cells. RPE-1 and BJ-5ta cells were released with growth media without CDK4/6 inhibition. Both cell lines show a convex relationship in the Phosphorylation Site Preference (PSP) plot of T373 versus S807/S811 (added in Extended Data Fig. 6d, e; left), confirming that T373 is preferentially phosphorylated over S807/S811 in the other two non-transformed cell lines. In contrast, we observed a linear relationship in the PSP plot of T826 versus S807/S811 (added in Extended Data Fig. 6d, e; right), indicating that these C-terminal sites are phosphorylated at the same time as observed in MCF-10A cells. In addition to non-transformed cell lines, we examined the Rb phosphorylation in cycling cancer cells: human osteosarcoma U2OS cells. The PSP plot of T373 versus S807/S811 shows that the Rb partial phosphorylation state is only evident with CDK4/6 inhibition but not without CDK4/6 inhibition (added in Extended Data Fig. 6f) likely because Rb is hyperphosphorylated right after mitosis. Interestingly, even in U2OS cells, we observed a convex relationship between T373 and S807/S811 phosphorylation, which is contrasted with a linear relationship between T826 and S807/S811 phosphorylation. Thus, we conclude that preferential phosphorylation at T373 over C-terminal sites in Rb is generally observed across the four cell lines we tested; the precise fraction of cells in the Rb partial phosphorylation state varies across cell lines likely due to the differential CDK4/6 and CDK2 activity and Rb-targeting phosphatase activity.

We further tested the generalizability of the intermediate E2F activity by additional experiments tracking E2F activity changes in RPE-1 cells. We first generated and validated RPE-1 cells that stably express the E2F activity reporter (added in Extended Data Fig. 6a). As observed in MCF-10A cells, cells that enter S phase activate E2F to higher levels (labeled as ‘S enter’) whereas cells that do not enter S phase do not activate E2F at all or activate E2F to an intermediate level and eventually turn it off (labeled as ‘E2F reverse’) (added in Extended Data Fig. 6c). The fraction of E2F reverse cells in E2F-active cells increases with CDK4/6 inhibition (added in Extended Data Fig. 6b).

Altogether, these data confirm that intermediate E2F activity and Rb partial phosphorylation are generalizable across human non-transformed cell lines without CDK4/6 inhibition. We included these results in the Main text, line 101-102 in page 5, and line 172-177, page 8.

2) A concluding distinction is made from the data in Fig. 2 that during “intermediate” E2F activity the positive CDK-RB-E2F feedback loop is not engaged, while during times of parallel increasing CDK2 and E2F activity the positive feedback is engaged. This conclusion is not well justified and the significance unclear. Although not well described, it seems the rationale for the author’s connection of the data to feedback loop activation is that once positive feedback is engaged there would be no fluctuation of CDK2 and E2F activity. However, simultaneous

fluctuation and/or intermediate activity of a positive feedback loop are possible if there are other inputs that could modulate the loop. A lack of correlation of E2F and CDK2 would be some evidence that the positive feedback loop is not engaged during periods of intermediate activity but this is not sufficiently demonstrated. In contrast, it appears in Fig. 2C that the overall trend is that E2F activity tracks with and follows CDK2 activity, whether there is a delay in the increase (red lines, long period between E2F activation and S phase) or not. This also seems to be the case for two of the three individual cells in Fig. 2A. The CDK2 activity of the cells in Fig. 2G should be shown to observe more examples of how the activities correlate. The authors should explain the rationale for this conclusion better and offer more justification from the analysis.

We included more examples of single-cell traces of CDK2 activity that correspond to cells in Fig. 2g (added in Extended Data Fig. 3k). Some cells show similar dynamics between CDK2 activity and E2F activity (cell #1 and #3 in Fig. 2a). In contrast, the other cells exhibit different dynamics, where only CDK2 is fluctuating while E2F is monotonically increasing (cell #2 in Fig. 2a) for example. This implies that the positive feedback between Rb-E2F and cyclin E-CDK2 is not fully engaged yet (the positive feedback is not strong enough to cause Rb hyperphosphorylation) during the intermediate E2F activity state. The reason why CDK2 and E2F traces in Fig. 2c look smooth and lack fluctuations is because they are traces averaged over hundreds of cells with similar E2F activation timing relative to S phase entry. We also included some single-cell traces used to generate averaged traces in Fig. 2c (added in Extended Data Fig. 3j). In addition to single-cell dynamics of CDK2 and E2F activity, we confirmed that cells in the intermediate state are still sensitive to EGFR signaling, suggesting that the positive feedback between Rb-E2F and cyclin E-CDK2 is not yet strong enough to cause Rb hyperphosphorylation and sustain full E2F activity. Accordingly, we revised the Main text, line 109-113 in page 5 for clarification.

3) The structural model in Fig. 3B is extraneous and should probably be removed. It is not used to make predictions or hypotheses, design experiments, or interpret data, and therefore does not add to the study. As a visual aid to understanding the two different previously identified E2F-RB1 interactions, it is distracting and hard to parse compared to panels 3A and 3E.

We apologize that we were not clear why we needed the structure model in Fig. 3b. Even though previous structural studies have proposed two interaction sites in the Rb-E2F-DP complex, there has been no study to demonstrate how Rb and E2F-DP as a whole complex that covers Pocket and C-terminal domains in Rb, transactivation domain (TD) in E2F, and Marked Box (MB) domains in E2F and DP due to technical difficulties of crystallizing such a big complex of three proteins. Our analysis using AlphaFold2 suggested that Rb-E2F-DP as a heterotrimeric complex interacts through two interaction regions: (1) RbP and E2F^{TD} and (2) RbC and E2F^{MB}-DP^{MB}. It seemed noteworthy to us that the regions between RbP and RbC^{core} and E2F^{TD} and E2F^{MB} are largely disordered, suggesting that these two interaction sites could be regulated independently. We revised the Main text, line 141-142 in page 6 to clarify the insights that can be gained from

inspecting the Rb-E2F-DP complex structure model generated by AlphaFold2.

4) It is not certain whether the authors conclusion that “E2F is gradually activated by increasing CDK2 activity proportional to the fraction of RB1 phosphorylated at T373” is meant to point out a correlation or if there is further implication that T373 phosphorylation is necessary and/or sufficient for this activity. A correlation is a rather weak conclusion, and there is not enough evidence to support the latter claim. At a minimum, it would be useful to examine the correlation between E2F activity and phosphorylation at other specific sites (i.e. the data in Fig. 4D plotted for other S807/S811 phosphorylation, S608, other C-terminal sites). Ideally, one would observe the impact of RB1 phosphorylation site mutations in this experiment, for example using an RB1 knock-down re-expression system (as in PMID: 30711375) or isogenic knock-in.

We thank the reviewer for bringing up great ideas to validate our hypothesis. We performed experiments to confirm the correlation and causality of E2F activity and Rb phosphorylation.

First, to test whether the linear positive correlation between E2F activity and Rb phosphorylation is specific to T373 or is generally true to other sites, we examined the relationship between E2F activity and Rb phosphorylation at specific sites other than T373. We observed that Rb phosphorylation at C-terminal sites, S807/S811 and T826, shows an ultrasensitive relationship with E2F activity, whereas T373 (and less so for S608) shows a gradual correlation with E2F activity (added Extended Data Fig. 7b). This result highlights the differential regulation of T373 from the other sites.

Next, to test whether Rb phosphorylation at T373 is necessary and/or sufficient for E2F activity, we expressed Rb mutant constructs in cells where endogenous Rb was knocked down using the same system reported in Sanidas, et al. *Molecular Cell*, 2019 (PMID: 30711375). As a proof of concept, we first confirmed that E2F activity in cells expressing Rb- Δ CDK, where all the CDK-phosphorylation sites are mutated to alanines, is significantly lower than Rb-WT (Fig. 6c, d). In this Rb re-expression assay, E2F activity in cells expressing Rb-WT-3A (T373A/S608A/S612A) is significantly lower than Rb-WT and comparable to Rb- Δ CDK. This result suggests that the central phosphorylation in Rb at T373/S608/S612 is necessary to induce intermediate E2F activity. Conversely, we tested the sufficiency of the central phosphorylation in Rb for intermediate E2F activity using a reverse phosphomimetic mutant from Rb- Δ CDK. E2F activity in cells expressing Rb- Δ CDK-3D (T373D/S608D/S612D) is significantly higher than Rb- Δ CDK, suggesting that the central phosphorylation in Rb at T373/S608/S612 is sufficient to induce intermediate E2F activity.

We included these data in the Main text, line 205-206 in page 9, and line 291-315, page 12-13.

5) The conclusion that “the western blot analysis supports the result from the single cell analysis that RB1 is phosphorylated at T373 and S608 before RB1 is hyperphosphorylated” is not well

justified. The data show that under conditions of CDK4-6 inhibition, the population of hypophosphorylated RB1 containing T373 and S608 phosphorylation are minor yet significantly greater than the population of other phosphorylated species. The connection of this observation to a temporal ordering of phosphorylation events is tenuous. The Western blot analysis could be performed in a time course as was done in Fig. 3D. It should also be noted that similar analysis of diverse phosphorylated species in a hypophosphorylated RB1 population has been previously reported (PMID: 30711375 and PMID: 24876129); this previous work should be acknowledged and referenced here. As a minor point, the middle panel in Fig. 4E showing total RB1 is not explained in the caption.

As a clarification, we performed the western blot for Rb phosphorylation at the time course that matches with the live-cell imaging experiment in Fig. 3d (added in Fig.4f). Now we are able to show that Rb phosphorylation at T373 occurs first, followed by S608, and lastly by C-terminal phosphorylation at S807/S811 and T826 by western blots. We included these new data in the Main text, line 223-227 in page 9-10.

We cited the important previous papers earlier in the Main text, line 132-135 in page 6, and in Conclusions, line 356-366 in page 14-15.

We also thank the reviewer for catching the missing information in Fig. 4e. We included the explanation about total-Rb in the caption in Fig. 4e.

6) The conclusion that “a reversible safety state of partial RB1 inactivation” and partial phosphorylation has been identified appears to be in contrast to this group’s conclusion from a previous study that RB1 is either hyperphosphorylated or not (PMID: 31543423). The different conclusions and the different experimental conditions (if any) should be addressed. Regardless, it is a shortcoming of this study that the consequences of phosphorylation of T373 and S608 to achieve partial RB1 inactivation prior to the other phosphorylation events is not explored.

We included the explanation for the apparent discrepancy between our previous work (PMID: 31543423) versus this work in the Main text, line 318-321 in page 13. The main difference is that Chung, et al. used growth media (containing serum, EGF, and insulin), whereas in this study we used the low mitogen media (EGF only media as opposed to growth media) and the CDK4/6 inhibitor to slow down the G1/S transition to mimic more physiological conditions for most of the experiments. Moreover, even though it was not highlighted in the text, the convex relationship between T373 and S807/S811 was observed in growth media as well (Fig. 2D, a scatter plot on the right in Chung, et al. *Molecular Cell*, 2019; PMID: 31543423).

To investigate the consequences of central phosphorylation in Rb (T373, S608, and S612), we expressed Rb mutant constructs in cells where endogenous Rb was knocked down using the same system reported in Sanidas, et al. *Molecular Cell*, 2019 (PMID: 30711375) and measured the percentage of cells that entered S phase. As a proof of concept, we first confirmed that the S

phase entered fraction in cells expressing Rb- Δ CDK is significantly lower than Rb-WT (added in Fig. 6d, right). Consistent with E2F activity, the S phase entered fraction is significantly lower in cells expressing Rb non-phosphorylatable mutant in the central region, Rb-WT-3A (T373A/S608A/S612A), compared to Rb-WT. Conversely, the S phase entered fraction is significantly higher in cells expressing a reverse phosphomimetic mutant, Rb- Δ CDK-3D (T373D/S608D/S612D) compared to Rb- Δ CDK. In sum, these results indicate that the central phosphorylation in Rb is a limiting factor for S phase entry.

7) In the analysis in Figs. 4A, 4B, and 4D more explanation of how the modeling was done and the rationale is needed. It is not clear why data in 4A and 4B were fit to a sigmoidal model, while the data in Fig. 4D were fit to a line with a rather poor R-squared value.

We selected sigmoid functions for fitting in Fig. 4b because Rb phosphorylation levels as a function of CDK2 activity are not linear. In Fig. 4a, we did not use fitting, but we showed the median Rb phosphorylation levels for CDK2 activity bins.

To improve the linear regression fitting in Fig. 4d, we performed robust linear regression using the bisquare weights, getting a better R^2 value (figures below and added in Fig. 4d).

As a comparison, we also performed polynomial fitting and obtained slightly better R^2 values in second- and third-degree than first-degree polynomial fitting (figures below). We speculate that the experimental noise in Rb phosphorylation and E2F activity leads to the rather poor R^2 values.

Referee #3 (Remarks to the Author):

In this study, Konagaya and colleagues try to understand how cells control when the positive feedback (E2F->Cdk2 -|RB -|E2F or E2F->Cdk2->E2F) that underlies G1/S-phase transition and commitment to proliferation is triggered. To do this, the authors developed a new activity sensor for the transcription factor E2F and measured the combined activities of Cdk2 and E2F in single cells. The authors propose that single cells maintain an intermediate level of E2F activity in which they can integrate proliferative and other cues and trigger positive feedback only when time is right. They show that this intermediate activity of E2F correlates with mono-phosphorylation of the E2F regulator RB at Threonine 373. The authors claim this site is phosphorylated by G1 CDKs (Cdk4/6 and Cdk2) and is slowly dephosphorylated. When T373 on RB is de-phosphorylated, cells undergo quiescence. The authors propose the slow dephosphorylation of T373 allows for E2F intermediate activity and to trigger the E2F->Cdk2->E2F positive feedback only when cells are committed to divide.

I was excited to read this manuscript. The study addresses an important and fundamental question in cell biology - how do cells avoid unruly cell division. The authors use single cell imaging approaches and biosensors and propose a new mechanism that explains why the classical E2F->Cdk2->E2F positive feedback isn't triggered in an unscheduled manner. The study is timely, the findings novel and of general interest. I think developing an E2F reporter is useful for the community.

I have, however, a few concerns that I believe are important to address in order to make sure the data fully supports the conclusions. Unfortunately, in some instances, the experimental design was not adequate. There are also gaps about the mechanistic details, how specific and how conserved the findings are and the functional importance of the intermediate E2F activity. Please see below the detailed comments. It is my hope that these help improve the manuscript.

Major concerns

1. Experiments in the context of CDK4/6 inhibition

Many of the experiments are done in the context of Cdk4/6 inhibition (e.g. Figure 1, Figure 2, Figure 4, Figure 5). The logic behind this is a bit unclear and the explanation seems to be that the "activity of Cdk4/6 in MCF10A is too high comparatively to in vivo". There are several problems with doing experiments under Cdk4/6 inhibition but one of the main issues here is showed in Figure 2f where one can appreciate that CDK4/6 inhibition results in the appearance of the E2F "reverse" state. In other words, intermediate E2F. This is problematic since the novel finding of the study is that E2F activity can be in an intermediate state. It will be important to both a) perform these experiments in the context of asynchronous, cycling cells and b) perform these experiments in different cell lines. The former can be done with cell cycle sensors such as PCNA or the FUCCI sensor. It would be helpful if Cdk4/6 activity is shown for the different lines in b).

We appreciate the insightful feedback from the reviewer. To address the first point, we tested whether there is an intermediate E2F activity state in cycling MCF-10A cells without CDK4/6 inhibition. Extending our investigation to cycling cells led to our surprising finding that dephosphorylation at T373 is slower than S807/811 after anaphase (added in Fig. 5g-i, Extended Data Fig. 8a-c). Thus, right after mitosis, cells are born either with Rb partial phosphorylation at T373 to stay in an intermediate E2F activity state (added as (i) or (ii) in Fig. 5i and Extended Data Fig. 8b), or with Rb hyperphosphorylation to quickly activate E2F (added as (iii) in Fig. 5i and Extended Data Fig. 8b).

To address the second point, we generated another non-transformed human cell line expressing the E2F reporter using retinal pigment epithelial (RPE-1) cells (added in Extended Data Fig. 6a). Consistent with MCF-10A cells, we observed 'E2F reverse' population in RPE-1 cells as well (added in Extended Data Fig. 6b, c). Furthermore, we confirmed that T373 is preferentially phosphorylated over S807/S811 in three other cell lines: (1) retinal pigment epithelial cells, RPE-1 cells, (2) foreskin fibroblasts, BJ-5ta cells, and (3) osteosarcoma U2OS cells (added in Extended Data Fig. 6d-f).

Altogether, these additional data support that the intermediate E2F activity state is generalizable to broader contexts such as cycling cells without CDK4/6 inhibition and different cell lines. We included these new data in the Main text, line 101-102 in page 5, line 172-177, page 8, and line 252-258, page 11.

2. Specificity of intermediate E2F and RB T373 phosphorylation

Related to the point above, because experiments were not performed in unsynchronous cells it is not known whether the intermediate E2F and RB T373 mono-phosphorylation are specific to MCF10A cells coming out of serum starvation, under Cdk4/6 inhibition. What happens in normal cycling cells and importantly, what happens under stress and DNA damage? Understanding the latter would be ideal since cells would not want to trigger the E2F -> Cdk2 -> E2F positive feedback under stress/DNA damage conditions

We further tested whether the intermediate E2F activity state is generalizable in cycling cells under DNA-damaging conditions without CDK4/6 inhibition. As we mentioned in Major Concern #1, under normal conditions without DNA-damaging agents, cycling cells are born either with Rb partial phosphorylation at T373 or with Rb hyperphosphorylation. As the reviewer expected, neocarzinostatin (NCS), an agent that causes DNA strand breaks, ablated the daughter cell population with Rb hyperphosphorylation. Instead, almost all (>99%) daughter cells were born with partial Rb phosphorylation/intermediate E2F activity when DNA damage was induced in mother cells by NCS (added in Fig. 5g, h and Extended Data Fig. 8c).

Similarly as released MCF-10A cells with CDK4/6 inhibition, asynchronously cycling MCF-10A cells without CDK4/6 inhibition also show a convex relationship between T373 and

S807/S811 phosphorylation in the Phosphorylation Site Preference (PSP) plot (added in Extended Data Fig. 8d). Notably, the convex relationship in the PSP plot of T373 versus S807/S811 is consistent across different conditions (mitogen levels, CDK4/6 activity, and DNA damage levels) and different cell lines (added in Extended Data Fig. 6d-f), but the frequency of cells in the intermediate state increases in response to lower mitogens, downregulated CDK4/6 activity, and higher DNA damage. These data suggest that the order of Rb phosphorylation (preferentially occurs at T373 over other sites) is universal across different conditions and different cell lines, but the upstream signaling converges on CDK4/6 and CDK2 activity determines how quickly cells undergo the intermediate state, which determines the fraction of the intermediate cells in the population. We included these new data in the Main text, line 172-177 in page 8, and line 252-258, page 11.

3. The mechanism underlying RB mono-phosphorylation of T373

Sanidas et al have a beautiful paper showing that RB is mono-phosphorylated in a variety of residues depending on context and that these mono-phosphorylations affect association with between RB and other proteins (Sanidas et al Mol Cell 2019). However, in Konagaya et al, the authors propose that RB is first phosphorylated only at T373 (Figure 3). This contradicts the results from Sanidas et al. a) How do the authors reconcile the two studies?

b) Konagaya et al also propose that T373 is slowly dephosphorylated but no mechanism is shown for how this would happen. In a dynamic view of kinase:phosphatase inside the cell, we know that no phosphorylation stays “static” but is instead a result of distinct kinetics of phosphorylation and dephosphorylation reactions. What is the mechanism that gives rise to slow dephosphorylation? c) Finally, the authors nicely show that Cdk4/6 and Cdk2 phosphorylate T373. What about Cdk1? This site could be phosphorylated during the previous mitosis and the mechanism could potentially involve both kinase and phosphatase dynamics (and perhaps spatial localization).

The reviewer brings up a good point. In respect to (a), Sanidas, et al. *Molecular Cell*, 2019 showed that each mono-phosphorylated Rb isoform has differential protein binding partners and associated transcriptional outputs using Rb mutant constructs. However, it was not clear whether all of those mono-phosphorylated Rb isoforms exist endogenously, or in what order they are phosphorylated in cell cycle regulation. Their work is founded on an earlier paper, Narasimha, et al. *ELife*, 2014, which showed that Rb is mono-phosphorylated in early G1 where E2F is not yet activated. On the other hand, our work investigated the timing of Rb endogenous phosphorylation in cells and identified an orderly phosphorylation of Rb, first occurring at T373 in early G1. Furthermore, we demonstrated that Rb T373 phosphorylation has functional relevance in that it induces E2F activity and limits S phase entry (added in Fig. 6d). A potential interpretation of the earlier data is that Rb is initially mono-phosphorylated stochastically at 14 different CDK-targeting sites without E2F activation. Only once CDK4/6 and/or CDK2 activity reach high enough levels, does Rb become increasingly phosphorylated at T373, which induces intermediate E2F activity, before even higher CDK activity hyperphosphorylates Rb. Our

imaging-based assay is likely not sensitive enough to detect the initial Rb isoforms mono-phosphorylated at 14 different CDK-targeting sites reported in Narasimha, et al. *Elife*, 2014. We focused on cell cycle control, the G1/S transition, but did not explore Rb function outside of cell cycle control. We reconcile seemingly contradictory models of Rb phosphorylation and its function between our work versus Sanidas, et al. *Molecular Cell*, 2019 by considering that Rb may play pleiotropic functions depending on the context; we showed that T373 phosphorylation is linked to intermediate E2F activity in the G1/S transition, whereas Sanidas, et al. *Molecular Cell*, 2019 showed S811 or T826 phosphorylation stimulates the expression of oxidative phosphorylation genes, increasing cellular oxygen consumption. We cited these important previous papers earlier in the Main text, line 132-135 in page 6, and we are now reconciling our study and the previous studies in Conclusions, line 356-366 in page 14-15.

In regard to (b), we think that it is likely that Rb phosphatase(s) targets C-terminal phosphorylation sites in Rb by having higher affinity or better local catalytic access compared to the central phosphorylation sites, particularly T373. We added new data to more directly show that PP1/PP2A are responsible for dephosphorylating T373 more potently than S807/S811 (added in Fig. 5d, e). Yet, more structural studies and biochemical studies remain to be done to uncover the interaction of Rb-E2F-DP and PP1/PP2A.

In regard to (c), The reviewer brought up a good point about the potential contribution of CDK1 on T373 phosphorylation. T373 phosphorylation is indeed also partially decreased by a CDK1 inhibitor (added in Extended Data Fig. 4k, i), suggesting that CDK1 phosphorylates T373 in late G1 until mitosis. It has been proposed that Rb phosphorylation can persist through mitosis into daughters in a subpopulation of cycling MCF-10A cells (Moser, et al. *PNAS*, 2018; Yang, et al. *Elife*, 2020). In this study, we additionally found that daughter cells are born with either partially phosphorylated Rb at T373 without C-terminal phosphorylation or hyperphosphorylated Rb (added in Fig. 5g-i and Extended Data Fig. 8a-e). It is possible that the balance between CDK1 activity versus Rb phosphatase(s) activity during mitosis in the previous cell cycle determines Rb phosphorylation status after mitosis in the next cycle. We included these new data in the Main text, line 252-258 in page 11.

4. Correlation, not causality, between E2F intermediate activity and RB T373 phosphorylation

The authors showed nice experiments using RB mutant to make T373 unable to be phosphorylated or mimic phosphorylation. However, these experiments were done with ectopically expressed constructs, where wt RB is present. It would be important to do these in the context of endogenous RB. If the model is true, what would happen if cells had constitutive endogenous RB mono-phosphorylation (T373EE/DD) or RB was unable to be phosphorylated (T373A) at T373? This would show some causality between the T373 and the buffering of positive feedback by intermediate levels of E2F (and entry into S-phase). Knock-ins are pretty straightforward in mammalian cells and could be a great way of testing causality.

We initially tried constitutive expression of Rb mutants but had technical issues with growing those cell lines. Thus, we first knocked down the endogenous Rb expression and re-expressed Rb mutants in a doxycycline-inducible manner so that we can control the timing of Rb mutant expression (added in Extended Data Fig. 9a, b). Using this Rb mutant re-expression system, we assessed the effect of Rb phosphorylation on E2F activity and S phase entry (added in Fig. 6c, d). We summarize our data on E2F activity in Reviewer #2 comment (4), and S phase entry in Reviewer #2 comment (6). In short, a non-phosphorylatable mutant in the Rb central region showed significantly lower E2F activity and S phase entry compared to WT, whereas a phosphomimetic mutant in the Rb central region exhibited significantly higher E2F activity and S phase entry compared to Δ CDK mutant. These results are consistent with our model that phosphorylation of T373 (without C-terminal hyperphosphorylation) is necessary and sufficient to partially inactivate Rb and activate E2F. We included these new data in the Main text, line 291-315, page 12-13.

5. Specificity of the E2F sensor. It isn't clear whether the E2F reporter is specific to any E2F isoform in particular. Different E2Fs can have distinct (even opposing) roles and it will be important to explore which isoforms the sensor reports activity of.

The reviewer brings up an important point. We validated that knockdown of activating E2Fs (E2F1, 2, and 3) abrogates the E2F reporter signal in G1 (added in Extended Data Fig. 2a) and overexpression of E2F1 increases the E2F reporter signal in G1 (Fig. 1i), whereas knockdown of a repressing E2F (E2F7) upregulates E2F reporter signal in S/G2 (added in Extended Data Fig. 2b, c). Furthermore, we confirmed that the CDC6 promoter region used in the E2F reporter shows enrichment of all E2Fs (E2F1-8) using the publically available ChIP-Seq data (added in Extended Data Fig. 2e). Thus, our E2F reporter is likely detecting both activating and repressing E2Fs' activity, and it can be used as a global E2F transcriptional activity reporter. We included these new data in the Main text, line 51-53 in page 3.

6. Does the intermediate E2F activity actually result in transcription? It will be important to understand what is the functional consequence of this reduced activation.

To validate whether intermediate E2F activity results in E2F target gene transcription, we performed mRNA FISH for the E2F target, CDC6, in MCF-10A cells released with EGF and the CDK4/6 inhibitor. We first confirmed that almost no CDC6 mRNA was detected in quiescent cells starved for two days. We next observed that CDC6 mRNA levels in G1 cells 36 hours after release, when most of the cells are in an intermediate E2F activity state, fall between quiescent cells and cells that entered S phase (added in Extended Data Fig. 3i). Additionally, we confirmed that CDC6 mRNA levels are linearly correlated with E2F activity and indeed induced at a lower level in low E2F activity range (added in Extended Data Fig. 2f-i). These results support our model of the intermediate E2F activity state where cells decide whether to fully engage the positive feedback and enter S phase or turn off E2F to stay in quiescence. We included these new

data in the Main text, line 96-97 in page 5.

7. How are the intermediate levels of E2F maintained?

We postulate that at least two mechanisms underlie the maintenance of the intermediate E2F activity state; (1) an only gradual and fluctuating increase in CDK2 activity (in Fig. 2a) and (2) potent dephosphorylation of Rb C-terminal sites by PP1/PP2A but not the central T373/S608 sites (Fig. 5a, b) explain the slow buildup of Rb phosphorylation (added in Fig. 3d and 4f). These mechanisms enable E2F activity to increase slowly (a few hours to 20 hours) and either continue to increase for cells to enter S phase or be turned off to undergo quiescence.

8. Figure 2 a. In the context of Cdk4/6 inhibition, E2F activity is supposed to rely on Cdk2 only. If this is the case why fluctuations of Cdk2 activity (e.g. middle panel) don't have any impact on the E2F sensor activity?

We thank the reviewer to bring up an important point. E2F activity increase should solely triggered by CDK2 activity in CDK4/6 inhibited condition, given that co-treatment of the CDK4/6 inhibitor and CDK2 inhibitor completely abrogates E2F activity induction. However, CDK2 and E2F activity dynamics are not necessarily correlated until the positive feedback is fully engaged in late G1. This is because Rb phosphorylation levels are determined by kinase-phosphatase competition (CDK2 versus PP1/PP2A activity), not just by CDK2 activity. Furthermore, once E2F is activated, E2F can be self-amplified through transcription of activating E2Fs. We included the clarification in the Main text, line 109-113 in page 5 for clarification.

Minor concerns

9. Multiplexing IF is great but sometimes difficult to achieve unless antibodies are very good. It will be important to see 1 example of the raw images of the 4i protocol with the different RB phospho-antibodies used. Extended figure 3 seems to try and address this but unfortunately doesn't show the same field of view i.e. the same cells.

*The former Fig. 3c, g are now Fig. 5c.

We appreciate the valid point brought up by the reviewer. We corrected it by showing the same cells for the first round and second round of 4i in Extended Data Fig. 5c.

10. it would be good to expand on the reasons why choosing the Cdc6 promoter as readout for the E2F reporter. The authors show (extended figure 1) that several genes could in principle be used. It would be good to expand on the rational beyond saying cdc6 does not report on Myc activity.

We described the workflow of development of the E2F activity reporter (added in Extended Data Fig. 1a), providing the rationale for why we use the CDC6 promoter for E2F transcriptional activity reporter. Please kindly refer to our response to Reviewer #1 comment on minor point (3) for more details. The reason why many promoters of E2F target genes except CDC6 did not serve as an E2F activity reporter is likely because regulatory regions on the chromosome beyond promoter regions are also necessary to output E2F activity. We added the explanation about the E2F reporter development in the Methods section.

11. Sanidas et al *Molecular Cell* 2019 on “A code of mono-phosphorylation modulates the function of RB” should be cited.

We cited the paper, Sanidas et al., *Molecular Cell*, 2019 in the Main text, line 132-135 in page 6, and now also added a discussion highlighting the results from this study in Conclusions, line 356-366 in page 14-15.

Reviewer Reports on the First Revision:

Referees' comments:

Referee #1 (Remarks to the Author):

The reviewers have addressed my concerns and my view is that the manuscript has significantly improved.

The main drawback could be viewed as the use of the low CDK4/6 activity state to mimic a more "physiological" state. However, the authors are clear throughout about their experimental setup and include additional experiments without the CDK4/6 inhibitors. Therefore, I do not think this is concern.

Minor typos:

- Extended Data Figure 4 - legend for this figure has mislabelled (I) as (i).

Referee #2 (Remarks to the Author):

In their revised manuscript, the authors have added a significant number of experiments to further support their claims. Many of the reviewer points have been well addressed, and it is compelling, for example, that several key observations were reproduced in other cell lines. On the other hand, I have a few remaining concerns about some critical points that were not well considered.

1) The functional relevance of the partial E2F activity is not explored beyond the experiment in once cell line, which was presented in the original manuscript, showing cells can exit the cell cycle from the intermediate state upon inhibition of EGFR. The conclusion, as in the title of the manuscript, that this partial E2F activity "safeguards proliferation commitment" is still not sufficiently supported. This was a major concern of the original manuscript that remains unaddressed.

2) The authors have still not sufficiently established the specific connection of T373 phosphorylation to intermediate E2F activity. i) To address this criticism of the original manuscript, the authors thoughtfully used the system in Sanidas et al to express mutants, but they only demonstrate that phosphorylation of those sites is necessary and sufficient for overall E2F activity at a single time point and S phase entry. Analysis of single cell dynamics as in Fig. 2 should be performed to test whether the intermediate activity depends on those sites. ii) It is claimed that T373 phosphorylation is proportional to E2F activity, and the authors were asked to address whether this correlation is also true of phosphorylation at other sites. In response, the authors included data for other sites in Extended Data Figure 7b, but those data are not described in the main text beyond the figure citation. In the rebuttal, it is argued that the other sites show an ultrasensitive response, but this interpretation is neither justified nor included in the manuscript. These data need to be analyzed and explained more to demonstrate that T373 (and S608 if true) is unique in its ability to generate an intermediate and proportional response. Also, it is confusing that these data and graphs look

different than the data in Fig. 4D, and the graphs are not sufficiently well explained.

3) There are still concerns regarding the conclusion that the dephosphorylation rates of T373 and other sites are different. i) What are the errors associated with the half-life values in Fig. 5a and 5b and are the differences significant? ii) The conclusion that the difference in loss of phosphospecific antibody signal and calyculin A potency reflects a phosphatase substrate preference for T373 over S807/S811 relies on a number of assumptions that are never clarified, including that the compound is a competitive inhibitor for substrate binding (need reference for this), that both PP1 and PP2A are similar in their activities towards these sites and similarly inhibited by the compound, that there is no indirect effect of Cdk2 inhibition on phosphatase activity (notably, there is a regulatory Cdk site in the PP1 catalytic domain), that there are no other kinases phosphorylating these sites, etc. iii) The data in Fig. 5d are not a convincing demonstration that there is a difference in T373 dephosphorylation rate upon addition of 1 nM calyculin A. Only the last time point seems to show a difference, and it is strange that the data for no inhibitor at 60 mins is so much less noisy than all the similar data throughout Fig. 5. It should also be specified in the caption what the asterisks mean. iv) In Fig. 5h, it is not clear that the data should be fit to an exponential that goes to zero considering the signal for T373 antibody never disappears. Is there a control or experiment that rules out background staining? This experiment is also noisy. What are the errors associated with the half-life values? v) The quantitative comparison of PSP coefficients in Fig. 5f does not add anything beyond the qualitative observation that dephosphorylation rates are different. Given the simple model and assumption that phosphorylation rates are similar, the dephosphorylation must be different. That it is observed to be different in Fig. 5a is important for supporting the model but that the quantified PSP coefficients from the two different experiments (dephosphorylation rate and phosphorylation site correlation) are not statistically different is not compelling considering the noisy data. Also, it is not clear at all what is being shown in Fig. 5f and how it relates to R, which is defined as dephosphorylation rate in the caption. If not taken out, the figure, its caption, and the analysis in the text all needs to be explained much better. In sum, the strong conclusion about enzymatic behavior from these measurements in cells remains dubious.

Author Rebuttals to Second Revision:

The added changes are highlighted in yellow in the revised manuscripts.

Referee #1

The reviewers have addressed my concerns and my view is that the manuscript has significantly improved.

The main drawback could be viewed as the use of the low CDK4/6 activity state to mimic a more "physiological" state. However, the authors are clear throughout about their experimental setup and include additional experiments without the CDK4/6 inhibitors. Therefore, I do not think this is concern.

Minor typos:

- Extended Data Figure 4 - legend for this figure has mislabelled (l) as (i).

Thank you very much for the encouraging comments and for catching the mislabelled figure legends in Extended Data Figure 4. It has been corrected.

Referee #2

In their revised manuscript, the authors have added a significant number of experiments to further support their claims. Many of the reviewer points have been well addressed, and it is compelling, for example, that several key observations were reproduced in other cell lines. On the other hand, I have a few remaining concerns about some critical points that were not well considered.

Thank you very much for the positive comments. We have addressed the remaining concerns.

1) The functional relevance of the partial E2F activity is not explored beyond the experiment in once cell line, which was presented in the original manuscript, showing cells can exit the cell cycle from the intermediate state upon inhibition of EGFR. The conclusion, as in the title of the manuscript, that this partial E2F activity “safeguards proliferation commitment” is still not sufficiently supported. This was a major concern of the original manuscript that remains unaddressed.

Thank you for bringing up this point. We agree that using safeguard in the title is not critical for the manuscript and that the functional relevance of reversible E2F activation in G1 is better discussed later in the manuscript. We therefore replaced the previous title with “Partial Rb phosphorylation allows for reversible E2F activation in G1 phase” in the revised manuscript.

We would nevertheless like to add two points:

First, we now explain that E2F activation in MCF-10A cells can be partial, variable, and reversible in G1 phase in more detail in Fig. 2h, i. By measuring the S phase entry transition using the CRL4^{Cdt2} reporter in the same cell lines, we think that this analysis can address the functional role of having an intermediate E2F activation in G1. We show that those cells that only partially activate E2F do not enter S. Cells can keep E2F activity at an intermediate level for over 20 hours (marked as “undecided” in Fig. 2f, g) or they can reversibly turn E2F activity off after many hours in G1 (marked as “E2F reverse”).

We also examined E2F activity dynamics in G1 phase using a second cell line, RPE-1 cells, and added the data in Fig. 2j. We expressed the E2F reporter in these cells and performed live-cell imaging followed by fixed imaging for DNA content and EdU incorporation measurements. This assay allowed us to correlate the preceding E2F activity dynamics with the cell cycle phase in the same cell. We again asked whether the cell had entered S phase or was still in G1 or G0 (quiescence). Similarly to MCF-10A cells, RPE-1 cells often reversed the E2F activity after a transient increase in G1 without S phase entry. We included these results in the Main text, line 105-109 of page 5. Thus, the observation that E2F can be partially and reversibly activated in G1 phase without cells entering S phase is not restricted to a single cell type or condition.

Fig. 2
Second, we now discuss in the manuscript why we believe that an important function of the period of partial E2F activation is to ensure that cells commit to the cell cycle and enter S phase only if external and internal conditions are optimal. Previous work has shown that cells can integrate external receptor and stress signals during this time. These studies showed that stress signals and mitogen removal can cause fractions of cells to reverse from G1 phase to quiescence until, but not after, the start of S phase (Heldt, et al. *Proceedings of the National Academy of Sciences*, 2018; Schwarz, et al. *Molecular Cell*, 2018; Cappell, et al. *Cell*, 2016; Chung, et al. *Molecular Cell*, 2019). The conclusion from these studies was that mitogen and stress signals control whether the cell enters S phase or exits to quiescence in part by regulating CDK2 activity in G1 phase. Our study now adds to this earlier work by showing that E2F activity is reversibly regulated in G1 phase, showing that cells continue to sense mitogen and stress signals in G1 phase by regulating both CDK2 and E2F activity.

As mentioned by the reviewer, we have also directly tested the hypothesis that mitogens control E2F in by addition of an EGFR inhibitor in G1 phase (Fig 2k-m, formerly Fig. 2h-j). These measurements showed that the E2F activity increase can be acutely reversed during G1 phase simply by the loss of growth factor receptor signaling. We also included data showing that cells that inactivated E2F in G1 can later reactivate E2F to enter S phase (Extended Data Fig. 8a, b).

The benefit of having a period of signal integration for CDK2 and E2F activation is also supported by additional literature data. Work studying the link from ERK signaling to the activation of FRA1, which controls cyclin D expression, needs to be integrated over many hours to mediate cell cycle entry (Albeck, et al. *Molecular Cell*, 2013). Thus, an often long integration period is needed to activate E2F and start S phase while stress signals or inhibition of growth factor receptor signals may stop S phase entry more acutely. In addition, theoretical work (Selimkhanov, et al. *Science*, 2014) has shown that temporal integration for a longer period makes signaling decisions more reliable, which is consistent with the interpretation that signal integration in G1 phase may help cells safeguard cell cycle commitment.

We agree with the reviewer that the best place for discussing the function of partial E2F activation is later in the manuscript, thus now include it in the Main text, line 377-384 of page 15.

2) The authors have still not sufficiently established the specific connection of T373 phosphorylation to intermediate E2F activity. i) To address this criticism of the original manuscript, the authors thoughtfully used the system in Sanidas et al to express mutants, but they only demonstrate that phosphorylation of those sites is necessary and sufficient for overall E2F activity at a single time point and S phase entry. Analysis of single cell dynamics as in Fig. 2 should be performed to test whether the intermediate activity depends on those sites. ii) It is claimed that T373 phosphorylation is proportional to E2F activity, and the authors were asked to address whether this correlation is also true of phosphorylation at other sites. In response, the authors included data for other sites in Extended Data Figure 7b, but those data are not described in the main text beyond the figure citation. In the rebuttal, it is argued that the other sites show an ultrasensitive response, but this interpretation is neither justified nor included in the manuscript. These data need to be analyzed and explained more to demonstrate that T373 (and S608 if true) is unique in its ability to generate an intermediate and proportional response. Also, it is confusing that these data and graphs look different than the data in Fig. 4D, and the graphs are not sufficiently well explained.

The reviewer brings up two good points and we address them separately below:

i) Following the suggestion from the reviewer, we did analyze E2F activity dynamics in single cells for cells expressing the different mutant Rb constructs (added in Extended Data Fig. 9c). Nevertheless, the heterogenous nature of G1 phase progression and variation if and when cells enter S phase makes it difficult to gain additional insights from showing all the single-cell E2F activity traces beyond what we can gain by analyzing the average mitogen-induced E2F increase (marked by red lines).

We want to emphasize that a key part of our analysis of the mutant Rb constructs is that we are gating individual cells by selecting only those cells for the analysis that express the same amount of the mutant or wildtype Rb construct as a control. We are then comparing the effect that the same level of mutant Rb has on E2F activity across different Rb constructs. This gating is based on fixing cells at the end of the live-cell imaging experiment and monitoring the expressed HA-tagged Rb in every cell (all the Rb mutant and control constructs have the HA tag) and by

selecting only individual cells with the same Rb expression for the analysis. We then averaged the E2F traces of all cells with the same expression level of the respective Rb construct. The trend of a relative increase in E2F activity among the Rb constructs is the same irrespective of a time point after release, and thus we are showing in Fig. 6d the averaged E2F activity at one time point (48 hours after release) to simplify data representation.

The observation that the inhibitory effect of the mutant Rb constructs is only partial is not surprising since the previous work by the Sanidas group and others have shown that the Δ CDK construct (with 15 sites mutated to Ala) can only partially suppress the G1/S progression (Sanidas, et al. *Molecular Cell*, 2019; Topacio, et al. *Molecular Cell*, 2019).

In our study, the effect from expressing either the wildtype (WT) or the Δ CDK Rb constructs represents the minimal and maximal suppression of E2F signals in this assay, respectively. This analysis has a dynamic range of about a factor of 2, which is sufficient to see significant effects of the mutants. Likely explanations for the lack of full inhibition of E2F activity by induction of Δ CDK could for example be incomplete knockdown of the endogenous Rb or the presence of other members of the Rb family proteins (p107 and p130).

The main point we wanted to make in this figure is that the same amount of WT-3A (T373A/S608A/S612A) Rb mutant has an equally suppressive effect as the Δ CDK Rb mutant and a much stronger effect than wildtype Rb control in suppressing E2F activity. This is consistent with the interpretation that preventing the phosphorylation of the central sites (T373, S608, and S612 sites) is sufficient to prevent Rb from being inactivated under these conditions. The Asp phosphomimetic mutant of the same three sites further suggests that the phosphorylation of these sites can inactivate Rb and restore E2F activation. The effects were statistically significant across biological replicates. Thus, even with the limitation of the Rb mutant assay, we believe that this Rb mutant data is useful in supporting our model and complementing the previous structural data that the T373 and S608 sites in Rb can release one of the two inhibitory interactions between Rb and E2F.

Extended Data Fig. 9

ii) We thank the reviewer for the great suggestions. To test whether T373 and S608 are unique in their ability to generate an intermediate E2F activity, we now measured Rb phosphorylation levels at a particular site along with E2F activity at the single cell level, and fit the data using Deming regression which computes a linear slope of the Rb phosphorylation versus E2F activity (added in Fig. 4d). Deming regression can account for errors in both X and Y, which is needed

because Rb phosphorylation levels and E2F activity have both biological and experimental noise. When comparing hundreds of single cells, the analysis shows a clear difference between the central phosphorylation sites versus the C-terminal phosphorylation sites in the initial slope of the increase in Rb phosphorylation versus E2F activity. T373 and S608 show shallow slopes, meaning that Rb is gradually phosphorylated at central sites up to about the half maximum of the respective sites in proportion to the E2F activity increase. In contrast, S807/S811 and T826 show ~10 times steeper slopes, suggesting that the C-terminal sites are only minimally phosphorylated while E2F increases to the same intermediate level.

This analysis is also useful because it shows the cooperative nature of the second step in the Rb phosphorylation, concomitant with a further increase in E2F activity. This second step is evidenced by the jump in the single-cell data points at the green dotted lines (E2F activity >200, AU) from cells with minimal S807/S811 and T826 phosphorylation to near maximal phosphorylation (hyperphosphorylation). Both T373 and S608 are also fully phosphorylated above the green dashed line once cells have hyperphosphorylated Rb, showing that all sites including T373 participate in the hyperphosphorylation.

The former Extended Data Fig. 7b looks different from Fig. 4d because cells released with CDK4/6i are plotted in Fig. 4d, whereas cells released with or without CDK4/6i are plotted in the former Extended Data Fig. 7b in order to populate the Rb phosphorylation versus E2F activity plots with cells with the wider range of Rb phosphorylation status (unphosphorylated Rb, partially phosphorylated Rb, hyperphosphorylated Rb). We removed the former Fig. 4d and replaced it with the new panels of Rb phosphorylation versus E2F activity plots because it would be redundant to show both and the new panels are more informative than the former Fig. 4d. We now included our explanation in the Main text, line 210-223 in page 9.

Fig. 4

3) There are still concerns regarding the conclusion that the dephosphorylation rates of T373 and other sites are different. i) What are the errors associated with the half-life values in Fig. 5a and 5b and are the differences significant? ii) The conclusion that the difference in loss of phosphospecific antibody signal and calyculin A potency reflects a phosphatase substrate preference for T373 over S807/S811 relies on a number of assumptions that are never clarified, including that the compound is a competitive inhibitor for substrate binding (need reference for this), that both PP1 and PP2A are similar in their activities towards these sites and similarly

inhibited by the compound, that there is no indirect effect of Cdk2 inhibition on phosphatase activity (notably, there is a regulatory Cdk site in the PP1 catalytic domain), that there are no other kinases phosphorylating these sites, etc. iii) The data in Fig. 5d are not a convincing demonstration that there is a difference in T373 dephosphorylation rate upon addition of 1 nM calyculin A. Only the last time point seems to show a difference, and it is strange that the data for no inhibitor at 60 mins is so much less noisy than all the similar data throughout Fig. 5. It should also be specified in the caption what the asterisks mean. iv) In Fig. 5h, it is not clear that the data should be fit to an exponential that goes to zero considering the signal for T373 antibody never disappears. Is there a control or experiment that rules out background staining? This experiment is also noisy. What are the errors associated with the half-life values? v) The quantitative comparison of PSP coefficients in Fig. 5f does not add anything beyond the qualitative observation that dephosphorylation rates are different. Given the simple model and assumption that phosphorylation rates are similar, the dephosphorylation must be different. That it is observed to be different in Fig. 5a is important for supporting the model but that the quantified PSP coefficients from the two different experiments (dephosphorylation rate and phosphorylation site correlation) are not statistically different is not compelling considering the noisy data. Also, it is not clear at all what is being shown in Fig. 5F and how it relates to R, which is defined as dephosphorylation rate in the caption. If not taken out, the figure, its caption, and the analysis in the text all needs to be explained much better. In sum, the strong conclusion about enzymatic behavior from these measurements in cells remains dubious.

We appreciate these suggestions from the reviewer. We are now providing a more detailed statistical analysis and explanation of the plots in Fig. 5.

i) For consistency, we are now showing the standard error (SE) using a shaded area in the fit of the kinetic analysis. The errors for the fit are asymmetric and are shown as superscript/subscript numbers in the legends. We further added in the analysis that the dephosphorylation half-life of T373 is slower than those of S807/S811 and T826 using an asterisk nomenclature to mark significance. We clarified the meaning of the asterisk in the figure legends ($p < 0.05$). We fit our data using an exponential decay function:

$$y = ae^{-\lambda t}$$

The half-life is given by:

$$t_{1/2} = \frac{\log_e 2}{\lambda}$$

The errors for λ are symmetric, while the errors for $t_{1/2}$ are asymmetric. For better visual comparison, we have now consolidated the former Fig. 5a and 5b into one panel (the new Fig. 5a).

Fig. 5

a 42 h after release with CDK4/6i

ii) The reviewer brought up important considerations regarding the hypothesis on phosphatase substrate preference for S807/S811 over T373. We now included the important assumptions that the reviewer pointed out as follows:

(1) the primary kinases in G1 cells in our assay are CDK4/6 and CDK2: we now clarified: “..., we measured the time-course of the phosphorylated Rb loss after inhibiting CDK2 activity in G1 cells under the CDK4/6-inhibited conditions.” in the Main text, line 249-250 in page 10.

(2) calyculin A is a competitive inhibitor for substrate binding and has similar potency toward PP1 and PP2A: we now included the assumption: “..., we used calyculin A, a competitive inhibitor for substrate binding (Wakimoto, et al., Natural Product Reports. 2016) with a similar potency for PP1 and PP2A (Ishihara, et al., Biochem. Biophys. Res. Commun. 1989).” in the Main text, line 257-258 in page 11.

(3) PP1/PP2A activity is regulated by CDK1/2: our data shows that there is a marked difference in the dephosphorylation rate between the two sites after we acutely inhibited CDK2 activity (in cells with low CDK4/6 activity). We did this by adding the new selective CDK2 inhibitor from Pfizer PF-07104091 to cells that had Rb already phosphorylated. Our main point in this analysis is that dephosphorylation of Rb starts immediately at all sites but that the rate of dephosphorylation of S807/S811 is 6.79 ± 2.15-fold faster than that of T373. We also approached the same question without using CDK2 inhibition by monitoring the dephosphorylation kinetics of the same sites after mitosis and confirmed that S807/S811 is 6.65 ± 1.05-fold more rapidly dephosphorylated compared to T373.

We now included the explanation: “We also evaluated the phosphatase preference in an independent assay, without using CDK2 inhibitors, by measuring the rate of dephosphorylation of the same Rb sites after mitosis. Markedly, T373 is dephosphorylated 6.65 ± 1.05-fold more slowly than S807/S811 after anaphase (Fig. 5e, f), which resulted in newborn cells with Rb partially phosphorylated at T373 (Fig. 5g and Extended Data Fig. 8a-c). These experiments further suggest that the known regulation of PP1 and PP2A activity by CDK1/2 (Salvi, et al., Chembiochem, 2021; Nasa, et al. Science Signaling, 2020) cannot explain the differential dephosphorylation rates of T373 versus S807/S811 (Fig. 5a, e, f).” in the Main text, line 278-284 in page 11-12.

iii) In response to the valid point of the reviewer, we measured earlier time points at 30 min and 45 min for T373 dephosphorylation and tested for statistically significant differences at the different time points for T373 and S807/S811 dephosphorylation with and without calyculin A. This was needed to test whether there is a significant difference already before the 60-minute time point in the dephosphorylation of T373. The T373 phosphorylation time course after CDK2 inhibitor and calyculin A addition (red line) stays significantly higher than that without calyculin A treatment (blue line) at both 45 min and 60 min. For the S807/S811 dephosphorylation, the effect of inhibiting calyculin A (green line) is significantly higher than control (blue line) at 15 and 30 min. We also clarified the meaning of the asterisk in the figure legends ($p < 0.05$, $p < 0.01$, and $p < 0.001$ for the single, dual, and triple asterisks, respectively).

Fig. 5

iv) We apologize for the lack of clarity in the figure panels we showed. The reason why the T373 phosphorylation signal never disappears in the former images is because the cytokinesis cells were shown as the last time point, which happens approximately 20 min after anaphase. We also examined the images of T373 phosphorylation in cells 120 min after anaphase and confirmed that the T373 phosphorylation signal disappears completely, which is consistent with the dephosphorylation kinetics plot of T373 in Fig. 5f (formerly Fig. 5h). We thank the reviewer for noticing this and now included the representative images of a cell 120 min after anaphase as the last panel in Fig. 5e (formerly Fig. 5g).

Fig. 5

v) We thank the reviewer for this suggestion. We removed the panel from the main figure and are now describing the mean values and standard errors (SEs) of the phosphorylation site preference (PSP) coefficient and the relative dephosphorylation rate in the Main text. The mean values and SEs are 4.4 ± 0.15 for the PSP coefficient and 6.79 ± 2.15 for the relative difference in the dephosphorylation kinetics. These values are comparable and both of them are significantly different from 1.0. Thus, this analysis is consistent with our model that PP1/PP2A preference in dephosphorylating S807/S811 over T373 could provide a plausible explanation of why cells have a higher relative steady-state phosphorylation of T373 at a low/intermediate kinase activity. However, we are now addressing that the sizable noise in the analysis does not exclude the possibility that differences in kinase, PP1, or PP2A selectivity could also contribute to the phosphorylation site preference (PSP) for T373 over the different C-terminal phosphorylation sites. We included it in the Main text, line 268-277 of page 11.

Referee #2's comment on Referee #3's point #1

Overall both find the response convincing, except one note from referee #2 with regards to point 1, where they say that that in the rebuttal, there is a comment that you "tested whether there is intermediate activity in cycling {i.e. without Cdk4/6i} MCF10 cells" but the rebuttal didn't point to a figure this experiment is in, and the referee couldn't find it.

Thank you for pointing out this omission. We now included single-cell traces of E2F activity in cycling MCF-10A cells in the right two panels in Extended Data Fig. 8a. We found that, after mitosis, cells are born in an intermediate state (category (i) and (ii)) or fully active state (category (iii)). Daughter cells born with partial Rb phosphorylation stay in an intermediate E2F activity state for ~2 hours after mitosis. Then these cells either dephosphorylate Rb and turn off E2F (category (i)) or hyperphosphorylate Rb and fully activate E2F (category (ii)). In contrast, the other subset of cells, born with hyperphosphorylated Rb, sharply increases E2F activity for proliferation (category (iii)). Thus, we concluded that there is an intermediate state in cycling MCF-10A also without CDK4/6 inhibition, and included these data in the Main text, line 105-109 in page 5. Due to space constraints, we included these figure panels in the Extended Data Figure, not in the Main Figure.

Extended Data Fig. 8

a Asynchronously cycling cells

Reviewer Reports on the Second Revision:

Referees' comments:

Referee #2 (Remarks to the Author):

The reviewers have addressed all remaining concerns as possible. The manuscript should be published, but a few conclusions remain overstated and should be tempered given the lack of support in the study. Here are some important examples:

- 1) Lines 31-34 in abstract. Not sufficient data showing signals are being integrated by cells during this intermediate E2F activity.
- 2) Lines 210-211. The experiments test whether partial E2F activity and T373 phosphorylation are correlated, not whether T373 is responsible for partial E2F activity, as the opening sentence of the paragraph suggests.
- 3) Lines 284-289. Comments are speculative and should be moved to the discussion.
- 4) Lines 304-305. As above, the experiments correlate T373 phosphorylation to another observation but do not test whether T373 "is responsible for" the observation.
- 5) Lines 369-370. Not sufficient data showing phosphorylation sites are "converging integration points."

Author Rebuttals to Second Revision:

We also changed the text according to the requests by Referee #2:

The reviewers have addressed all remaining concerns as possible. The manuscript should be published, but a few conclusions remain overstated and should be tempered given the lack of support in the study. Here are some important examples:

1) Lines 31-34 in abstract. Not sufficient data showing signals are being integrated by cells during this intermediate E2F activity.

We removed the signal integration point.

2) Lines 210-211. The experiments test whether partial E2F activity and T373 phosphorylation are correlated, not whether T373 is responsible for partial E2F activity, as the opening sentence of the paragraph suggests.

We made the change.

3) Lines 284-289. Comments are speculative and should be moved to the discussion.

We adjusted the text.

4) Lines 304-305. As above, the experiments correlate T373 phosphorylation to another observation but do not test whether T373 “is responsible for” the observation.

We changed the text.

5) Lines 369-370. Not sufficient data showing phosphorylation sites are “converging integration points.”

We changed the text